# Non-convex entropic mean-field optimization via Best Response flow

**Razvan-Andrei Lascu**
RIKEN AIP
razvan-andrei.lascu@riken.jp

**Mateusz B. Majka**
Heriot-Watt University
m.majka@hw.ac.uk

## Abstract

We study the problem of minimizing non-convex functionals on the space of probability measures, regularized by the relative entropy (KL divergence) with respect to a fixed reference measure, as well as the corresponding problem of solving entropy-regularized non-convex-non-concave min-max problems. We utilize the Best Response flow (also known in the literature as the fictitious play flow) and study how its convergence is influenced by the relation between the degree of non-convexity of the functional under consideration, the regularization parameter and the tail behaviour of the reference measure. In particular, we demonstrate how to choose the regularizer, given the non-convex functional, so that the Best Response operator becomes a contraction with respect to the $L^1$-Wasserstein distance, which ensures the existence of its unique fixed point that is then shown to be the unique global minimizer for our optimization problem. This extends recent results where the Best Response flow was applied to solve convex optimization problems regularized by the relative entropy with respect to arbitrary reference measures, and with arbitrary values of the regularization parameter. Our results explain precisely how the assumption of convexity can be relaxed, at the expense of making a specific choice of the regularizer. Additionally, we demonstrate how these results can be applied in reinforcement learning in the context of policy optimization for Markov Decision Processes and Markov games with softmax parametrized policies in the mean-field regime.

## 1 Introduction

We consider the problem of minimizing an entropy-regularized, non-convex functional $F : \mathcal{P}_1(\mathbb{R}^d) \to \mathbb{R}$ over the Wasserstein space $\left(\mathcal{P}_1(\mathbb{R}^d), \mathcal{W}_1\right)$, that is,

$$\min_{\nu \in \mathcal{P}_1(\mathbb{R}^d)} F^\sigma(\nu), \text{ with } F^\sigma(\nu) := F(\nu) + \sigma \operatorname{KL}(\nu|\xi), \tag{1}$$

where the function $F$ is bounded below, i.e., $\inf_{\nu \in \mathcal{P}_1(\mathbb{R}^d)} F(\nu) > -\infty$, $\xi \in \mathcal{P}_1(\mathbb{R}^d)$ is a fixed reference probability measure with finite first moment and $\sigma > 0$ is a regularization parameter, while KL and $\mathcal{W}_1$ denote the KL-divergence (relative entropy) and the $L^1$-Wasserstein distance, respectively.

In recent years, there has been considerable interest in such problems, motivated by their applications in machine learning, including the task of training two-layer neural networks (NNs) in the mean-field regime [18, 33, 9, 10, 31, 37, 35] and Reinforcement Learning (RL) [1, 24, 43, 28]. In particular, several works studied problem (1) in a setting where $F$ is assumed to be convex, by utilizing various gradient flows such as the Best Response/fictitious play flow [8, 32], the Wasserstein gradient flow [18, 33], or the Fisher-Rao gradient flow [25].

In these works, the convergence of the flow to the minimizer of (1) holds for any value of $\sigma > 0$ (although the convergence rate may degenerate when $\sigma$ approaches zero) and for an arbitrary reference

39th Conference on Neural Information Processing Systems (NeurIPS 2025).

measure $\xi$, which indicates that for a convex $F$, one needs to add *any* regularization to ensure convergence. The exact form of the regularizer matters only for the convergence rate, but not for the sheer fact that the convergence holds. In the present paper we focus on the Best Response flow and we make an observation that, in order to ensure its convergence, the assumption about the convexity of $F$ is superfluous. One can demonstrate convergence for a non-convex $F$, as long as a specific choice of an appropriate regularizer is made. This will be precisely characterized by a formula that links the degree of non-convexity of $F$, the value of $\sigma$ and the first moment of the reference measure $\xi$, see (10) in Section 2 and the discussion therein. Hence we extend the results from [8] to non-convex functions $F$.

On the other hand, as motivation for this extension, we study applications in RL in the context of policy optimization for Markov Decision Processes (MDPs) with softmax parametrized policies with single-hidden-layer neural network in the mean-field regime [1, 24, 43, 28]. In order to explain the main ideas, let us focus on the simplified setting of a one-state MDP (also known in the literature as the bandit problem [1, 30]; for the general setting, see Section 2.2), where our goal is to minimize the function

$$F(\nu) = \int_A c(a)\pi_\nu(\mathrm{d}a), \tag{2}$$

for a cost function $c : A \to \mathbb{R}$, where $A$ is a Polish space denoting the player's action space and the measure $\pi_\nu$ represents the policy. Note that in the context of the bandit problem, one typically considers the goal of maximizing the reward, but here for the sake of notational consistency with [8] and the related optimization literature discussed above, we consider the equivalent problem of minimizing the cost. In the setting of (2), for any measure $\nu \in \mathcal{P}_1(\mathbb{R}^d)$, the mean-field softmax policy $\pi_\nu \in \mathcal{P}^\eta(A)$ is defined by

$$\pi_\nu(\mathrm{d}a) \propto \exp\left(\int_{\mathbb{R}^d} f(z,a)\nu(\mathrm{d}z)\right)\eta(\mathrm{d}a), \tag{3}$$

where $\mathcal{P}^\eta(A)$ denotes the set of probability measures that are absolutely continuous with respect to the strictly positive finite reference measure $\eta$ on $A$. In formula (3), the function

$$\mathcal{P}_1(\mathbb{R}^d) \times A \ni (\nu, a) \mapsto \int_{\mathbb{R}^d} f(z,a)\nu(\mathrm{d}z)$$

represents a mean-field neural network with a bounded, continuous, non-constant activation function $f : \mathbb{R}^d \times A \to \mathbb{R}$, whereas $\nu \in \mathcal{P}_1(\mathbb{R}^d)$ denotes a measure representing the parameters of the network. For a detailed discussion about the mean-field approach to MDPs, see [24]. Note that the normalization constant in (3) renders the objective in (2) non-convex. As a result, standard mean-field optimization results [33, 9, 8, 25], which rely on the convexity of $F$ do not guarantee convergence to a global minimizer.

In this work, we address (1) using the Best Response flow, a learning algorithm originally proposed in [14, 29] for computing Nash equilibria in games with finite-dimensional strategy spaces (see also [16, 17] for the analysis of its convergence in that setting). Only recently has it been introduced as an optimization method for minimizing entropy-regularized convex functionals over the space of probability measures [8], and for computing mixed Nash equilibria (MNEs) in two-player and multi-player zero-sum games with entropy-regularized convex-concave payoff functions [23, 21, 27]. The Best Response flow is defined by

$$\mathrm{d}\nu_t = \alpha\left(\Psi_\sigma[\nu_t] - \nu_t\right)\mathrm{d}t, \quad \nu|_{t=0} := \nu_0 \in \mathcal{P}_1(\mathbb{R}^d), \tag{4}$$

where $\alpha > 0$ is the learning rate of the flow and $\Psi_\sigma$ is the Best Response operator.

**Definition 1.1** (Best Response operator). *For any $\nu \in \mathcal{P}_1(\mathbb{R}^d)$ and any $\sigma > 0$, define the Best Response operator* $\Psi_\sigma : \mathcal{P}_1(\mathbb{R}^d) \to \mathcal{P}_1^\xi(\mathbb{R}^d)$ *by*

$$\Psi_\sigma[\nu](\mathrm{d}x) \propto \exp\left(-\frac{1}{\sigma}\frac{\delta F}{\delta \nu}(\nu, x)\right)\xi(\mathrm{d}x). \tag{5}$$

Here $\frac{\delta F}{\delta \nu}(\nu, \cdot)$ represents the flat derivative of $F$ at $\nu \in \mathcal{P}_1(\mathbb{R}^d)$ (cf. Definition F.1), whereas $\mathcal{P}_1^\xi(\mathbb{R}^d)$ is the space of probability measures on $\mathbb{R}^d$ that are absolutely continuous with respect to $\xi$ and have finite first moment. The Best Response operator (5) was introduced as part of the definition of the flow (4) in [8], but it appeared also in [33, 9] under the name *proximal operator* as a tool for studying convergence of mean-field Langevin dynamics.

**Remark 1.2.** *We note that there exists a dynamic known as the fictitious play flow (originally proposed in [4] for finite-dimensional games), which is closely related to the flow* (4), *and which can be interpreted as the best response computed with respect to the historical average of strategies. We refer to [23, Remark 2.6] for a detailed comparison between the two flows in the context of mean-field min-max problems.*

Following [8, Definition 3], we observe that for a given $\nu \in \mathcal{P}_1(\mathbb{R}^d)$ and $\sigma > 0$, the map $\Psi_\sigma$ satisfies the following variational representation:

$$\Psi_\sigma[\nu] = \operatorname*{argmin}_{\nu' \in \mathcal{P}_1^\xi(\mathbb{R}^d)} \left\{ \int_{\mathbb{R}^d} \frac{\delta F}{\delta \nu}(\nu, x)(\nu' - \nu)(\mathrm{d}x) + \sigma \operatorname{KL}(\nu'|\xi) \right\}.$$

This is a direct consequence of the first-order condition characterizing the minimizers of mean-field optimization problems (see [18, Proposition 2.5] and the discussion in [8, Section 3]). Hence $\Psi_\sigma$ is the unique minimizer (i.e., *Best Response*) of entropy-regularized linearization of $F$. As Proposition C.1 will demonstrate, every minimizer $\nu_\sigma^*$ of $F^\sigma$ must satisfy the fixed-point equation $\nu = \Psi_\sigma[\nu]$.

Our results in the MDP setting of Section 2.2 are closely related to [24], which considers the same MDP under the assumption that $\xi \propto e^{-U^\xi}$, where $U^\xi : \mathbb{R}^d \to \mathbb{R}$ is a measurable function, and utilizes the Wasserstein gradient flow to solve (1). In the present paper we will demonstrate that, compared to the results in [24], one can obtain exponential convergence to the minimizer for the Best Response flow, under weaker assumptions than the ones required in [24] for the Wasserstein flow. The comparison between our results and [24] will be discussed in detail in Section 2.3.

## 1.1 Non-convex-non-concave min-max problems

We extend our approach from the single-agent optimization setting to the two-player min-max setting by studying an entropy-regularized, non-convex–non-concave, zero-sum game:

$$\min_{\nu \in \mathcal{P}_1(\mathbb{R}^d)} \max_{\mu \in \mathcal{P}_1(\mathbb{R}^d)} F^{\sigma_\nu, \sigma_\mu}(\nu, \mu), \text{ where } F^{\sigma_\nu, \sigma_\mu}(\nu, \mu) := F(\nu, \mu) + \sigma_\nu \operatorname{KL}(\nu|\xi) - \sigma_\mu \operatorname{KL}(\mu|\rho), \tag{6}$$

with a function $F : \mathcal{P}_1(\mathbb{R}^d) \times \mathcal{P}_1(\mathbb{R}^d) \to \mathbb{R}$, reference measures $\xi, \rho \in \mathcal{P}_1(\mathbb{R}^d)$ and regularization parameters $\sigma_\nu, \sigma_\mu > 0$. A solution $(\nu_{\sigma_\nu}^*, \mu_{\sigma_\mu}^*)$ to (6) is referred to as a mixed Nash equilibrium (MNE) and can be characterized by the condition

$$\nu_{\sigma_\nu}^* \in \operatorname*{argmin}_{\nu \in \mathcal{P}_1(\mathbb{R}^d)} F^{\sigma_\nu, \sigma_\mu}(\nu, \mu_{\sigma_\mu}^*), \quad \mu_{\sigma_\mu}^* \in \operatorname*{argmax}_{\mu \in \mathcal{P}_1(\mathbb{R}^d)} F^{\sigma_\nu, \sigma_\mu}(\nu_{\sigma_\nu}^*, \mu).$$

Problem (6) arises in a variety of machine learning contexts, including the training of Generative Adversarial Networks (GANs) [13, 42, 26, 22, 2, 5, 23], adversarial robustness [38] or Multi-Agent Reinforcement Learning (MARL) [44, 12, 7, 21], and has been the subject of extensive theoretical investigation. In particular, several works considered problem (6) under the assumption that $F$ is either bilinear or convex-concave, employing various gradient flow dynamics, including the Best Response/fictitious play flow [23, 21], the Wasserstein gradient flow [26, 42, 13, 2, 5], and the Fisher–Rao gradient flow [22]. Following the single-agent optimization setting presented above, we consider the Best Response flow for (6) and demonstrate that the convexity-concavity of $F$ is not needed for convergence. Specifically, we show that convergence can still be guaranteed in the non-convex-non-concave setting, provided that appropriately chosen regularizers are employed. This relationship is made precise through a formula that connects the degree of non-convexity-non-concavity of $F$, the regularization parameters $\sigma_\nu$ and $\sigma_\mu$, the first moments of the reference measures $\xi$ and $\rho$ and, in comparison to the single-agent optimization case, additionally the learning rates of the players' flows $\alpha_\nu$ and $\alpha_\mu$. From this standpoint, our results can be interpreted as an extension of the analysis in [23] to the case of non-convex-non-concave objective functions $F$ with distinct regularization parameters and learning rates for each player.

Another key motivation for investigating the non-convex–non-concave min–max problem stems from applications in Multi-Agent Reinforcement Learning (MARL), particularly in the setting of policy optimization for two-player zero-sum Markov games [7, 21]. Specifically, we extend the examples presented in [21], which cover the convex–concave setting of a two-player zero-sum Markov game without policy parametrization, to the more general case of non-convex–non-concave objective functions with softmax parametrized policies, analogous to those encountered in the single-agent

optimization framework presented above. A detailed comparison between our results and those of [21] is provided in Section E.2. In the simplified setting of a two-player bandit problem, we take

$$F(\nu, \mu) = \int_A \int_A c(a, b) \pi_\nu(\mathrm{d}a) \pi_\mu(\mathrm{d}b) \tag{7}$$

for a cost function $c : A \times A \to \mathbb{R}$, where $A$ is the players' action space. In (7), for any measures $\nu$, $\mu \in \mathcal{P}_1(\mathbb{R}^d)$ the policies $\pi_\nu, \pi_\mu \in \mathcal{P}^\eta(A)$ are given by (3) for $\pi_\nu$ and by an analogous formula for $\pi_\mu$. Similarly to the single-agent bandit problem, the objective function $F$ is non-convex-non-concave due to the parametrization. As a result, existing mean-field optimization results for computing mixed Nash equilibria (MNEs) in min–max games with convex–concave payoff functions [26, 5, 2, 23, 21, 22] do not, in general, guarantee convergence to a global MNE.

Following the optimization setting, we tackle (6) by utilizing the Best Response flow, which is defined as
$$\begin{cases} \mathrm{d}\nu_t = \alpha_\nu \left( \Psi_{\sigma_\nu}(\nu_t, \mu_t) - \nu_t \right) \mathrm{d}t, \\ \mathrm{d}\mu_t = \alpha_\mu \left( \Phi_{\sigma_\mu}(\nu_t, \mu_t) - \mu_t \right) \mathrm{d}t, \quad (\nu, \mu)|_{t=0} = (\nu_0, \mu_0) \in \mathcal{P}_1(\mathbb{R}^d) \times \mathcal{P}_1(\mathbb{R}^d), \end{cases} \tag{8}$$
where $\alpha_\nu, \alpha_\mu > 0$ are the learning rates of each player and $\Psi_{\sigma_\nu}, \Phi_{\sigma_\mu}$ are the Best Response operators defined in Section B analogously to Definition 1.1. As Proposition C.4 will show, any MNE $(\nu^*_{\sigma_\nu}, \mu^*_{\sigma_\mu})$ of $F^{\sigma_\nu, \sigma_\mu}$ must satisfy the coupled fixed-point equations $\nu = \Psi_{\sigma_\nu}[\nu, \mu]$ and $\mu = \Phi_{\sigma_\mu}[\nu, \mu]$.

As showed in [23, Corollary 1], when $\alpha_\nu = \alpha_\mu = \alpha$, flow (8) converges exponentially at rate $\mathcal{O}(e^{-\alpha t})$ in KL divergence and $\mathrm{TV}^2$ distance to the unique MNE of (6), under the assumption that $F$ is convex-concave. A key observation is that the convergence rate depends solely on the learning rate $\alpha$ and is independent of the regularization parameter $\sigma = \sigma_\nu = \sigma_\mu > 0$. Our results extend those of [23] by removing the convexity-concavity assumption on $F$, the requirement of uniform bounds on the second-order flat derivatives of $F$, and allowing for different regularization parameters and learning rates for each player. A detailed discussion about the differences between the single-agent optimization and the min-max settings can be found in Remark 3.5.

## 1.2 Our contribution

Our results extend those of [8, 24] in the single-agent optimization setting, and [23, 21] in the mix-max setting. We summarize our contribution as follows:

- In the general setting of the problem (1), we extend the analysis of [8] by addressing the case where $F$ is non-convex. In contrast to [8], we show in Theorem 2.3 that when $\sigma > 0$ is sufficiently large, the Best Response map $\Psi_\sigma : \mathcal{P}_1(\mathbb{R}^d) \to \mathcal{P}_1(\mathbb{R}^d)$ becomes a contraction in the $L^1$-Wasserstein metric, which guarantees the existence of a unique global minimizer for (1) (cf. Corollary 2.4). Moreover, in Theorem 2.5 we derive a stability estimate for the flow, with respect to both the initial condition and the regularization parameter $\sigma$. As a consequence, in Theorem 2.6 we obtain exponential convergence of the Best Response flow to the unique minimizer in the $L^1$-Wasserstein distance — an improvement over [8, Theorem 9], which establishes convergence in the $L^1$-Wasserstein metric without an explicit rate. In the context of MDPs, compared to [24], we employ the Best Response flow instead of the Wasserstein gradient flow and achieve a similar convergence rate under less restrictive assumptions on $F$ and $U^\xi$. Hence our result can be interpreted as providing an alternative method of choosing an optimal policy for MDPs with softmax parametrized policies in the mean field regime. We also briefly discuss the implementation of the Best Response flow in Section 2.4, however, a precise analysis of the corresponding numerical methods still remains an open problem and is left for future work.

- We extend the results of [23] by removing the convexity-concavity assumption on $F$, the requirement of uniformly bounded second-order flat derivatives and allowing for different regularization parameters $\sigma_\nu, \sigma_\mu$ and learning rates $\alpha_\nu, \alpha_\mu$ for each player. Specifically, in Theorem B.2, we prove that, when $\sigma_\nu, \sigma_\mu$ are sufficiently large, the joint Best Response operator $\left( \Psi_{\sigma_\nu}, \Phi_{\sigma_\mu} \right) : \mathcal{P}_1(\mathbb{R}^d) \times \mathcal{P}_1(\mathbb{R}^d) \to \mathcal{P}_1(\mathbb{R}^d) \times \mathcal{P}_1(\mathbb{R}^d)$ is a contraction with respect to the $L^1$-Wasserstein distance. Consequently, we show that the game (6) admits a unique global MNE (cf. Corollary B.3). Additionally, in Theorem 3.3, we establish that under this strong regularization regime, the Best Response flow (8) converges exponentially in the $L^1$-Wasserstein metric to the equilibrium. Lastly, in Subsection E.2, we illustrate

how our theoretical results for min-max games apply to policy optimization in two-player zero-sum Markov games, thereby generalizing the examples in [21, Section 5] from the convex-concave to the non-convex-non-concave setting with softmax parametrized policies.

# 2 Assumptions and main results for single-agent optimization

In this section, we first introduce the necessary notations and assumptions used throughout the paper, followed by our main results.

## 2.1 Notation and main results

By $\mathcal{P}_1(\mathbb{R}^d)$ we denote the space of probability measures with finite first moment. We equip $\mathcal{P}_1(\mathbb{R}^d)$ with the $L^1$-Wasserstein distance $\mathcal{W}_1$. The set $\mathcal{P}_1(\mathbb{R}^d)$ endowed with the topology induced by the $\mathcal{W}_1$ distance is a complete separable metric space [40, Theorem 6.18]. Let $\mathcal{B}(\mathbb{R}^d)$ denote the Borel $\sigma$-algebra over $\mathbb{R}^d$. For any measure $\nu$ on $\left(\mathbb{R}^d, \mathcal{B}(\mathbb{R}^d)\right)$, the set of absolutely continuous measures in $\mathcal{P}_1(\mathbb{R}^d)$ with respect to $\nu$ is denoted by $\mathcal{P}_1^\nu(\mathbb{R}^d)$. In particular, we will use this notation for $\nu = \lambda$, where $\lambda$ is the $d$-dimensional Lebesgue measure. The Kantorovich–Rubinstein duality theorem [40, Theorem 5.10] implies that for all $\nu, \nu' \in \mathcal{P}_1(\mathbb{R}^d)$,

$$\mathcal{W}_1(\nu, \nu') = \sup_{\phi \in \mathrm{Lip}_1(\mathbb{R}^d)} \int_{\mathbb{R}^d} \phi(x)(\nu - \nu')(\mathrm{d}x), \tag{9}$$

where $\mathrm{Lip}_1(\mathbb{R}^d)$ denotes the space of all functions $\phi : \mathbb{R}^d \to \mathbb{R}$ with Lipschitz constant equal to 1. For $\xi \in \mathcal{P}_1(\mathbb{R}^d)$, the relative entropy $\mathrm{KL}(\cdot|\xi) : \mathcal{P}_1(\mathbb{R}^d) \to [0, \infty]$ with respect to $\xi$ is given for any $\nu \in \mathcal{P}_1(\mathbb{R}^d)$ by

$$\mathrm{KL}(\nu|\xi) := \begin{cases} \int_{\mathbb{R}^d} \log \frac{\mathrm{d}\nu}{\mathrm{d}\xi}(x)\nu(\mathrm{d}x), & \text{if } \nu \in \mathcal{P}_1^\xi(\mathbb{R}^d), \\ +\infty, & \text{otherwise.} \end{cases}$$

and analogously for $\mathrm{KL}(\cdot|\rho)$.

**Assumption 2.1** (Lipschitzness and boundedness of the flat derivative). *Assume that $F \in \mathcal{C}^1$ (cf. Definition F.1) and there exist $C_F, L_F > 0$ such that for all $\nu, \nu' \in \mathcal{P}_1(\mathbb{R}^d)$ and all $x, x' \in \mathbb{R}^d$, we have*

$$\left|\frac{\delta F}{\delta \nu}(\nu, x)\right| \leq C_F, \quad \left|\frac{\delta F}{\delta \nu}(\nu', x') - \frac{\delta F}{\delta \nu}(\nu, x)\right| \leq L_F\left(|x' - x| + \mathcal{W}_1(\nu', \nu)\right).$$

**Assumption 2.2.** *Assume that $\xi(\mathrm{d}x) := e^{-U^\xi(x)}\mathrm{d}x$[1] for a measurable function $U^\xi : \mathbb{R}^d \to \mathbb{R}$ such that $U^\xi$ is bounded below and has at least linear growth, i.e.,*

$$\operatorname*{ess\,inf}_{x \in \mathbb{R}^d} U^\xi(x) > -\infty, \quad \liminf_{x \to \infty} \frac{U^\xi(x)}{|x|} > 0.$$

We are ready to show that, under Assumptions 2.1 and 2.2, the Best Response map $\Psi_\sigma : \mathcal{P}_1(\mathbb{R}^d) \to \mathcal{P}_1(\mathbb{R}^d)$ is $\mathcal{W}_1$-Lipschitz for any $\sigma > 0$, and becomes a contraction for sufficiently large $\sigma$.

**Theorem 2.3** ($L^1$-Wasserstein contraction). *Let Assumptions 2.1, 2.2 hold. For any $\sigma > 0$, let*

$$L_{\Psi_\sigma, U^\xi} := \frac{L_F}{\sigma} \exp\left(\frac{2C_F}{\sigma}\right)\left(1 + \exp\left(\frac{2C_F}{\sigma}\right)\right)\int_{\mathbb{R}^d}|x|e^{-U^\xi(x)}\mathrm{d}x > 0.$$

*Then*

$$\mathcal{W}_1\left(\Psi_\sigma[\nu], \Psi_\sigma[\nu']\right) \leq L_{\Psi_\sigma, U^\xi}\mathcal{W}_1(\nu, \nu'),$$

*for any $\nu, \nu' \in \mathcal{P}_1(\mathbb{R}^d)$. If*

$$\sigma > 2C_F + e(e+1)L_F\int_{\mathbb{R}^d}|x|e^{-U^\xi(x)}\mathrm{d}x, \tag{10}$$

*then $L_{\Psi_\sigma, U^\xi} \in (0, 1)$, hence $\Psi_\sigma : \mathcal{P}_1(\mathbb{R}^d) \to \mathcal{P}_1(\mathbb{R}^d)$ is an $L^1$-Wasserstein contraction.*

---

[1]For simplicity, we assume that $Z_\xi := \int_{\mathbb{R}^d} e^{-U^\xi(x)}\mathrm{d}x = 1$ since we adopt the convention that the potential function $U^\xi$ is shifted by $\log Z_\xi$.

Note that (10) is a condition that relates the degree of non-convexity of $F$, the regularization parameter and the tail behaviour of the reference measure. Indeed, the convexity of $F$ can be expressed in terms of $\frac{\delta F}{\delta \nu}$ as

$$F(\nu') - F(\nu) \geq \int_{\mathbb{R}^d} \frac{\delta F}{\delta \nu}(\nu, x)(\nu' - \nu)(\mathrm{d}x),$$

for any $\nu', \nu \in \mathcal{P}_1(\mathbb{R}^d)$. Hence if $F$ can be represented as $F_1 + F_2$, where $F_1$ is convex and $F_2$ is non-convex, then the upper bound on $\left|\frac{\delta F_2}{\delta \nu}\right|$ will measure the degree of non-convexity of the problem. Moreover, the quantity $\int_{\mathbb{R}^d} |x| e^{-U^\xi(x)} \mathrm{d}x$ is the first moment of the reference measure $\xi$. Evidently from (10), making this quantity smaller helps to achieve contractivity of $\Psi_\sigma$ and this corresponds to choosing $\xi$ with lighter tails (or equivalently, $U^\xi$ with higher growth).

Note that the lower bound on $\sigma$ in condition (10) does not directly imply that the functional $F^\sigma$ in (1) is strongly convex, see Remark A.1 for details. Moreover, as shown in Remark A.1, enforcing strong convexity of $F^\sigma$ would require a lower bound on $\sigma$ that is independent of the choice of $\xi$. In contrast, a key insight of our work is that the proof of $\mathcal{W}_1$-contractivity of the Best Response operator allows us to identify a link between $\sigma$ and the reference measure $\xi$, given by (10), which ensures convergence of the Best Response flow to the global minimizer of (1). This connection enables the use of smaller values of $\sigma$ when $\xi$ is chosen appropriately.

Having proven that $\Psi_\sigma$ is an $L^1$-Wasserstein contraction in the high-regularization regime, we can conclude that it has a unique fixed point. Combining this with the fact that any minimizer of (1) has to be a fixed point of $\Psi_\sigma$, we arrive at the following corollary.

**Corollary 2.4** (Existence and uniqueness of the minimizer). *Let Assumptions 2.1, 2.2 hold. If* (10) *holds, then* (1) *has a unique global minimizer.*

The proofs of Theorem 2.3 and Corollary 2.4 can be found in Section A. Before presenting the stability estimate for the flow, both with respect to the initial condition and the regularization parameter $\sigma$, we note that Proposition C.2 ensures the existence and uniqueness of a solution $(\nu_t)_{t \geq 0} \in C\left([0, \infty]; \mathcal{P}_1(\mathbb{R}^d)\right)$ to the Best Response flow (4).

**Theorem 2.5** (Stability of the flow with respect to $\sigma$ and $\nu_0$ in $\mathcal{W}_1$). *Let Assumptions 2.1, 2.2 hold. Let* $(\nu_t)_{t \geq 0}, (\nu'_t)_{t \geq 0} \subset \mathcal{P}_1(\mathbb{R}^d)$ *be the solutions of* (4) *with parameters* $\sigma, \sigma' > 0$ *and initial conditions* $\nu_0, \nu'_0 \in \mathcal{P}_1^\lambda(\mathbb{R}^d)$, *respectively. Let* $\nu_\sigma^*, \nu_{\sigma'}^*$ *be the unique minimizers of* (1) *with parameters* $\sigma, \sigma'$, *respectively. Let*

$$L_{\Psi, \sigma, \sigma', U^\xi} := \frac{C_F}{\sigma \sigma'} \exp\left(C_F\left(\min\{\sigma, \sigma'\} + \frac{1}{\sigma'}\right)\right)\left(1 + \exp\left(\frac{2 C_F}{\sigma}\right)\right)\int_{\mathbb{R}^d} |x| e^{-U^\xi(x)} \mathrm{d}x > 0.$$

*If* (10) *holds, then for all* $t \geq 0$,

$$\mathcal{W}_1(\nu_t, \nu'_t) \leq e^{-\alpha t\left(1 - L_{\Psi_\sigma, U^\xi}\right)} \mathcal{W}_1(\nu_0, \nu'_0) + |\sigma' - \sigma| \frac{L_{\Psi, \sigma, \sigma', U^\xi}}{1 - L_{\Psi_\sigma, U^\xi}}\left(1 - e^{-\alpha t\left(1 - L_{\Psi_\sigma, U^\xi}\right)}\right),$$

$$\mathcal{W}_1(\nu_\sigma^*, \nu_{\sigma'}^*) \leq |\sigma - \sigma'| \frac{L_{\Psi, \sigma, \sigma', U^\xi}}{1 - L_{\Psi_\sigma, U^\xi}}.$$

An immediate consequence of Theorem 2.5 is our main convergence result for single-agent optimization.

**Theorem 2.6** (Convergence of (4) in the high-regularized regime). *Let Assumptions 2.1,2.2 hold. Let* $(\nu_t)_{t \geq 0} \subset \mathcal{P}_1(\mathbb{R}^d)$ *be the solution of* (4) *with parameter* $\sigma > 0$ *and initial condition* $\nu_0 \in \mathcal{P}_1^\lambda(\mathbb{R}^d)$. *If* (10) *holds, then for all* $t \geq 0$,

$$\mathcal{W}_1(\nu_t, \nu_\sigma^*) \leq e^{-\alpha t\left(1 - L_{\Psi_\sigma, U^\xi}\right)} \mathcal{W}_1(\nu_0, \nu_\sigma^*).$$

We next show how our general results can be applied to the MDP setting of [24], followed by a detailed comparison with their work, and conclude Section 2 with a numerical scheme for the Best Response flow (4) in the bandit case.

## 2.2 Markov Decision Processes

We demonstrate how our results on single-agent optimization can be applied to infinite-horizon discounted MDPs studied in [1, 24]. We refer the reader to [34, 3, 15] for a thorough introduction to MDPs.

We consider an MDP defined by the seven-tuple $\mathcal{M} = (S, A, P, c, \delta, \tau, \eta)$ with Polish spaces $S$ and $A$ representing the state and the action space, respectively, a transition kernel $P : S \times A \to \mathcal{P}(S)$, a cost function $c \in B_b(S \times A)$, a discount factor $\delta \in [0, 1)$, a regularization parameter $\tau > 0$ and a fixed reference measure $\eta \in \mathcal{M}_+(A)$. For a discussion on possible choices of $\eta$, see Remark E.4. Below we follow the kernel notation from [24] (see also Section D), i.e., we denote $P \in \mathcal{P}(S|S \times A)$, which means that $P(\cdot|s, a) \in \mathcal{P}(S)$ for any $(s, a) \in S \times A$, and similarly we consider policies $\pi \in \mathcal{P}(A|S)$, i.e, $\pi(\cdot|s) \in \mathcal{P}(A)$ for any $s \in S$.

While prior works such as [1, 24] focus on maximizing expected rewards, we adopt a cost minimization perspective to remain consistent with the general optimization framework in (1). In an MDP, at each time step, the agent observes a state $s \in S$, and based on this state, it chooses an action $a \sim \pi(\cdot|s)$ according to the policy. Then the environment returns a cost $c(s, a)$ and transitions to a new state $s' \sim P(\cdot|s, a)$ according to the transition kernel.

For a given policy $\pi \in \mathcal{P}^\eta(A|S)$ and $s \in S$, define the $\tau$-entropy regularized expected cumulative cost by

$$V_\tau^\pi(s) := \mathbb{E}_s^\pi \left[ \sum_{n=0}^\infty \delta^n \left( c(s_n, a_n) + \tau \log \frac{\mathrm{d}\pi}{\mathrm{d}\eta}(a_n|s_n) \right) \right] \in \mathbb{R},$$

where $\mathbb{E}_s^\pi$ denotes the expectation over the state-action trajectory $(s_0, a_0, s_1, a_1, ...)$ generated by policy $\pi$ and kernel $P$ such that $s_0 := s$, $a_n \sim \pi(\cdot|s_n)$ and $s_{n+1} \sim P(\cdot|s_n, a_n)$, for all $n \geq 0$. For any measure $\gamma \in \mathcal{P}(S)$ and any policy $\pi \in \mathcal{P}(A|S)$, we rigorously define $\mathbb{E}_\gamma^\pi$ in Appendix D. For a given initial distribution $\gamma \in \mathcal{P}(S)$, the agent's goal is to solve

$$\min_{\pi \in \mathcal{P}^\eta(A|S)} V_\tau^\pi(\gamma), \text{ with } V_\tau^\pi(\gamma) := \int_S V_\tau^\pi(s)\gamma(\mathrm{d}s). \tag{11}$$

Due to $\tau$-strong-convexity of the value function $\pi(\cdot|s) \mapsto V_\tau^\pi(s)$, it follows from [24, Theorem 2.1] (see also [18, Proposition 2.5]) that (11) admits a unique policy $\pi_\tau^* \in \mathcal{P}^\eta(A|S)$ independent of $\gamma$ such that

$$\pi_\tau^*(\mathrm{d}a|s) \propto \exp\left( -\frac{1}{\tau} Q_\tau^{\pi_\tau^*}(s, a) \right) \eta(\mathrm{d}a),$$

where for all $\pi \in \mathcal{P}^\eta(A|S)$, the state-value function is defined by

$$Q_\tau^\pi(s, a) = c(s, a) + \delta \int_S V_\tau^\pi(s') P(\mathrm{d}s'|s, a). \tag{12}$$

To prevent the agent from having to search for the optimal policy $\pi_\tau^*(\cdot|s)$ at each state $s \in S$ over the entire space of probability measures $\mathcal{P}^\eta(A)$, the expression of $\pi_\tau^*$ suggests that it suffices to search for the minimizer of (11) among the class of softmax policies $\{\pi_\nu(\cdot|s)|\nu \in \mathcal{P}_1(\mathbb{R}^d)\} \subset \mathcal{P}^\eta(A)$, where for each $\nu \in \mathcal{P}_1(\mathbb{R}^d)$, the kernel $\pi_\nu \in \mathcal{P}^\eta(A|S)$ is defined by

$$\pi_\nu(\mathrm{d}a|s) \propto \exp\left( f_\nu(s, a) \right) \eta(\mathrm{d}a) \tag{13}$$

and

$$f_\nu(s, a) := \int_{\mathbb{R}^d} f(\theta, s, a)\nu(\mathrm{d}\theta)$$

is a mean-field neural network with $f : \mathbb{R}^d \times S \times A \to \mathbb{R}$ denoting a bounded, differentiable, non-constant activation function and $\nu \in \mathcal{P}_1(\mathbb{R}^d)$ denoting a measure which represents the parameters of the network. Note that various activation functions such as softplus, tanh, sigmoid satisfy these assumptions.

Assuming that the policy $\pi_\nu(\cdot|s)$ is of the softmax form (13) for all states $s \in S$, the agent's objective is now to learn the optimal distribution of the network's parameters which solve the problem

$$\min_{\nu \in \mathcal{P}_1(\mathbb{R}^d)} V_\tau^{\pi_\nu}(\gamma).$$

Due to the normalization constant in (13), the value function $\nu \mapsto V_\tau^{\pi_\nu}(\gamma)$ is non-convex (see also [30]). Therefore, one should not expect convergence of gradient flows to a global minimizer unless further entropic regularization at the level of the network's parameters is added, which leads us to consider our initial optimization problem (1) with $F(\nu) := V_\tau^{\pi_\nu}(\gamma)$ and an appropriately chosen regulariser. Note that, in contrast to the general MDP problem presented above, in the bandit problem the actions and the cost are state-independent and the general setting of (11)-(13) simplifies to the setting (2)-(3) presented in Section 1.

In Proposition E.3 we will verify that the function $F(\nu) := V_\tau^{\pi_\nu}(\gamma)$ satisfies Assumption 2.1 and hence all our main results (Theorem 2.3, Corollary 2.4, Theorem 2.5) and Theorem 2.6 apply to the minimization problems (1) corresponding to energy functions $F^\sigma(\nu) := V_\tau^{\pi_\nu}(\gamma) + \sigma \operatorname{KL}(\nu|\xi)$ for any reference measure $\xi$ satisfying Assumption 2.2. In particular, we will provide explicit expressions for the constants $C_F$ and $L_F$ that appear in Assumption 2.1, in terms of the parameters of the MDP under consideration, and we will explain how to choose these parameters to ensure that the crucial condition (10) is satisfied, cf. Remark E.4.

## 2.3  Comparison to the Wasserstein gradient flow

Our work is closely related to [24], which also considers the MDP presented in Section 2.2, and aims to minimize the function $\nu \mapsto V_\tau^{\pi_\nu}(\gamma) + \sigma \operatorname{KL}(\nu|\xi)$ with $\xi \propto e^{-U^\xi}$. In [24, Theorem 2.12], it is shown that the Wasserstein gradient flow $(\nu_t)_{t \geq 0}$ defined by

$$\partial_t \nu_t = \nabla \cdot \left( \left( \nabla \frac{\delta F}{\delta \nu}(\nu_t, \cdot) + \sigma \nabla U^\xi \right) \nu_t \right) + \sigma \Delta \nu_t, \quad \nu|_{t=0} := \nu_0 \in \mathcal{P}_2(\mathbb{R}^d), \qquad (14)$$

converges exponentially at rate $\mathcal{O}(e^{-\beta t})$ to the unique global minimizer of (1) in the $L^2$-Wasserstein distance, provided that $\sigma > 0$ is sufficiently large relative to the regularity constants of $F$ and $U^\xi$.

More precisely, the convergence rate $\beta := \sigma\kappa - C_2 - L$ is determined by the strong convexity constant $\kappa > 0$ of $U^\xi$, i.e.,

$$\left( \nabla U^\xi(x') - \nabla U^\xi(x) \right) \cdot (x' - x) \geq \kappa |x' - x|^2,$$

the uniform bound $C_2 > 0$ on the Hessian of the flat derivative of $F$ and the Lipschitz constant $L > 0$ of the map $\nu \mapsto \nabla \frac{\delta F}{\delta \nu}(\nu, \cdot)$ with respect to the $L^1$-Wasserstein distance, i.e.,

$$\left| \nabla^2 \frac{\delta F}{\delta \nu}(\nu, \cdot) \right| \leq C_2, \quad \left| \nabla \frac{\delta F}{\delta \nu}(\nu', \cdot) - \nabla \frac{\delta F}{\delta \nu}(\nu, \cdot) \right| \leq L \mathcal{W}_1(\nu', \nu).$$

In principle, choosing either $\sigma$ or $\kappa$ sufficiently large relative to $C_2$ and $L$ ensures that $\beta > 0$. However, additional assumptions on $U^\xi$ required for the well-posedness of the flow (14), such as Lipschitzness and at most linear growth of $\nabla U^\xi$, impose an upper bound on $\kappa$ (see Assumptions 2.5, 2.8 in [24]). Consequently, the quadratic potential $U^\xi(x) = |x|^2$ becomes essentially the only viable choice.

In contrast, under significantly weaker regularity assumptions on $F$ and $U^\xi$, we establish exponential convergence of the Best Response flow to the unique global minimizer of (1) in the $L^1$-Wasserstein distance, with rate $\mathcal{O}\left( e^{-\alpha\left(1 - L_{\Psi_\sigma, U^\xi}\right)t} \right)$, provided that $\sigma > 0$ is sufficiently large. In particular, taking

$$\sigma > 2C_F + e(e+1)L_F \int_{\mathbb{R}^d} |x| e^{-U^\xi(x)} \mathrm{d}x,$$

where $C_F, L_F > 0$ are the constants in Assumption 2.1 ensures that

$$L_{\Psi_\sigma, U^\xi} := \frac{L_F}{\sigma} \exp\left( \frac{2C_F}{\sigma} \right) \left( 1 + \exp\left( \frac{2C_F}{\sigma} \right) \right) \int_{\mathbb{R}^d} |x| e^{-U^\xi(x)} \mathrm{d}x < 1.$$

Furthermore, in comparison to [24], we only require $U^\xi$ to be bounded below and to have at least linear growth (cf. Assumption 2.2). Therefore, selecting a potential $U^\xi$ with sufficiently rapid growth can effectively reduce the lower bound required on $\sigma$.

Regarding the possible choice of $\alpha$, we would like to remark that, while in the continuous-time setting, a larger value of $\alpha > 0$ may accelerate the convergence rate of (4), it was shown in [32, Theorem 7] that convergence of explicit Euler discretization of the flow is only guaranteed as long as the step size $\tau > 0$ satisfies $\alpha\tau \leq 1$, effectively imposing a restriction on the range of $\alpha$.

## 2.4 Numerical simulation

We now describe a numerical scheme for the Best Response flow (4) in the bandit setting ($S = \emptyset$ in Section 2.2). The idea is to update the parameter distribution $\nu$ of the mean-field neural network in (13) using a two-loop Langevin particle algorithm.

The algorithm takes as input the cost function $c$, activation function $f$, potential $U^\xi$ and reference measure $\eta$ (as in Remark E.4), together with the regularization parameter $\sigma$ prescribed by (10) (with $C_F, L_F$ from Proposition E.3), the regularization $\tau > 0$, the initial distribution of parameters $\nu_0$, outer and inner step sizes $h_{\text{out}}, h_{\text{in}}$, horizons $T, K$, learning rate $\alpha > 0$ and number of parameters $N$.

In the outer loop, we approximate $\nu_t$ by the empirical measure

$$\nu_t^N(\mathrm{d}\theta) = \frac{1}{N} \sum_{i=1}^N \delta_{\theta_t^i},$$

where $(\theta_t^i)_{i=1}^N$ are the network parameters. Since $\Psi_\sigma[\nu_t^N]$ has the exponential form (5), we compute it via an inner-loop Langevin dynamics. For $K$ steps,

$$\theta_{t,k+1}^i = \theta_{t,k}^i - h_{\text{in}} \left( \nabla_\theta \frac{\delta V_\tau^{\pi_\nu}}{\delta \nu}(\nu_t^N, \theta_{t,k}^i) + \sigma \nabla_\theta U^\xi(\theta_{t,k}^i) \right) + \sqrt{2 h_{\text{in}} \sigma} \mathcal{N}_{t,k}^i,$$

for $i = 1, ..., N$, where $\left( \mathcal{N}_{t,k}^i \right)_{i=1}^N$ are i.i.d normally distributed random variables and $h_{\text{in}} > 0$ is the step size. Note that $\frac{\delta V_\tau^{\pi_\nu}}{\delta \nu}(\nu, \cdot)$ is explicitly computed in Lemma E.2. For $K$ large enough, we set

$$\Psi_\sigma[\nu_t^N] = \frac{1}{N} \sum_{i=1}^N \delta_{\theta_{t,K}^i}.$$

The Best Response flow (4) can be computed in the outer loop via the Euler step

$$\nu_{t+1}^N = (1 - \alpha h_{\text{out}}) \nu_t^N + \alpha h_{\text{out}} \Psi_\sigma[\nu_t^N],$$

where $h_{\text{out}} > 0$ is the step size. The algorithm outputs $\nu_T^N$, which by Theorem 2.6 converges in $\mathcal{W}_1$ to $\nu_\sigma^*$, provided that $T, N$ are sufficiently large, yielding a near-optimal policy $\pi_{\nu_T^N} \approx \pi_\tau^*$. The description of the algorithm is summarized in Algorithm 1 in Appendix E.

The continuous-time analysis omits a key feature of the algorithm. In continuous time, as expressed in condition (10), the parameter $\sigma$ is determined solely by the cost function $c$, activation $f$, potential $U^\xi$, regularization $\tau$ and reference measure $\eta$ (cf. Proposition E.3). By contrast, in discrete time one must also account for the interaction of $\sigma$ with the step sizes $h_{\text{in}}$ and $h_{\text{out}}$. A rigorous analysis of this discrete-time scheme lies beyond the scope of the present paper and will be studied in future work.

## 3 Assumptions and main results for min-max problems

We now turn our attention to the class of min-max problems given by (6). We start with the necessary assumptions, and then formulate the main results. Let us first consider the metric space $(\mathcal{P}_1(\mathbb{R}^d) \times \mathcal{P}_1(\mathbb{R}^d), \widetilde{\mathcal{W}}_1)$ equipped with the $L^1$-Wasserstein distance

$$\widetilde{\mathcal{W}}_1 \left( (\nu, \mu), (\nu', \mu') \right) \coloneqq \mathcal{W}_1(\nu, \nu') + \mathcal{W}_1(\mu, \mu'),$$

for any $(\nu, \mu), (\nu', \mu') \in \mathcal{P}_1(\mathbb{R}^d) \times \mathcal{P}_1(\mathbb{R}^d)$.

**Assumption 3.1** (Lipschitzness and boundedness of the flat derivative)**.** *Assume that $F \in \mathcal{C}^1$ (cf. Definition F.1) and there exist $C_F, \bar{C}_F, L_F, \bar{L}_F > 0$ such that for all $\nu, \mu, \nu', \mu' \in \mathcal{P}_1(\mathbb{R}^d)$ and all $x, y, x', y' \in \mathbb{R}^d$, we have*

$$\left| \frac{\delta F}{\delta \nu}(\nu, \mu, x) \right| \le C_F, \quad \left| \frac{\delta F}{\delta \nu}(\nu', \mu', x') - \frac{\delta F}{\delta \nu}(\nu, \mu, x) \right| \le L_F \left( |x' - x| + \widetilde{\mathcal{W}}_1 \left( (\nu, \mu), (\nu', \mu') \right) \right),$$

$$\left| \frac{\delta F}{\delta \mu}(\nu, \mu, y) \right| \le \bar{C}_F, \quad \left| \frac{\delta F}{\delta \mu}(\nu', \mu', y') - \frac{\delta F}{\delta \mu}(\nu, \mu, y) \right| \le \bar{L}_F \left( |y' - y| + \widetilde{\mathcal{W}}_1 \left( (\nu, \mu), (\nu', \mu') \right) \right),$$

**Assumption 3.2.** *Assume that $\xi(\mathrm{d}x) := e^{-U^\xi(x)}\mathrm{d}x$ and $\rho(\mathrm{d}y) := e^{-U^\rho(y)}\mathrm{d}y$ for measurable functions $U^\xi, U^\rho : \mathbb{R}^d \to \mathbb{R}$ such that $U^\xi, U^\rho$ are bounded below and have at least linear growth, i.e.,*

$$\operatorname*{ess\,inf}_{x \in \mathbb{R}^d} U^\xi(x) > -\infty,\ \operatorname*{ess\,inf}_{y \in \mathbb{R}^d} U^\rho(y) > -\infty,\ \liminf_{x \to \infty} \frac{U^\xi(x)}{|x|} > 0,\ \liminf_{y \to \infty} \frac{U^\rho(y)}{|y|} > 0.$$

In Theorem B.2 we show that, under Assumptions 3.1 and 3.2, the pair of Best Response maps $\left(\Psi_{\sigma_\nu}, \Phi_{\sigma_\mu}\right) : \mathcal{P}_1(\mathbb{R}^d) \times \mathcal{P}_1(\mathbb{R}^d) \to \mathcal{P}_1(\mathbb{R}^d) \times \mathcal{P}_1(\mathbb{R}^d)$ is $\widetilde{\mathcal{W}}_1$-Lipschitz for any $\sigma_\nu, \sigma_\mu > 0$, and for sufficiently large $\sigma_\nu, \sigma_\mu$ (cf. conditions (29)-(30)) becomes a contraction on $\left(\mathcal{P}_1(\mathbb{R}^d) \times \mathcal{P}_1(\mathbb{R}^d), \widetilde{\mathcal{W}}_1\right)$.

Having proven that $\left(\Psi_{\sigma_\nu}, \Phi_{\sigma_\mu}\right)$ is an $L^1$-Wasserstein contraction in the high-regularization regime, we can conclude that it has a unique fixed point. Combining this with the fact that any MNE of (6) has to be a fixed point of $\left(\Psi_{\sigma_\nu}, \Phi_{\sigma_\mu}\right)$, we arrive at uniqueness of the MNE of (6) via Corollary B.3. Proposition C.5 ensures the existence and uniqueness of a solution $(\nu_t, \mu_t)_{t \geq 0} \in C\left([0, \infty]; \mathcal{P}_1(\mathbb{R}^d) \times \mathcal{P}_1(\mathbb{R}^d)\right)$ to the Best Response flow (8).

Subsequently, we prove in Theorem B.4 a stability estimate for the flow, both with respect to the initial condition and the regularization parameters $\sigma_\nu, \sigma_\mu$. An immediate consequence of Theorem B.4 is our main convergence result for min-max problems.

**Theorem 3.3** (Convergence of (8) in the high-regularized regime)**.** *Let Assumptions 3.1,3.2 hold. Let $(\nu_t, \mu_t)_{t \geq 0} \subset \mathcal{P}_1(\mathbb{R}^d) \times \mathcal{P}_1(\mathbb{R}^d)$ be the solution of (8) with parameters $\sigma_\nu, \sigma_\mu > 0$ and initial condition $(\nu_0, \mu_0) \in \mathcal{P}_1^\lambda(\mathbb{R}^d) \times \mathcal{P}_1^\lambda(\mathbb{R}^d)$. If (41) and (42) hold, then for all $t \geq 0$,*

$$\widetilde{\mathcal{W}}_1\left((\nu_t, \mu_t), (\nu_{\sigma_\nu}^*, \mu_{\sigma_\mu}^*)\right) \leq e^{-t\left(\min\{\alpha_\nu, \alpha_\mu\} - \left(\alpha_\nu L_{\Psi_{\sigma_\nu}, U^\xi} + \alpha_\mu L_{\Phi_{\sigma_\mu}, U^\rho}\right)\right)} \widetilde{\mathcal{W}}_1\left((\nu_0, \mu_0), (\nu_{\sigma_\nu}^*, \mu_{\sigma_\mu}^*)\right).$$

**Remark 3.4.** *Note that, similarly to the case of single-agent optimization, the Wasserstein contractivity of the Best Response flow in Theorem B.2 does not depend on the choice of the learning rates $\alpha_\nu$, $\alpha_\mu$, however, these rates influence the stability estimates and convergence rates of (8) in Theorems B.4 and 3.3. Unlike in the single-agent case, in the min-max setting, in the stability and convergence results we work with learning rate-dependent counterparts of the lower bounds on $\sigma_\nu, \sigma_\mu$. Note that (41)-(42) imply (29)-(30) since $\alpha_\nu, \alpha_\mu \geq \min\{\alpha_\nu, \alpha_\mu\}$. Moreover, in the case where $\alpha := \alpha_\nu = \alpha_\mu$, (29)-(30) and (41)-(42) coincide.*

**Remark 3.5.** *Evidently, our results for the min-max problems (6) are a direct counterpart to our results for the minimization problem (1). However, working with the Best Response flow in the context of min-max problems leads to additional difficulties compared to its application in single-agent optimization.*

*We note that each player is allowed to have distinct regularization parameters, $\sigma_\nu$ and $\sigma_\mu$, as well as different learning rates, $\alpha_\nu$ and $\alpha_\mu$. The presence of differing learning rates, in particular, plays a significant role in determining suitable regularizers to ensure convergence (cf. Theorems B.4, 3.3), in contrast to the single-agent setting, where the choices of $\sigma$ and $\alpha$ are independent, and the choice of $\alpha$ is unrestricted in the continuous-time setting, cf. the remark at the end of Section 2.3.*

*Problem (1) was previously examined in [8] under the assumption that $F$ is convex. It was shown that the Best Response flow (4) exhibits exponential convergence in the value function $F^\sigma$ at rate $\mathcal{O}(\sigma e^{-\alpha t})$ for any $\sigma > 0$. Subsequently, [23] extended the results of [8] to the min–max setting (6), assuming $F$ is convex–concave and choosing $\sigma := \sigma_\nu = \sigma_\mu$. It was proved that the Best Response flow (8) with $\alpha := \alpha_\nu = \alpha_\mu$ converges exponentially at rate $\mathcal{O}(e^{-\alpha t})$ in both KL divergence and $\mathrm{TV}^2$ distance, and at rate $\mathcal{O}(\sigma e^{-\alpha t})$ in the Nikaidó–Isoda (NI) error (see [23, Subsection 1.2]). However, [23] did not analyze convergence in terms of the Wasserstein distance.*

*The present work significantly generalizes the findings of [23]. We show that exponential convergence to the MNE can be achieved without assuming convexity–concavity of $F$, while also allowing for different regularization parameters $\sigma_\nu, \sigma_\mu$ and learning rates $\alpha_\nu, \alpha_\mu$ for each player. Notably, the convergence result in [23, Theorem 2] fundamentally relies on the fact that the players update their strategies at the same rate, i.e., $\alpha_\nu = \alpha_\mu$, whereas Theorem 3.3 establishes that this condition is not necessary.*

# 4 Acknowledgments and Disclosure of Funding

We thank the anonymous reviewers for their insightful comments that helped to improve the paper. R-AL gratefully acknowledges the support of the EPSRC Centre for Doctoral Training in Mathematical Modelling, Analysis and Computation (MAC-MIGS), funded by the UK Engineering and Physical Sciences Research Council (grant EP/S023291/1), Heriot-Watt University, and the University of Edinburgh. This work was carried out while R-AL was a PhD student at Heriot-Watt University.

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

# A  Proofs of the main results in Section 2

In this section we present the proofs of the results from Section 2. We start with the proof of Theorem 2.3.

*Proof of Theorem 2.3.* The proof is an adaptation of [8, Proposition 14, Corollary 15] for the $L^1$-Wasserstein distance. We give the proof in two steps.

*Step 1:* First, we show that the the Best Response map $\Psi_\sigma[\nu]$ belongs to $\mathcal{P}_1(\mathbb{R}^d)$, and that $\frac{\mathrm{d}\Psi_\sigma[\nu]}{\mathrm{d}\xi}(x)$ is uniformly bounded from below and above. From Assumption 2.1, we have

$$\exp\left(-\frac{1}{\sigma}C_F - U^\xi(x)\right) \leq \exp\left(-\frac{1}{\sigma}\frac{\delta F}{\delta \nu}(\nu, x) - U^\xi(x)\right) \leq \exp\left(\frac{1}{\sigma}C_F - U^\xi(x)\right). \quad (15)$$

Integrating over $\mathbb{R}^d$ and using the fact that $\int_{\mathbb{R}^d} e^{-U^\xi(x)}\mathrm{d}x = 1$, we obtain

$$\exp\left(-\frac{C_F}{\sigma}\right) \leq Z_\sigma(\nu) := \int_{\mathbb{R}^d} \exp\left(-\frac{1}{\sigma}\frac{\delta F}{\delta \nu}(\nu, x) - U^\xi(x)\right)\mathrm{d}x \leq \exp\left(\frac{C_F}{\sigma}\right). \quad (16)$$

Thus, we obtain

$$k_{\Psi_\sigma} e^{-U^\xi(x)} \leq \Psi_\sigma[\nu](x) \leq K_{\Psi_\sigma} e^{-U^\xi(x)}, \quad (17)$$

with constant $K_{\Psi_\sigma} = \frac{1}{k_{\Psi_\sigma}} = \exp\left(\frac{2C_F}{\sigma}\right) > 1$, where, by an abuse of notation, $\Psi_\sigma[\nu](x)$ denotes the density of $\Psi_\sigma[\nu]$ with respect to $\lambda$ on $\mathbb{R}^d$. Moreover, by definition,

$$\int_{\mathbb{R}^d} \Psi_\sigma[\nu](\mathrm{d}x) = \int_{\mathbb{R}^d} \Psi_\sigma[\nu](x)\,\mathrm{d}x = \frac{1}{Z_\sigma(\nu)}\int_{\mathbb{R}^d}\exp\left(-\frac{1}{\sigma}\frac{\delta F}{\delta \nu}(\nu, x) - U^\xi(x)\right)\mathrm{d}x = 1,$$

and using Assumption 2.2,

$$\int_{\mathbb{R}^d} |x|\Psi_\sigma[\nu](\mathrm{d}x) = \int_{\mathbb{R}^d} |x|\Psi_\sigma[\nu](x)\,\mathrm{d}x \leq K_{\Psi_\sigma}\int_{\mathbb{R}^d}|x|e^{-U^\xi(x)}\mathrm{d}x < \infty.$$

Therefore, $\Psi_\sigma[\nu] \in \mathcal{P}_1(\mathbb{R}^d)$.

*Step 2:* We show that the map $\Psi_\sigma : \mathcal{P}_1(\mathbb{R}^d) \to \mathcal{P}_1(\mathbb{R}^d)$ is $\mathcal{W}_1$-Lipschitz and then that for sufficiently large $\sigma$ it is actually a $\mathcal{W}_1$-contraction. From Assumption 2.1 and the estimate $|e^x - e^y| \leq e^{\max\{x,y\}}|x - y|$, we have

$$\left|\exp\left(-\frac{1}{\sigma}\frac{\delta F}{\delta \nu}(\nu, x) - U^\xi(x)\right) - \exp\left(-\frac{1}{\sigma}\frac{\delta F}{\delta \nu}(\nu', x) - U^\xi(x)\right)\right|$$
$$\leq \frac{L_F}{\sigma}\exp\left(\frac{1}{\sigma}C_F\right)e^{-U^\xi(x)}\mathcal{W}_1(\nu, \nu'). \quad (18)$$

Integrating the previous inequality with respect to $x$, we obtain

$$|Z_\sigma(\nu) - Z_\sigma(\nu')| \leq \frac{L_F}{\sigma}\exp\left(\frac{1}{\sigma}C_F\right)\mathcal{W}_1(\nu, \nu'). \quad (19)$$

Therefore, we have that

$$
\begin{aligned}
\left|\Psi_\sigma[\nu](x) - \Psi_\sigma[\nu'](x)\right| &= \left| \frac{1}{Z_\sigma(\nu)} \exp\left( -\frac{1}{\sigma} \frac{\delta F}{\delta \nu}(\nu, x) - U^\xi(x) \right) \right. \\
&\quad - \frac{1}{Z_\sigma(\nu')} \exp\left( -\frac{1}{\sigma} \frac{\delta F}{\delta \nu}(\nu, x) - U^\xi(x) \right) \\
&\quad + \frac{1}{Z_\sigma(\nu')} \exp\left( -\frac{1}{\sigma} \frac{\delta F}{\delta \nu}(\nu, x) - U^\xi(x) \right) \\
&\quad \left. - \frac{1}{Z_\sigma(\nu')} \exp\left( -\frac{1}{\sigma} \frac{\delta F}{\delta \nu}(\nu', x) - U^\xi(x) \right) \right| \\
&\leq \exp\left( -\frac{1}{\sigma} \frac{\delta F}{\delta \nu}(\nu, x) - U^\xi(x) \right) \frac{\left| Z_\sigma(\nu') - Z_\sigma(\nu) \right|}{Z_\sigma(\nu) Z_\sigma(\nu')} \\
&\quad + \frac{1}{Z_\sigma(\nu')} \left| \exp\left( -\frac{1}{\sigma} \frac{\delta F}{\delta \nu}(\nu, x) - U^\xi(x) \right) \right. \\
&\quad \left. - \exp\left( -\frac{1}{\sigma} \frac{\delta F}{\delta \nu}(\nu', x) - U^\xi(x) \right) \right|.
\end{aligned}
$$

Using estimates (15), (16), (18) and (19), we arrive at the Lipschitz property

$$
|\Psi_\sigma[\nu](x) - \Psi_\sigma[\nu'](x)| \leq L_{\Psi_\sigma} e^{-U^\xi(x)} \mathcal{W}_1(\nu, \nu'), \tag{20}
$$

with

$$
L_{\Psi_\sigma} := \frac{L_F}{\sigma} \exp\left( \frac{2C_F}{\sigma} \right) \left( 1 + \exp\left( \frac{2C_F}{\sigma} \right) \right) > 0.
$$

Now, applying [8, Lemma 16] with $p = 1$ and $\mu(\mathrm{d}x) = e^{-U^\xi(x)}\mathrm{d}x$ gives

$$
\mathcal{W}_1\left( \Psi_\sigma[\nu], \Psi_\sigma[\nu'] \right) \leq \int_{\mathbb{R}^d} |x| e^{-U^\xi(x)} \mathrm{d}x \left\| \frac{\Psi_\sigma[\nu](\cdot)}{e^{-U^\xi(\cdot)}} - \frac{\Psi_\sigma[\nu'](\cdot)}{e^{-U^\xi(\cdot)}} \right\|_{L^\infty(\mathbb{R}^d)}.
$$

Hence, using (20) and setting

$$
L_{\Psi_\sigma, U^\xi} := L_{\Psi_\sigma} \int_{\mathbb{R}^d} |x| e^{-U^\xi(x)} \mathrm{d}x > 0
$$

gives

$$
\mathcal{W}_1\left( \Psi_\sigma[\nu], \Psi_\sigma[\nu'] \right) \leq L_{\Psi_\sigma, U^\xi} \mathcal{W}_1(\nu, \nu').
$$

If

$$
\sigma > 2C_F + e(e+1)L_F \int_{\mathbb{R}^d} |x| e^{-U^\xi(x)} \mathrm{d}x,
$$

then immediately $L_{\Psi_\sigma, U^\xi} \in (0, 1)$, hence $\Psi_\sigma : \mathcal{P}_1(\mathbb{R}^d) \to \mathcal{P}_1(\mathbb{R}^d)$ is a $\mathcal{W}_1$-contraction. $\qquad \square$

**Remark A.1.** *The lower bound on $\sigma$ in Theorem 2.3 does not directly imply that the functional $F^\sigma$ in* (1) *is strongly convex with respect to* $\mathrm{TV}^2$, *as the following formal computation demonstrates. Note that, for any $\nu', \nu \in \mathcal{P}_1(\mathbb{R}^d)$,*

$$
\begin{aligned}
&F^\sigma(\nu') - F^\sigma(\nu) - \int_{\mathbb{R}^d} \frac{\delta F^\sigma}{\delta \nu}(\nu, x)(\nu' - \nu)(\mathrm{d}x) \\
&= F(\nu') - F(\nu) - \int \frac{\delta F}{\delta \nu}(\nu, x)(\nu' - \nu)(\mathrm{d}x) + \\
&\quad + \sigma \mathrm{KL}(\nu'|\xi) - \sigma \mathrm{KL}(\nu|\xi) - \sigma \int \frac{\delta \mathrm{KL}(\cdot|\xi)}{\delta \nu}(\nu, x)(\nu' - \nu)(\mathrm{d}x).
\end{aligned}
$$

*Remark F.2 and Assumption 2.1 give*

$$\left| F(\nu') - F(\nu) - \int_{\mathbb{R}^d} \frac{\delta F}{\delta \nu}(\nu, x)(\nu' - \nu)(\mathrm{d}x) \right|$$

$$= \left| \int_0^1 \int \left( \frac{\delta F}{\delta \nu}(\nu + \varepsilon(\nu' - \nu), x) - \frac{\delta F}{\delta \nu}(\nu, x) \right)(\nu' - \nu)(\mathrm{d}x)\mathrm{d}\varepsilon \right|$$

$$\leq L_F \int_0^1 \int \mathcal{W}_1(\nu + \varepsilon(\nu' - \nu), \nu)|\nu' - \nu|(\mathrm{d}x)\mathrm{d}\varepsilon$$

$$= 2L_F \operatorname{TV}(\nu', \nu) \int_0^1 \mathcal{W}_1(\nu + \varepsilon(\nu' - \nu), \nu)\mathrm{d}\varepsilon$$

$$\leq 2L_F \operatorname{TV}(\nu', \nu)\mathcal{W}_1(\nu', \nu) \int_0^1 \varepsilon \mathrm{d}\varepsilon$$

$$= L_F \operatorname{TV}(\nu', \nu)\mathcal{W}_1(\nu', \nu),$$

*where the second equality is due to [39, Lemma 2.1] and the second inequality uses the convexity of the Wasserstein distance.*

*On the other hand, since*

$$\frac{\delta \operatorname{KL}(\cdot|\xi)}{\delta \nu}(\nu, x) = \log \frac{\mathrm{d}\nu}{\mathrm{d}\xi}(x) - \operatorname{KL}(\nu|\xi),$$

*we have*

$$\operatorname{KL}(\nu'|\xi) - \operatorname{KL}(\nu|\xi) - \int_{\mathbb{R}^d} \frac{\delta \operatorname{KL}(\cdot|\xi)}{\delta \nu}(\nu, x)(\nu' - \nu)(\mathrm{d}x) = \operatorname{KL}(\nu'|\nu). \tag{21}$$

*Moreover, by Pinsker's inequality, $\operatorname{TV}^2(\nu', \nu) \leq \frac{1}{2}\operatorname{KL}(\nu'|\nu)$, so we obtain*

$$\operatorname{KL}(\nu'|\xi) - \operatorname{KL}(\nu|\xi) - \int_{\mathbb{R}^d} \frac{\delta \operatorname{KL}(\cdot|\xi)}{\delta \nu}(\nu, x)(\nu' - \nu)(\mathrm{d}x) \geq 2\operatorname{TV}^2(\nu', \nu). \tag{22}$$

*Therefore,*

$$F^\sigma(\nu') - F^\sigma(\nu) - \int \frac{\delta F^\sigma}{\delta \nu}(\nu, x)(\nu' - \nu)(dx) \geq -L_F \operatorname{TV}(\nu', \nu)\mathcal{W}_1(\nu', \nu) + 2\sigma \operatorname{TV}^2(\nu', \nu).$$

*This shows that while the $\operatorname{KL}$ divergence has natural strong convexity in terms of $\operatorname{TV}^2$ (cf. equation (22)), this does not align with the $\mathcal{W}_1$-based Lipschitz continuity of $F$. Therefore, it does not seem possible to directly deduce strong convexity of $F^\sigma$ in the $\operatorname{TV}^2$ sense simply by choosing $\sigma$ large enough relative to $L_F$. This is one of the motivations for proving convergence via contraction of the Best Response operator rather than by attempting to enforce strong convexity of the objective function.*

*As a further remark, it would be possible to modify Assumption 2.1 to assume Lipschitz continuity with respect to $\operatorname{TV}$ rather than $\mathcal{W}_1$. Then we would obtain*

$$F^\sigma(\nu') - F^\sigma(\nu) - \int \frac{\delta F^\sigma}{\delta \nu}(\nu, x)(\nu' - \nu)(\mathrm{d}x) \geq -L_F \operatorname{TV}^2(\nu', \nu) + 2\sigma \operatorname{TV}^2(\nu', \nu),$$

*so choosing $\sigma > \frac{1}{2}L_F$ would ensure strong convexity of $F^\sigma$ relative to $\operatorname{TV}^2$. Even though existence and uniqueness of a minimizer could now be established, it is not obvious how to prove that $\operatorname{TV}^2(\nu_t, \nu_\sigma^*) \to 0$ for the Best Response flow $(\nu_t)_{t \geq 0}$. Moreover, this would change the setting of the problem, which would no longer be comparable to [8, 24], which work in the Wasserstein space.*

*Finally, we emphasize that this argument eliminates any dependence on the choice of the reference measure $\xi$ (cf. equation (21)) and hence any lower bound on $\sigma$ obtained this way, which would guarantee strong convexity of $F^\sigma$ with respect to $\operatorname{TV}^2$, would be independent of the choice of $\xi$. On the other hand, one of the important conclusions from our paper is that by following the proof via $\mathcal{W}_1$-contractivity of the Best Response operator, we can identify the connection between the choice of $\sigma$ and the reference measure $\xi$ expressed in terms of (10), that guarantees convergence of the BR flow to the global minimizer. This allows one to choose a smaller value of $\sigma$ if the measure $\xi$ is chosen appropriately.*

*Proof of Corollary 2.4.* Since $\sigma > 2C_F + e(e+1)L_F \int_{\mathbb{R}^d} |x|e^{-U^\xi(x)}\mathrm{d}x$, we can apply Banach's fixed point theorem for the contraction $\Psi_\sigma$ on the complete metric space $\left(\mathcal{P}_1(\mathbb{R}^d), \mathcal{W}_1\right)$ and deduce that the fixed point problem $\Psi_\sigma[\nu] = \nu$ admits a unique solution. We also know from Proposition C.1 that any local minimizer $\nu_\sigma^* \in \mathcal{P}_1(\mathbb{R}^d)$ of $F^\sigma$ is equivalent to $\lambda$, and for $\lambda$-a.a. $x \in \mathbb{R}^d$,

$$\nu_\sigma^*(\mathrm{d}x) = \frac{e^{-\frac{1}{\sigma}\frac{\delta F}{\delta \nu}(\nu_\sigma^*, x) - U^\xi(x)}}{\int_{\mathbb{R}^d} e^{-\frac{1}{\sigma}\frac{\delta F}{\delta \nu}(\nu_\sigma^*, x') - U(x')}\mathrm{d}x'}\mathrm{d}x.$$

Therefore, $\Psi_\sigma[\nu_\sigma^*] = \nu_\sigma^*$ due to the definition of $\Psi_\sigma$. Since $\Psi_\sigma[\nu] = \nu$ admits a unique solution, it follows that $F^\sigma$ has a unique minimizer, that is $\nu_\sigma^*$, and hence it is actually the global minimizer. $\quad\square$

*Proof of Theorem 2.5.* *Step 1:* First, we show that the map $\Psi_\sigma$ is $\mathcal{W}_1$-Lipschitz with respect to $\sigma$. From Assumption 2.1 and the estimate $|e^x - e^y| \le e^{\max\{x,y\}}|x - y|$, we have

$$\left|\exp\left(-\frac{1}{\sigma}\frac{\delta F}{\delta \nu}(\nu, x) - U^\xi(x)\right) - \exp\left(-\frac{1}{\sigma'}\frac{\delta F}{\delta \nu}(\nu, x) - U^\xi(x)\right)\right|$$
$$\le C_F \frac{|\sigma - \sigma'|}{\sigma\sigma'}\exp\left(C_F \min\{\sigma, \sigma'\}\right)e^{-U^\xi(x)}. \quad (23)$$

Integrating the previous inequality with respect to $x$, we obtain

$$|Z_\sigma(\nu) - Z_{\sigma'}(\nu)| \le C_F \frac{|\sigma - \sigma'|}{\sigma\sigma'}\exp\left(C_F \min\{\sigma, \sigma'\}\right). \quad (24)$$

Therefore, we have that

$$\left|\Psi_\sigma[\nu](x) - \Psi_{\sigma'}[\nu](x)\right| = \left|\frac{1}{Z_\sigma(\nu)}\exp\left(-\frac{1}{\sigma}\frac{\delta F}{\delta \nu}(\nu, x) - U^\xi(x)\right)\right.$$
$$- \frac{1}{Z_{\sigma'}(\nu)}\exp\left(-\frac{1}{\sigma}\frac{\delta F}{\delta \nu}(\nu, x) - U^\xi(x)\right)$$
$$+ \frac{1}{Z_{\sigma'}(\nu)}\exp\left(-\frac{1}{\sigma}\frac{\delta F}{\delta \nu}(\nu, x) - U^\xi(x)\right)$$
$$\left. - \frac{1}{Z_{\sigma'}(\nu)}\exp\left(-\frac{1}{\sigma'}\frac{\delta F}{\delta \nu}(\nu, x) - U^\xi(x)\right)\right|$$
$$\le \exp\left(-\frac{1}{\sigma}\frac{\delta F}{\delta \nu}(\nu, x) - U^\xi(x)\right)\frac{\left|Z_{\sigma'}(\nu) - Z_\sigma(\nu)\right|}{Z_\sigma(\nu)Z_{\sigma'}(\nu)}$$
$$+ \frac{1}{Z_{\sigma'}(\nu)}\left|\exp\left(-\frac{1}{\sigma}\frac{\delta F}{\delta \nu}(\nu, x) - U^\xi(x)\right)\right.$$
$$\left. - \exp\left(-\frac{1}{\sigma'}\frac{\delta F}{\delta \nu}(\nu, x) - U^\xi(x)\right)\right|.$$

Using estimates (15), (16), (23) and (24), we arrive at

$$|\Psi_\sigma[\nu](x) - \Psi_{\sigma'}[\nu](x)| \le L_{\Psi, \sigma, \sigma'}|\sigma - \sigma'|e^{-U^\xi(x)}, \quad (25)$$

with

$$L_{\Psi, \sigma, \sigma'} := \frac{C_F}{\sigma\sigma'}\exp\left(C_F\left(\min\{\sigma, \sigma'\} + \frac{1}{\sigma'}\right)\right)\left(1 + \exp\left(\frac{2C_F}{\sigma}\right)\right) > 0.$$

Now, applying [8, Lemma 16] with $p = 1$ and $\mu(\mathrm{d}x) = e^{-U^\xi(x)}\mathrm{d}x$ gives

$$\mathcal{W}_1\left(\Psi_\sigma[\nu], \Psi_{\sigma'}[\nu]\right) \le \int_{\mathbb{R}^d} |x|e^{-U^\xi(x)}\mathrm{d}x\left\|\frac{\Psi_\sigma[\nu](\cdot)}{e^{-U^\xi(\cdot)}} - \frac{\Psi_{\sigma'}[\nu](\cdot)}{e^{-U^\xi(\cdot)}}\right\|_{L^\infty(\mathbb{R}^d)}.$$

Hence, using (25) and setting

$$L_{\Psi, \sigma, \sigma', U^\xi} := L_{\Psi, \sigma, \sigma'}\int_{\mathbb{R}^d} |x|e^{-U^\xi(x)}\mathrm{d}x > 0$$

gives
$$\mathcal{W}_1\left(\Psi_\sigma[\nu],\Psi_{\sigma'}[\nu]\right) \le L_{\Psi,\sigma,\sigma',U^\xi}|\sigma-\sigma'|.$$

*Step 2:* Now, we prove the stability of the solution to (4) with respect to $\sigma$ and $\nu_0$. Since $\nu_0,\nu_0' \in \mathcal{P}_1^\lambda(\mathbb{R}^d)$, it follows by Theorem C.2 that $(\nu_t)_{t\ge0}, (\nu_t')_{t\ge0} \subset \mathcal{P}_1^\lambda(\mathbb{R}^d)$. Let $\psi \in \mathrm{Lip}_1(\mathbb{R}^d)$. Then we have that

$$\int_{\mathbb{R}^d} \psi(x)\left(\nu_t(x)-\nu_t'(x)\right)\mathrm{d}x$$

$$= \int_{\mathbb{R}^d} \psi(x)\left(e^{-\alpha t}\nu_0(x) + \int_0^t \alpha e^{-\alpha(t-s)}\Psi_\sigma\left[\nu_s\right](x)\mathrm{d}s\right.$$

$$\left. - e^{-\alpha t}\nu_0'(x) - \int_0^t \alpha e^{-\alpha(t-s)}\Psi_{\sigma'}[\nu_s'](x)\mathrm{d}s\right)\mathrm{d}x$$

$$= \int_{\mathbb{R}^d} \psi(x)\left(e^{-\alpha t}\left(\nu_0(x)-\nu_0'(x)\right) + \int_0^t \alpha e^{-\alpha(t-s)}\left(\Psi_\sigma\left[\nu_s\right](x) - \Psi_{\sigma'}\left[\nu_s'\right](x)\right)\mathrm{d}s\right)\mathrm{d}x$$

$$= e^{-\alpha t}\int_{\mathbb{R}^d}\psi(x)\left(\nu_0(x)-\nu_0'(x)\right)\mathrm{d}x + \int_{\mathbb{R}^d}\psi(x)\int_0^t \alpha e^{-\alpha(t-s)}\left(\Psi_\sigma\left[\nu_s\right](x) - \Psi_{\sigma'}\left[\nu_s'\right](x)\right)\mathrm{d}s\mathrm{d}x$$

$$= e^{-\alpha t}\int_{\mathbb{R}^d}\psi(x)\left(\nu_0(x)-\nu_0'(x)\right)\mathrm{d}x + \int_0^t \alpha e^{-\alpha(t-s)}\int_{\mathbb{R}^d}\psi(x)\left(\Psi_\sigma\left[\nu_s\right](x) - \Psi_{\sigma'}\left[\nu_s'\right](x)\right)\mathrm{d}x\mathrm{d}s$$

$$\le e^{-\alpha t}\mathcal{W}_1(\nu_0,\nu_0') + \int_0^t \alpha e^{-\alpha(t-s)}\mathcal{W}_1\left(\Psi_\sigma\left[\nu_s\right],\Psi_{\sigma'}\left[\nu_s'\right]\right)\mathrm{d}s.$$

where in the last equality we used Fubini's theorem, in the first inequality we used the definition of $\mathcal{W}_1$. Now, using Theorem 2.3, *Step 1* and the triangle inequality, we have

$$\mathcal{W}_1\left(\Psi_\sigma\left[\nu_t\right],\Psi_{\sigma'}\left[\nu_t'\right]\right) \le \mathcal{W}_1\left(\Psi_\sigma\left[\nu_t\right],\Psi_\sigma\left[\nu_t'\right]\right) + \mathcal{W}_1\left(\Psi_\sigma\left[\nu_t'\right],\Psi_{\sigma'}\left[\nu_t'\right]\right)$$

$$\le L_{\Psi_\sigma,U^\xi}\mathcal{W}_1(\nu_t,\nu_t') + L_{\Psi,\sigma,\sigma',U^\xi}|\sigma-\sigma'|.$$

Taking supremum over $\psi$ gives

$$\mathcal{W}_1(\nu_t,\nu_t') \le e^{-\alpha t}\mathcal{W}_1(\nu_0,\nu_0') + \int_0^t \alpha e^{-\alpha(t-s)}\left[L_{\Psi_\sigma,U^\xi}\mathcal{W}_1(\nu_s,\nu_s') + L_{\Psi,\sigma,\sigma',U^\xi}|\sigma-\sigma'|\right]\mathrm{d}s.$$

Since $\sigma > 2C_F + e(e+1)L_F\int_{\mathbb{R}^d}|x|e^{-U^\xi(x)}\mathrm{d}x$, it follows that $L_{\Psi_\sigma,U^\xi} < 1$.

Therefore, applying Gronwall's lemma to the function $t \mapsto e^{\alpha t}\mathcal{W}_1(\nu_t,\nu_t')$ yields

$$\mathcal{W}_1(\nu_t,\nu_t') \le e^{-\alpha t\left(1-L_{\Psi_\sigma,U^\xi}\right)}\mathcal{W}_1(\nu_0,\nu_0') + |\sigma'-\sigma|\frac{L_{\Psi,\sigma,\sigma',U^\xi}}{1-L_{\Psi_\sigma,U^\xi}}\left(1-e^{-\alpha t\left(1-L_{\Psi_\sigma,U^\xi}\right)}\right). \quad (26)$$

Due to the fact that $\nu_\sigma^* = \Psi_\sigma[\nu_\sigma^*]$, for any $\sigma > 0$, it follows that $\nu_\sigma^*,\nu_{\sigma'}^*$ are solutions to (4) with parameters $\sigma,\sigma'$, respectively. Then using (26) yields

$$\mathcal{W}_1(\nu_\sigma^*,\nu_{\sigma'}^*) \le e^{-\alpha\left(1-L_{\Psi_\sigma,U^\xi}\right)t}\mathcal{W}_1(\nu_\sigma^*,\nu_{\sigma'}^*) + |\sigma'-\sigma|\frac{L_{\Psi,\sigma,\sigma',U^\xi}}{1-L_{\Psi_\sigma,U^\xi}}\left(1-e^{-\alpha\left(1-L_{\Psi_\sigma,U^\xi}\right)t}\right).$$

Therefore, since $L_{\Psi_\sigma,U^\xi} \in (0,1)$, we obtain

$$\mathcal{W}_1(\nu_\sigma^*,\nu_{\sigma'}^*) \le |\sigma-\sigma'|\frac{L_{\Psi,\sigma,\sigma',U^\xi}}{1-L_{\Psi_\sigma,U^\xi}}.$$

$\square$

# B  Proofs of the main results in Section 3

In this section we present the proofs of the results from Section 3. Before starting with the proof of Theorem B.2, we define the Best Response operators for game (6).

**Definition B.1** (Best Response operators [23]). *For any $\nu, \mu \in \mathcal{P}_1(\mathbb{R}^d)$ and any $\sigma_\nu, \sigma_\mu > 0$, define the Best Response operators $\Psi_{\sigma_\nu} : \mathcal{P}_1(\mathbb{R}^d) \to \mathcal{P}_1^\xi(\mathbb{R}^d)$ and $\Phi_{\sigma_\mu} : \mathcal{P}_1(\mathbb{R}^d) \to \mathcal{P}_1^\rho(\mathbb{R}^d)$ by*

$$\Psi_{\sigma_\nu}[\nu, \mu](\mathrm{d}x) := \frac{\exp\left(-\frac{1}{\sigma_\nu} \frac{\delta F}{\delta \nu}(\nu, \mu, x)\right)}{\int_{\mathbb{R}^d} \exp\left(-\frac{1}{\sigma_\nu} \frac{\delta F}{\delta \nu}(\nu, \mu, x')\right) \xi(\mathrm{d}x')} \xi(\mathrm{d}x), \tag{27}$$

$$\Phi_{\sigma_\mu}[\nu, \mu](\mathrm{d}y) := \frac{\exp\left(\frac{1}{\sigma_\mu} \frac{\delta F}{\delta \mu}(\nu, \mu, y)\right)}{\int_{\mathbb{R}^d} \exp\left(\frac{1}{\sigma_\mu} \frac{\delta F}{\delta \mu}(\nu, \mu, y')\right) \rho(\mathrm{d}y')} \rho(\mathrm{d}y). \tag{28}$$

**Theorem B.2** ($L^1$-Wasserstein contraction). *Let Assumptions 3.1, 3.2 hold. For any $\sigma_\nu, \sigma_\mu > 0$, let*

$$L_{\Psi_{\sigma_\nu}, U^\xi} := \frac{L_F}{\sigma_\nu} \exp\left(\frac{2C_F}{\sigma_\nu}\right) \left(1 + \exp\left(\frac{2C_F}{\sigma_\nu}\right)\right) \int_{\mathbb{R}^d} |x| e^{-U^\xi(x)} \mathrm{d}x > 0,$$

$$L_{\Phi_{\sigma_\mu}, U^\rho} := \frac{\bar{L}_F}{\sigma_\mu} \exp\left(\frac{2\bar{C}_F}{\sigma_\mu}\right) \left(1 + \exp\left(\frac{2\bar{C}_F}{\sigma_\mu}\right)\right) \int_{\mathbb{R}^d} |x| e^{-U^\rho(x)} \mathrm{d}x > 0.$$

*Then*

$$\widetilde{\mathcal{W}}_1 \left(\left(\Psi_{\sigma_\nu}, \Phi_{\sigma_\mu}\right)[\nu, \mu], \left(\Psi_{\sigma_\nu}, \Phi_{\sigma_\mu}\right)[\nu', \mu']\right) \le \left(L_{\Psi_{\sigma_\nu}, U^\xi} + L_{\Phi_{\sigma_\mu}, U^\rho}\right) \widetilde{\mathcal{W}}_1 \left((\nu, \mu), (\nu', \mu')\right),$$

*for any $(\nu, \mu), (\nu', \mu') \in \mathcal{P}_1(\mathbb{R}^d) \times \mathcal{P}_1(\mathbb{R}^d)$. If*

$$\sigma_\nu > 2C_F + 2e(e+1)L_F \int_{\mathbb{R}^d} |x| e^{-U^\xi(x)} \mathrm{d}x, \tag{29}$$

*and*

$$\sigma_\mu > 2\bar{C}_F + 2e(e+1)\bar{L}_F \int_{\mathbb{R}^d} |y| e^{-U^\rho(y)} \mathrm{d}y, \tag{30}$$

*then $L_{\Psi_\sigma, U^\xi} + L_{\Phi_\sigma, U^\rho} \in (0, 1)$, hence $\left(\Psi_{\sigma_\nu}, \Phi_{\sigma_\mu}\right) : \mathcal{P}_1(\mathbb{R}^d) \times \mathcal{P}_1(\mathbb{R}^d) \to \mathcal{P}_1(\mathbb{R}^d) \times \mathcal{P}_1(\mathbb{R}^d)$ is a $L^1$-Wasserstein contraction on $\left(\mathcal{P}_1(\mathbb{R}^d) \times \mathcal{P}_1(\mathbb{R}^d), \widetilde{\mathcal{W}}_1\right)$.*

*Proof.* The proof is an adaptation of the proof of Theorem 2.3. As before, we give the proof in two steps.

*Step 1:* First, we show that the the Best Response maps $\Psi_{\sigma_\nu}[\nu, \mu], \Psi_{\sigma_\mu}[\nu, \mu]$ belong to $\mathcal{P}_1(\mathbb{R}^d)$ and that $\frac{\mathrm{d}\Psi_{\sigma_\nu}[\nu, \mu]}{\mathrm{d}\xi}(x), \frac{\mathrm{d}\Phi_{\sigma_\mu}[\nu, \mu]}{\mathrm{d}\rho}(y)$ are uniformly bounded from below and above. From Assumption 3.1, we have

$$\exp\left(-\frac{1}{\sigma_\nu} C_F - U^\xi(x)\right) \le \exp\left(-\frac{1}{\sigma_\nu} \frac{\delta F}{\delta \nu}(\nu, \mu, x) - U^\xi(x)\right) \le \exp\left(\frac{1}{\sigma_\nu} C_F - U^\xi(x)\right). \tag{31}$$

Integrating over $\mathbb{R}^d$ and using the fact that $\int_{\mathbb{R}^d} e^{-U^\xi(x)} \mathrm{d}x = 1$, we obtain

$$\exp\left(-\frac{C_F}{\sigma_\nu}\right) \le Z_{\sigma_\nu}(\nu, \mu) := \int_{\mathbb{R}^d} \exp\left(-\frac{1}{\sigma_\nu} \frac{\delta F}{\delta \nu}(\nu, \mu, x) - U^\xi(x)\right) \mathrm{d}x \le \exp\left(\frac{C_F}{\sigma_\nu}\right). \tag{32}$$

Thus, we obtain

$$k_{\Psi_{\sigma_\nu}} e^{-U^\xi(x)} \le \Psi_{\sigma_\nu}[\nu, \mu](x) \le K_{\Psi_{\sigma_\nu}} e^{-U^\xi(x)}, \tag{33}$$

with constant $K_{\Psi_{\sigma_\nu}} = \frac{1}{k_{\Psi_{\sigma_\nu}}} = \exp\left(\frac{2C_F}{\sigma_\nu}\right) > 1$, where, by an abuse of notation, $\Psi_{\sigma_\nu}[\nu, \mu](x)$ denotes the density of $\Psi_{\sigma_\nu}[\nu, \mu]$ with respect to $\lambda$ on $\mathbb{R}^d$. Moreover, by definition,

$$\int_{\mathbb{R}^d} \Psi_{\sigma_\nu}[\nu, \mu](\mathrm{d}x) = \int_{\mathbb{R}^d} \Psi_{\sigma_\nu}[\nu, \mu](x) \mathrm{d}x$$

$$= \frac{1}{Z_{\sigma_\nu}(\nu, \mu)} \int_{\mathbb{R}^d} \exp\left(-\frac{1}{\sigma_\nu} \frac{\delta F}{\delta \nu}(\nu, \mu, x) - U^\xi(x)\right) \mathrm{d}x = 1,$$

and using Assumption 3.2,

$$\int_{\mathbb{R}^d} |x| \Psi_{\sigma_\nu}[\nu, \mu] (\mathrm{d}x) = \int_{\mathbb{R}^d} |x| \Psi_{\sigma_\nu}[\nu, \mu] (x) \, \mathrm{d}x \leq K_{\Psi_{\sigma_\nu}} \int_{\mathbb{R}^d} |x| e^{-U^\xi(x)} \mathrm{d}x < \infty.$$

Therefore, $\Psi_{\sigma_\nu}[\nu, \mu] \in \mathcal{P}_1\left(\mathbb{R}^d\right)$. One can argue similarly for $\Phi_{\sigma_\mu}[\nu, \mu]$ and obtain

$$k_{\Phi_{\sigma_\mu}} e^{-U^\rho(y)} \leq \Phi_\sigma[\nu, \mu](y) \leq K_{\Phi_{\sigma_\mu}} e^{-U^\rho(y)}, \tag{34}$$

with constant $K_{\Phi_{\sigma_\mu}} = \frac{1}{k_{\Phi_{\sigma_\mu}}} = \exp\left(\frac{2\bar{C}_F}{\sigma_\mu}\right) > 1$.

*Step 2:* We show that the map $\left(\Psi_{\sigma_\nu}, \Phi_{\sigma_\mu}\right) : \mathcal{P}_1\left(\mathbb{R}^d\right) \times \mathcal{P}_1\left(\mathbb{R}^d\right) \to \mathcal{P}_1\left(\mathbb{R}^d\right) \times \mathcal{P}_1\left(\mathbb{R}^d\right)$ is $\widetilde{\mathcal{W}}_1$-Lipschitz and then that for sufficiently large $\sigma_\nu, \sigma_\mu$ it is actually a $\widetilde{\mathcal{W}}_1$-contraction. From Assumption 3.1 and the estimate $|e^x - e^y| \leq e^{\max\{x,y\}} |x - y|$, we have

$$\left| \exp\left(-\frac{1}{\sigma_\nu} \frac{\delta F}{\delta \nu}(\nu, \mu, x) - U^\xi(x)\right) - \exp\left(-\frac{1}{\sigma_\nu} \frac{\delta F}{\delta \nu}(\nu', \mu', x) - U^\xi(x)\right) \right|$$
$$\leq \frac{L_F}{\sigma_\nu} \exp\left(\frac{1}{\sigma_\nu} C_F\right) e^{-U^\xi(x)} \left(\mathcal{W}_1(\nu, \nu') + \mathcal{W}_1(\mu, \mu')\right). \tag{35}$$

Integrating the previous inequality with respect to $x$, we obtain

$$|Z_{\sigma_\nu}(\nu, \mu) - Z_{\sigma_\nu}(\nu', \mu')| \leq \frac{L_F}{\sigma_\nu} \exp\left(\frac{1}{\sigma_\nu} C_F\right) \left(\mathcal{W}_1(\nu, \nu') + \mathcal{W}_1(\mu, \mu')\right). \tag{36}$$

Therefore, we have that

$$\left| \Psi_{\sigma_\nu}[\nu, \mu](x) - \Psi_{\sigma_\nu}[\nu', \mu'](x) \right| = \left| \frac{1}{Z_{\sigma_\nu}(\nu, \mu)} \exp\left(-\frac{1}{\sigma_\nu} \frac{\delta F}{\delta \nu}(\nu, \mu, x) - U^\xi(x)\right) \right.$$
$$- \frac{1}{Z_{\sigma_\nu}(\nu', \mu')} \exp\left(-\frac{1}{\sigma_\nu} \frac{\delta F}{\delta \nu}(\nu, \mu, x) - U^\xi(x)\right)$$
$$+ \frac{1}{Z_{\sigma_\nu}(\nu', \mu')} \exp\left(-\frac{1}{\sigma_\nu} \frac{\delta F}{\delta \nu}(\nu, \mu, x) - U^\xi(x)\right)$$
$$\left. - \frac{1}{Z_{\sigma_\nu}(\nu', \mu')} \exp\left(-\frac{1}{\sigma_\nu} \frac{\delta F}{\delta \nu}(\nu', \mu', x) - U^\xi(x)\right) \right|$$
$$\leq \exp\left(-\frac{1}{\sigma_\nu} \frac{\delta F}{\delta \nu}(\nu, \mu, x) - U^\xi(x)\right) \frac{\left| Z_{\sigma_\nu}(\nu', \mu') - Z_{\sigma_\nu}(\nu, \mu) \right|}{Z_{\sigma_\nu}(\nu, \mu) Z_{\sigma_\nu}(\nu', \mu')}$$
$$+ \frac{1}{Z_{\sigma_\nu}(\nu', \mu')} \left| \exp\left(-\frac{1}{\sigma_\nu} \frac{\delta F}{\delta \nu}(\nu, \mu, x) - U^\xi(x)\right) \right.$$
$$\left. - \exp\left(-\frac{1}{\sigma_\nu} \frac{\delta F}{\delta \nu}(\nu', \mu', x) - U^\xi(x)\right) \right|.$$

Using estimates (31), (32), (35) and (36), we arrive at the Lipschitz property

$$|\Psi_{\sigma_\nu}[\nu, \mu](x) - \Psi_{\sigma_\nu}[\nu', \mu'](x)| \leq L_{\Psi_{\sigma_\nu}} e^{-U^\xi(x)} \left(\mathcal{W}_1(\nu, \nu') + \mathcal{W}_1(\mu, \mu')\right), \tag{37}$$

with

$$L_{\Psi_{\sigma_\nu}} := \frac{L_F}{\sigma_\nu} \exp\left(\frac{2C_F}{\sigma_\nu}\right)\left(1 + \exp\left(\frac{2C_F}{\sigma_\nu}\right)\right) > 0.$$

Proving that

$$|\Phi_{\sigma_\nu}[\nu, \mu](y) - \Phi_{\sigma_\nu}[\nu', \mu'](y)| \leq L_{\Phi_{\sigma_\nu}} e^{-U^\rho(y)} \left(\mathcal{W}_1(\nu, \nu') + \mathcal{W}_1(\mu, \mu')\right), \tag{38}$$

follows the same steps as above but with

$$L_{\Phi_{\sigma_\mu}} := \frac{\bar{L}_F}{\sigma_\mu} \exp\left(\frac{2\bar{C}_F}{\sigma_\mu}\right)\left(1 + \exp\left(\frac{2\bar{C}_F}{\sigma_\mu}\right)\right).$$

Now, applying [8, Lemma 16] with $p = 1$ and $\mu(\mathrm{d}x) = e^{-U^\xi(x)}\mathrm{d}x$ gives

$$\mathcal{W}_1\left(\Psi_{\sigma_\nu}[\nu, \mu], \Psi_{\sigma_\nu}[\nu', \mu']\right) \leq \int_{\mathbb{R}^d} |x|e^{-U^\xi(x)}\mathrm{d}x \left\| \frac{\Psi_{\sigma_\nu}[\nu, \mu](\cdot)}{e^{-U^\xi(\cdot)}} - \frac{\Psi_{\sigma_\nu}[\nu', \mu'](\cdot)}{e^{-U^\xi(\cdot)}} \right\|_{L^\infty(\mathbb{R}^d)}.$$

Hence, using (37) and setting

$$L_{\Psi_{\sigma_\nu}, U^\xi} := L_{\Psi_{\sigma_\nu}} \int_{\mathbb{R}^d} |x|e^{-U^\xi(x)}\mathrm{d}x > 0$$

gives

$$\mathcal{W}_1\left(\Psi_{\sigma_\nu}[\nu, \mu], \Psi_{\sigma_\nu}[\nu', \mu']\right) \leq L_{\Psi_{\sigma_\nu}, U^\xi}\left(\mathcal{W}_1(\nu, \nu') + \mathcal{W}_1(\mu, \mu')\right). \tag{39}$$

One similarly obtains that $\Phi_{\sigma_\mu}$ is $\mathcal{W}_1$-Lipschitz, i.e.,

$$\mathcal{W}_1\left(\Phi_{\sigma_\mu}[\nu, \mu], \Phi_{\sigma_\mu}[\nu', \mu']\right) \leq L_{\Phi_{\sigma_\mu}, U^\rho}\left(\mathcal{W}_1(\nu, \nu') + \mathcal{W}_1(\mu, \mu')\right), \tag{40}$$

with constant

$$L_{\Phi_{\sigma_\mu}, U^\rho} := L_{\Phi_{\sigma_\mu}} \int_{\mathbb{R}^d} |y|e^{-U^\rho(y)}\mathrm{d}y > 0.$$

Therefore, we obtain

$$\mathcal{W}_1\left(\Psi_{\sigma_\nu}[\nu, \mu], \Psi_{\sigma_\nu}[\nu', \mu']\right) + \mathcal{W}_1\left(\Phi_{\sigma_\mu}[\nu, \mu], \Phi_{\sigma_\mu}[\nu', \mu']\right)$$
$$\leq \left(L_{\Psi_{\sigma_\nu}, U^\xi} + L_{\Phi_{\sigma_\mu}, U^\rho}\right)\left(\mathcal{W}_1(\nu, \nu') + \mathcal{W}_1(\mu, \mu')\right),$$

and using the definition of $\widetilde{\mathcal{W}}_1$ and the notation $\left(\Psi_{\sigma_\nu}, \Phi_{\sigma_\mu}\right)[\nu, \mu] := \left(\Psi_{\sigma_\nu}[\nu, \mu], \Phi_{\sigma_\mu}[\nu, \mu]\right)$ gives

$$\widetilde{\mathcal{W}}_1\left(\left(\Psi_{\sigma_\nu}, \Phi_{\sigma_\mu}\right)[\nu, \mu], \left(\Psi_{\sigma_\nu}, \Phi_{\sigma_\mu}\right)[\nu', \mu']\right) \leq \left(L_{\Psi_{\sigma_\nu}, U^\xi} + L_{\Phi_{\sigma_\mu}, U^\rho}\right)\widetilde{\mathcal{W}}_1\left((\nu, \mu), (\nu', \mu')\right).$$

If $\sigma_\nu > 2C_F + 2e(e + 1)L_F \int_{\mathbb{R}^d} |x|e^{-U^\xi(x)}\mathrm{d}x$ and $\sigma_\mu > 2\bar{C}_F + 2e(e + 1)\bar{L}_F \int_{\mathbb{R}^d} |y|e^{-U^\rho(y)}\mathrm{d}y$, then immediately $L_{\Psi_{\sigma_\nu}, U^\xi} + L_{\Phi_{\sigma_\mu}, U^\rho} \in (0, 1)$, hence $\left(\Psi_{\sigma_\nu}, \Phi_{\sigma_\mu}\right) : \mathcal{P}_1(\mathbb{R}^d) \times \mathcal{P}_1(\mathbb{R}^d) \to \mathcal{P}_1(\mathbb{R}^d) \times \mathcal{P}_1(\mathbb{R}^d)$ is a $\widetilde{\mathcal{W}}_1$-contraction. $\qquad\square$

**Corollary B.3** (Existence and uniqueness of the MNE). *Let Assumptions 3.1, 3.2 hold. If* (29) *and* (30) *hold, then* (6) *has a unique global MNE.*

*Proof.* Since $\sigma_\nu > 2C_F + 2e(e + 1)L_F \int_{\mathbb{R}^d} |x|e^{-U^\xi(x)}\mathrm{d}x$ and $\sigma_\mu > 2\bar{C}_F + 2e(e + 1)\bar{L}_F \int_{\mathbb{R}^d} |y|e^{-U^\rho(y)}\mathrm{d}y$, we can apply Banach's fixed point theorem for the contraction $\left(\Psi_{\sigma_\nu}, \Phi_{\sigma_\mu}\right)$ on the complete metric space $\left(\mathcal{P}_1(\mathbb{R}^d) \times \mathcal{P}_1(\mathbb{R}^d), \widetilde{\mathcal{W}}_1\right)$ and deduce that the fixed point problem $\left(\Psi_{\sigma_\nu}[\nu, \mu], \Phi_{\sigma_\mu}[\nu, \mu]\right) = (\nu, \mu)$ admits a unique solution. We also know from Proposition C.4 that any MNE $\left(\nu^*_{\sigma_\nu}, \mu^*_{\sigma_\mu}\right) \in \mathcal{P}_1(\mathbb{R}^d) \times \mathcal{P}_1(\mathbb{R}^d)$ of $F^{\sigma_\nu, \sigma_\mu}$ is equivalent to $\lambda$, and for $\lambda$-a.a. $x, y \in \mathbb{R}^d$,

$$\nu^*_{\sigma_\nu}(\mathrm{d}x) = \frac{\exp\left(-\frac{1}{\sigma_\nu}\frac{\delta F}{\delta \nu}(\nu^*_{\sigma_\nu}, \mu^*_{\sigma_\mu}, x) - U^\xi(x)\right)}{\int_{\mathbb{R}^d} \exp\left(-\frac{1}{\sigma_\nu}\frac{\delta F}{\delta \nu}(\nu^*_{\sigma_\nu}, \mu^*_{\sigma_\mu}, x') - U^\xi(x')\right)\mathrm{d}x'}\mathrm{d}x.$$

$$\mu^*_{\sigma_\mu}(\mathrm{d}y) = \frac{\exp\left(\frac{1}{\sigma_\mu}\frac{\delta F}{\delta \mu}(\nu^*_{\sigma_\nu}, \mu^*_{\sigma_\mu}, y) - U^\rho(y)\right)}{\int_{\mathbb{R}^d} \exp\left(\frac{1}{\sigma_\mu}\frac{\delta F}{\delta \mu}(\nu^*_{\sigma_\nu}, \mu^*_{\sigma_\mu}, y') - U^\rho(y')\right)\mathrm{d}y'}\mathrm{d}y.$$

Therefore, $\left(\Psi_{\sigma_\nu}[\nu^*_{\sigma_\nu}, \mu^*_{\sigma_\mu}], \Phi_{\sigma_\mu}[\nu^*_{\sigma_\nu}, \mu^*_{\sigma_\mu}]\right) = (\nu^*_{\sigma_\nu}, \mu^*_{\sigma_\mu})$ due to the definition of $\Psi_{\sigma_\nu}, \Phi_{\sigma_\mu}$. Since $\left(\Psi_{\sigma_\nu}[\nu, \mu], \Phi_{\sigma_\mu}[\nu, \mu]\right) = (\nu, \mu)$ admits a unique solution, it follows that $F^{\sigma_\nu, \sigma_\mu}$ has a unique MNE, that is $\left(\nu^*_{\sigma_\nu}, \mu^*_{\sigma_\mu}\right)$, and hence it is actually the global MNE. $\qquad\square$

**Theorem B.4** (Stability of the flow with respect to $\sigma_\nu, \sigma_\mu$ and $(\nu_0, \mu_0)$ in $\widetilde{\mathcal{W}}_1$). *Let Assumptions 3.1, 3.2 hold. Let $(\nu_t, \mu_t)_{t\geq 0}, (\nu'_t, \mu'_t)_{t\geq 0} \subset \mathcal{P}_1(\mathbb{R}^d)$ be the solutions of* (8) *with parameters $(\sigma_\nu, \sigma_\mu), (\sigma'_\nu, \sigma'_\mu)$ and initial conditions $(\nu_0, \mu_0), (\nu'_0, \mu'_0) \in \mathcal{P}_1^\lambda(\mathbb{R}^d)$, respectively. Let $(\nu^*_{\sigma_\nu}, \mu^*_{\sigma_\mu})$ and $(\nu^*_{\sigma'_\nu}, \mu^*_{\sigma'_\mu})$ be the unique MNE of* (6) *with parameters $(\sigma_\nu, \sigma_\mu)$ and $(\sigma'_\nu, \sigma'_\mu)$, respectively. If*

$$\sigma_\nu > 2C_F + 2e(e+1)L_F \frac{\alpha_\nu}{\min\{\alpha_\nu, \alpha_\mu\}} \int_{\mathbb{R}^d} |x| e^{-U^\xi(x)} \mathrm{d}x, \tag{41}$$

*and*

$$\sigma_\mu > 2\bar{C}_F + 2e(e+1)\bar{L}_F \frac{\alpha_\mu}{\min\{\alpha_\nu, \alpha_\mu\}} \int_{\mathbb{R}^d} |y| e^{-U^\rho(y)} \mathrm{d}y, \tag{42}$$

*there exists*

$$
\begin{aligned}
&L_{\Psi, \sigma_\nu, \sigma'_\nu, U^\xi} \\
&:= \frac{C_F}{\sigma_\nu \sigma'_\nu} \exp\left( C_F \left( \min\{\sigma_\nu, \sigma'_\nu\} + \frac{1}{\sigma'_\nu} \right) \right) \left( 1 + \exp\left( \frac{2C_F}{\sigma_\nu} \right) \right) \int_{\mathbb{R}^d} |x| e^{-U^\xi(x)} \mathrm{d}x > 0,
\end{aligned}
$$

$$
\begin{aligned}
&L_{\Phi, \sigma_\mu, \sigma'_\mu, U^\rho} \\
&:= \frac{\bar{C}_F}{\sigma_\mu \sigma'_\mu} \exp\left( \bar{C}_F \left( \min\{\sigma_\mu, \sigma'_\mu\} + \frac{1}{\sigma'_\mu} \right) \right) \left( 1 + \exp\left( \frac{2\bar{C}_F}{\sigma_\mu} \right) \right) \int_{\mathbb{R}^d} |y| e^{-U^\rho(y)} \mathrm{d}y > 0,
\end{aligned}
$$

*such that for all $t \geq 0$,*

$$
\begin{aligned}
\widetilde{\mathcal{W}}_1 \left( (\nu_t, \mu_t), (\nu'_t, \mu'_t) \right) &\leq e^{-t\left( \min\{\alpha_\nu, \alpha_\mu\} - \left( \alpha_\nu L_{\Psi_{\sigma_\nu}, U^\xi} + \alpha_\mu L_{\Phi_{\sigma_\mu}, U^\rho} \right) \right)} \widetilde{\mathcal{W}}_1 \left( (\nu_0, \mu_0), (\nu'_0, \mu'_0) \right) \\
&+ \frac{\alpha_\nu L_{\Psi, \sigma_\nu, \sigma'_\nu, U^\xi} |\sigma'_\nu - \sigma_\nu| + \alpha_\mu L_{\Phi, \sigma_\mu, \sigma'_\mu, U^\rho} |\sigma'_\mu - \sigma_\mu|}{\min\{\alpha_\nu, \alpha_\mu\} - \left( \alpha_\nu L_{\Psi_{\sigma_\nu}, U^\xi} + \alpha_\mu L_{\Phi_{\sigma_\mu}, U^\rho} \right)} \times \\
&\times \left( 1 - e^{-t\left( \min\{\alpha_\nu, \alpha_\mu\} - \left( \alpha_\nu L_{\Psi_{\sigma_\nu}, U^\xi} + \alpha_\mu L_{\Phi_{\sigma_\mu}, U^\rho} \right) \right)} \right),
\end{aligned}
$$

$$
\begin{aligned}
\widetilde{\mathcal{W}}_1 \left( (\nu^*_{\sigma_\nu}, \mu^*_{\sigma_\mu}), (\nu^*_{\sigma'_\nu}, \mu^*_{\sigma'_\mu}) \right) &\leq \frac{\alpha_\nu L_{\Psi, \sigma_\nu, \sigma'_\nu, U^\xi} |\sigma'_\nu - \sigma_\nu| + \alpha_\mu L_{\Phi, \sigma_\mu, \sigma'_\mu, U^\rho} |\sigma'_\mu - \sigma_\mu|}{\min\{\alpha_\nu, \alpha_\mu\} - \left( \alpha_\nu L_{\Psi_{\sigma_\nu}, U^\xi} + \alpha_\mu L_{\Phi_{\sigma_\mu}, U^\rho} \right)} \times \\
&\times \left( 1 - e^{-t\left( \min\{\alpha_\nu, \alpha_\mu\} - \left( \alpha_\nu L_{\Psi_{\sigma_\nu}, U^\xi} + \alpha_\mu L_{\Phi_{\sigma_\mu}, U^\rho} \right) \right)} \right).
\end{aligned}
$$

*Proof. Step 1:* First, we show that the maps $\Psi_{\sigma_\nu}, \Phi_{\sigma_\mu}$ are $\mathcal{W}_1$-Lipschitz with respect to $\sigma_\nu, \sigma_\mu$, respectively. From Assumption 3.1 and the estimate $|e^x - e^y| \leq e^{\max\{x,y\}} |x - y|$, we have

$$
\begin{aligned}
&\left| \exp\left( -\frac{1}{\sigma_\nu} \frac{\delta F}{\delta \nu}(\nu, \mu, x) - U^\xi(x) \right) - \exp\left( -\frac{1}{\sigma'_\nu} \frac{\delta F}{\delta \nu}(\nu, \mu, x) - U^\xi(x) \right) \right| \\
&\qquad\qquad\qquad\qquad\qquad\qquad \leq C_F \frac{|\sigma_\nu - \sigma'_\nu|}{\sigma_\nu \sigma'_\nu} \exp\left( C_F \min\{\sigma_\nu, \sigma'_\nu\} \right) e^{-U^\xi(x)}. \tag{43}
\end{aligned}
$$

Integrating the previous inequality with respect to $x$, we obtain

$$\left| Z_{\sigma_\nu}(\nu, \mu) - Z_{\sigma'_\nu}(\nu, \mu) \right| \leq C_F \frac{|\sigma_\nu - \sigma'_\nu|}{\sigma_\nu \sigma'_\nu} \exp\left( C_F \min\{\sigma_\nu, \sigma'_\nu\} \right). \tag{44}$$

Therefore, we have that

$$
\begin{aligned}
\left|\Psi_{\sigma_\nu}[\nu,\mu](x) - \Psi_{\sigma'_\nu}[\nu,\mu](x)\right| = &\left| \frac{1}{Z_{\sigma_\nu}(\nu,\mu)} \exp\left(-\frac{1}{\sigma_\nu}\frac{\delta F}{\delta \nu}(\nu,\mu,x) - U^\xi(x)\right) \right.\\
&- \frac{1}{Z_{\sigma'_\nu}(\nu,\mu)} \exp\left(-\frac{1}{\sigma_\nu}\frac{\delta F}{\delta \nu}(\nu,\mu,x) - U^\xi(x)\right)\\
&+ \frac{1}{Z_{\sigma'_\nu}(\nu,\mu)} \exp\left(-\frac{1}{\sigma_\nu}\frac{\delta F}{\delta \nu}(\nu,\mu,x) - U^\xi(x)\right)\\
&\left.- \frac{1}{Z_{\sigma'_\nu}(\nu,\mu)} \exp\left(-\frac{1}{\sigma'_\nu}\frac{\delta F}{\delta \nu}(\nu,\mu,x) - U^\xi(x)\right) \right|\\
\leq &\exp\left(-\frac{1}{\sigma_\nu}\frac{\delta F}{\delta \nu}(\nu,\mu,x) - U^\xi(x)\right) \frac{\left|Z_{\sigma'_\nu}(\nu,\mu) - Z_{\sigma_\nu}(\nu,\mu)\right|}{Z_{\sigma_\nu}(\nu,\mu)Z_{\sigma'_\nu}(\nu,\mu)}\\
&+ \frac{1}{Z_{\sigma'_\nu}(\nu,\mu)} \left| \exp\left(-\frac{1}{\sigma_\nu}\frac{\delta F}{\delta \nu}(\nu,\mu,x) - U^\xi(x)\right) \right.\\
&\left.- \exp\left(-\frac{1}{\sigma'_\nu}\frac{\delta F}{\delta \nu}(\nu,\mu,x) - U^\xi(x)\right) \right|.
\end{aligned}
$$

Using estimates (31), (32), (43) and (44), we arrive at

$$
\left|\Psi_{\sigma_\nu}[\nu,\mu](x) - \Psi_{\sigma'_\nu}[\nu,\mu](x)\right| \leq L_{\Psi,\sigma_\nu,\sigma'_\nu}|\sigma_\nu - \sigma'_\nu|e^{-U^\xi(x)}, \tag{45}
$$

with

$$
L_{\Psi,\sigma_\nu,\sigma'_\nu} := \frac{C_F}{\sigma_\nu\sigma'_\nu} \exp\left(C_F\left(\min\{\sigma_\nu,\sigma'_\nu\} + \frac{1}{\sigma'_\nu}\right)\right)\left(1 + \exp\left(\frac{2C_F}{\sigma_\nu}\right)\right) > 0.
$$

Proving

$$
\left|\Phi_{\sigma_\mu}[\nu,\mu](y) - \Phi_{\sigma'_\mu}[\nu,\mu](y)\right| \leq L_{\Phi,\sigma_\mu,\sigma'_\mu}|\sigma_\mu - \sigma'_\mu|e^{-U^\rho(y)} \tag{46}
$$

follows the same steps as above but with

$$
L_{\Phi,\sigma_\mu,\sigma'_\mu} := \frac{\bar{C}_F}{\sigma_\mu\sigma'_\mu} \exp\left(\bar{C}_F\left(\min\{\sigma_\mu,\sigma'_\mu\} + \frac{1}{\sigma'_\mu}\right)\right)\left(1 + \exp\left(\frac{2\bar{C}_F}{\sigma_\mu}\right)\right).
$$

Now, applying [8, Lemma 16] with $p = 1$ and $\mu(\mathrm{d}x) = e^{-U^\xi(x)}\mathrm{d}x$ gives

$$
\mathcal{W}_1\left(\Psi_{\sigma_\nu}[\nu,\mu], \Psi_{\sigma'_\nu}[\nu,\mu]\right) \leq \int_{\mathbb{R}^d} |x|e^{-U^\xi(x)}\mathrm{d}x \left\| \frac{\Psi_{\sigma_\nu}[\nu,\mu](\cdot)}{e^{-U^\xi(\cdot)}} - \frac{\Psi_{\sigma'_\nu}[\nu,\mu](\cdot)}{e^{-U^\xi(\cdot)}} \right\|_{L^\infty(\mathbb{R}^d)}.
$$

Hence, using (45) and setting

$$
L_{\Psi,\sigma_\nu,\sigma'_\nu,U^\xi} := L_{\Psi,\sigma_\nu,\sigma'_\nu}\int_{\mathbb{R}^d}|x|e^{-U^\xi(x)}\mathrm{d}x > 0
$$

gives

$$
\mathcal{W}_1\left(\Psi_{\sigma_\nu}[\nu,\mu], \Psi_{\sigma'_\nu}[\nu,\mu]\right) \leq L_{\Psi,\sigma_\nu,\sigma'_\nu,U^\xi}|\sigma_\nu - \sigma'_\nu|.
$$

We similarly obtain

$$
\mathcal{W}_1\left(\Phi_{\sigma_\mu}[\nu,\mu], \Phi_{\sigma'_\mu}[\nu,\mu]\right) \leq L_{\Phi,\sigma_\mu,\sigma'_\mu,U^\rho}|\sigma_\mu - \sigma'_\mu|
$$

with constant

$$
L_{\Phi,\sigma_\mu,\sigma'_\mu,U^\rho} := L_{\Phi,\sigma_\mu,\sigma'_\mu}\int_{\mathbb{R}^d}|y|e^{-U^\rho(y)}\mathrm{d}y > 0.
$$

*Step 2:* Now, we prove the stability of the solution to (8) with respect to $\sigma_\nu, \sigma_\mu$ and $(\nu_0, \mu_0)$. Since $(\nu_0, \mu_0), (\nu_0', \mu_0') \in \mathcal{P}_1^\lambda(\mathbb{R}^d) \times \mathcal{P}_1^\lambda(\mathbb{R}^d)$, it follows by Theorem C.5 that $(\nu_t, \mu_t)_{t \geq 0}, (\nu_t', \mu_t')_{t \geq 0} \subset \mathcal{P}_1^\lambda(\mathbb{R}^d) \times \mathcal{P}_1^\lambda(\mathbb{R}^d)$. Let $\psi \in \mathrm{Lip}_1(\mathbb{R}^d)$. Then we have that

$$\int_{\mathbb{R}^d} \psi(x)\,(\nu_t(x) - \nu_t'(x))\,\mathrm{d}x$$

$$= \int_{\mathbb{R}^d} \psi(x) \left( e^{-\alpha_\nu t} \nu_0(x) + \int_0^t \alpha_\nu e^{-\alpha_\nu(t-s)} \Psi_{\sigma_\nu}\left[\nu_s, \mu_s\right](x)\mathrm{d}s \right.$$

$$\left. - e^{-\alpha_\nu t} \nu_0'(x) - \int_0^t \alpha_\nu e^{-\alpha_\nu(t-s)} \Psi_{\sigma_\nu'}[\nu_s', \mu_s'](x)\mathrm{d}s \right) \mathrm{d}x$$

$$= \int_{\mathbb{R}^d} \psi(x) \left( e^{-\alpha_\nu t}\left(\nu_0(x) - \nu_0'(x)\right) \right.$$

$$\left. + \int_0^t \alpha_\nu e^{-\alpha_\nu(t-s)} \left( \Psi_{\sigma_\nu}\left[\nu_s, \mu_s\right](x) - \Psi_{\sigma_\nu'}\left[\nu_s', \mu_s'\right](x) \right) \mathrm{d}s \right) \mathrm{d}x$$

$$= e^{-\alpha_\nu t} \int_{\mathbb{R}^d} \psi(x)\,(\nu_0(x) - \nu_0'(x))\,\mathrm{d}x$$

$$+ \int_{\mathbb{R}^d} \psi(x) \int_0^t \alpha_\nu e^{-\alpha_\nu(t-s)} \left( \Psi_{\sigma_\nu}\left[\nu_s, \mu_s\right](x) - \Psi_{\sigma_\nu'}\left[\nu_s', \mu_s'\right](x) \right) \mathrm{d}s\mathrm{d}x$$

$$= e^{-\alpha_\nu t} \int_{\mathbb{R}^d} \psi(x)\,(\nu_0(x) - \nu_0'(x))\,\mathrm{d}x$$

$$+ \int_0^t \alpha_\nu e^{-\alpha_\nu(t-s)} \int_{\mathbb{R}^d} \psi(x) \left( \Psi_{\sigma_\nu}\left[\nu_s, \mu_s\right](x) - \Psi_{\sigma_\nu'}\left[\nu_s', \mu_s'\right](x) \right) \mathrm{d}x\mathrm{d}s$$

$$\leq e^{-\alpha_\nu t} \mathcal{W}_1(\nu_0, \nu_0') + \int_0^t \alpha_\nu e^{-\alpha_\nu(t-s)} \mathcal{W}_1\left( \Psi_{\sigma_\nu}\left[\nu_s, \mu_s\right], \Psi_{\sigma_\nu'}\left[\nu_s', \mu_s'\right] \right) \mathrm{d}s$$

$$\leq e^{-\alpha_\nu t} \mathcal{W}_1(\nu_0, \nu_0') + \int_0^t \alpha_\nu e^{-\alpha_\nu(t-s)} \Big( \mathcal{W}_1\left( \Psi_{\sigma_\nu}\left[\nu_s, \mu_s\right], \Psi_{\sigma_\nu}\left[\nu_s', \mu_s'\right] \right)$$

$$+ \mathcal{W}_1\left( \Psi_{\sigma_\nu}\left[\nu_s', \mu_s'\right], \Psi_{\sigma_\nu'}\left[\nu_s', \mu_s'\right] \right) \Big) \mathrm{d}s$$

$$\leq e^{-\alpha_\nu t} \mathcal{W}_1(\nu_0, \nu_0')$$

$$+ \int_0^t \alpha_\nu e^{-\alpha_\nu(t-s)} \left( \mathcal{W}_1\left( \Psi_{\sigma_\nu}\left[\nu_s, \mu_s\right], \Psi_{\sigma_\nu}\left[\nu_s', \mu_s'\right] \right) + L_{\Psi, \sigma_\nu, \sigma_\nu', U^\xi} |\sigma_\nu - \sigma_\nu'| \right) \mathrm{d}s$$

$$\leq e^{-\alpha_\nu t} \mathcal{W}_1(\nu_0, \nu_0')$$

$$+ \int_0^t \alpha_\nu e^{-\alpha_\nu(t-s)} \left( L_{\Psi_{\sigma_\nu}, U^\xi} \left( \mathcal{W}_1(\nu_s, \nu_s') + \mathcal{W}_1(\mu_s, \mu_s') \right) + L_{\Psi, \sigma_\nu, \sigma_\nu', U^\xi} |\sigma_\nu - \sigma_\nu'| \right) \mathrm{d}s,$$

where in the last equality we used Fubini's theorem, in the first inequality we used the definition of $\mathcal{W}_1$ and the last three inequalities follow from the triangle inequality, *Step 1* and (39), respectively. Taking supremum over $\psi$ gives

$$\mathcal{W}_1(\nu_t, \nu_t') \leq e^{-\alpha_\nu t} \mathcal{W}_1(\nu_0, \nu_0')$$

$$+ \int_0^t \alpha_\nu e^{-\alpha_\nu(t-s)} \left( L_{\Psi_{\sigma_\nu}, U^\xi} \left( \mathcal{W}_1(\nu_s, \nu_s') + \mathcal{W}_1(\mu_s, \mu_s') \right) + L_{\Psi, \sigma_\nu, \sigma_\nu', U^\xi} |\sigma_\nu - \sigma_\nu'| \right) \mathrm{d}s.$$

Similarly using (40), we obtain

$$\mathcal{W}_1(\mu_t, \mu_t') \leq e^{-\alpha_\mu t} \mathcal{W}_1(\mu_0, \mu_0')$$

$$+ \int_0^t \alpha_\mu e^{-\alpha_\mu(t-s)} \left( L_{\Phi_{\sigma_\mu}, U^\rho} \left( \mathcal{W}_1(\nu_s, \nu_s') + \mathcal{W}_1(\mu_s, \mu_s') \right) + L_{\Phi, \sigma_\mu, \sigma_\mu', U^\rho} |\sigma_\mu - \sigma_\mu'| \right) \mathrm{d}s.$$

Using the bound $\alpha_\nu, \alpha_\mu \geq \min\{\alpha_\nu, \alpha_\mu\}$ we can add the previous two inequalities, and using the definition of $\widetilde{\mathcal{W}}_1$ and the notation $\left(\Psi_{\sigma_\nu}, \Phi_{\sigma_\mu}\right)[\nu, \mu] := \left(\Psi_{\sigma_\nu}[\nu, \mu], \Phi_{\sigma_\mu}[\nu, \mu]\right)$ we obtain

$$
\begin{aligned}
\widetilde{\mathcal{W}}_1\left((\nu_t, \mu_t), (\nu'_t, \mu'_t)\right) &\leq e^{-\min\{\alpha_\nu, \alpha_\mu\}t}\widetilde{\mathcal{W}}_1\left((\nu_0, \mu_0), (\nu'_0, \mu'_0)\right) \\
&+ \int_0^t e^{-\min\{\alpha_\nu, \alpha_\mu\}(t-s)}\left(\alpha_\nu L_{\Psi_{\sigma_\nu}, U^\xi} + \alpha_\mu L_{\Phi_{\sigma_\mu}, U^\rho}\right)\widetilde{\mathcal{W}}_1\left((\nu_s, \mu_s), (\nu'_s, \mu'_s)\right)\mathrm{d}s \\
&+ \int_0^t e^{-\min\{\alpha_\nu, \alpha_\mu\}(t-s)}\left(\alpha_\nu L_{\Psi, \sigma_\nu, \sigma'_\nu, U^\xi}|\sigma_\nu - \sigma'_\nu| + \alpha_\mu L_{\Phi, \sigma_\mu, \sigma'_\mu, U^\rho}|\sigma_\mu - \sigma'_\mu|\right)\mathrm{d}s.
\end{aligned}
$$

Since $\sigma_\nu > 2C_F + 2e(e+1)L_F\frac{\alpha_\nu}{\min\{\alpha_\nu, \alpha_\mu\}}\int_{\mathbb{R}^d}|x|e^{-U^\xi(x)}\mathrm{d}x$
and $\sigma_\mu > 2\bar{C}_F + 2e(e+1)\bar{L}_F\frac{\alpha_\mu}{\min\{\alpha_\nu, \alpha_\mu\}}\int_{\mathbb{R}^d}|y|e^{-U^\rho(y)}\mathrm{d}y$, it follows that $\alpha_\nu L_{\Psi_{\sigma_\nu}, U^\xi} + \alpha_\mu L_{\Phi_{\sigma_\mu}, U^\rho} < \min\{\alpha_\nu, \alpha_\mu\}$.

Therefore, applying Gronwall's lemma to the function $t \mapsto e^{\min\{\alpha_\nu, \alpha_\mu\}t}\widetilde{\mathcal{W}}_1\left((\nu_t, \mu_t), (\nu'_t, \mu'_t)\right)$ yields

$$
\begin{aligned}
\widetilde{\mathcal{W}}_1\left((\nu_t, \mu_t), (\nu'_t, \mu'_t)\right) &\leq e^{-t\left(\min\{\alpha_\nu, \alpha_\mu\} - \left(\alpha_\nu L_{\Psi_{\sigma_\nu}, U^\xi} + \alpha_\mu L_{\Phi_{\sigma_\mu}, U^\rho}\right)\right)}\widetilde{\mathcal{W}}_1\left((\nu_0, \mu_0), (\nu'_0, \mu'_0)\right) \\
&+ \frac{\alpha_\nu L_{\Psi, \sigma_\nu, \sigma'_\nu, U^\xi}|\sigma_\nu - \sigma'_\nu| + \alpha_\mu L_{\Phi, \sigma_\mu, \sigma'_\mu, U^\rho}|\sigma_\mu - \sigma'_\mu|}{\min\{\alpha_\nu, \alpha_\mu\} - \left(\alpha_\nu L_{\Psi_{\sigma_\nu}, U^\xi} + \alpha_\mu L_{\Phi_{\sigma_\mu}, U^\rho}\right)} \times \\
&\times \left(1 - e^{-t\left(\min\{\alpha_\nu, \alpha_\mu\} - \left(\alpha_\nu L_{\Psi_{\sigma_\nu}, U^\xi} + \alpha_\mu L_{\Phi_{\sigma_\mu}, U^\rho}\right)\right)}\right).
\end{aligned}
\tag{47}
$$

Since $\alpha_\nu, \alpha_\mu \geq \min\{\alpha_\nu, \alpha_\mu\}$, it follows that the lower bounds for $\sigma_\nu, \sigma_\mu$ in Corollary B.3 hold, hence $(\nu^*_{\sigma_\nu}, \mu^*_{\sigma_\mu}), (\nu^*_{\sigma'_\nu}, \mu^*_{\sigma'_\mu})$ are the MNE of (6) with parameters $(\sigma_\nu, \sigma_\mu), (\sigma'_\nu, \sigma'_\mu)$, respectively. Moreover, since $\nu^*_{\sigma_\nu} = \Psi_{\sigma_\nu}[\nu^*_{\sigma_\nu}, \mu^*_{\sigma_\mu}], \mu^*_{\sigma_\mu} = \Phi_{\sigma_\mu}[\nu^*_{\sigma_\nu}, \mu^*_{\sigma_\mu}]$, we deduce that $(\nu^*_{\sigma_\nu}, \mu^*_{\sigma_\mu}), (\nu^*_{\sigma'_\nu}, \mu^*_{\sigma'_\mu})$ are solutions to (8) with parameters $(\sigma_\nu, \sigma_\mu), (\sigma'_\nu, \sigma'_\mu)$, respectively. respectively. Then using (47) yields

$$
\begin{aligned}
&\widetilde{\mathcal{W}}_1\left((\nu^*_{\sigma_\nu}, \mu^*_{\sigma_\mu}), (\nu^*_{\sigma'_\nu}, \mu^*_{\sigma'_\mu})\right) \\
&\leq e^{-t\left(\min\{\alpha_\nu, \alpha_\mu\} - \left(\alpha_\nu L_{\Psi_{\sigma_\nu}, U^\xi} + \alpha_\mu L_{\Phi_{\sigma_\mu}, U^\rho}\right)\right)}\widetilde{\mathcal{W}}_1\left((\nu^*_{\sigma_\nu}, \mu^*_{\sigma_\mu}), (\nu^*_{\sigma'_\nu}, \mu^*_{\sigma'_\mu})\right) \\
&+ \frac{\alpha_\nu L_{\Psi, \sigma_\nu, \sigma'_\nu, U^\xi}|\sigma_\nu - \sigma'_\nu| + \alpha_\mu L_{\Phi, \sigma_\mu, \sigma'_\mu, U^\rho}|\sigma_\mu - \sigma'_\mu|}{\min\{\alpha_\nu, \alpha_\mu\} - \left(\alpha_\nu L_{\Psi_{\sigma_\nu}, U^\xi} + \alpha_\mu L_{\Phi_{\sigma_\mu}, U^\rho}\right)} \times \\
&\times \left(1 - e^{-t\left(\min\{\alpha_\nu, \alpha_\mu\} - \left(\alpha_\nu L_{\Psi_{\sigma_\nu}, U^\xi} + \alpha_\mu L_{\Phi_{\sigma_\mu}, U^\rho}\right)\right)}\right).
\end{aligned}
$$

Therefore, since $\min\{\alpha_\nu, \alpha_\mu\} - \left(\alpha_\nu L_{\Psi_{\sigma_\nu}, U^\xi} + \alpha_\mu L_{\Phi_{\sigma_\mu}, U^\rho}\right) > 0$, we obtain

$$
\begin{aligned}
&\widetilde{\mathcal{W}}_1\left((\nu^*_{\sigma_\nu}, \mu^*_{\sigma_\mu}), (\nu^*_{\sigma'_\nu}, \mu^*_{\sigma'_\mu})\right) \\
&\leq \frac{\alpha_\nu L_{\Psi, \sigma_\nu, \sigma'_\nu, U^\xi}|\sigma_\nu - \sigma'_\nu| + \alpha_\mu L_{\Phi, \sigma_\mu, \sigma'_\mu, U^\rho}|\sigma_\mu - \sigma'_\mu|}{\min\{\alpha_\nu, \alpha_\mu\} - \left(\alpha_\nu L_{\Psi_{\sigma_\nu}, U^\xi} + \alpha_\mu L_{\Phi_{\sigma_\mu}, U^\rho}\right)} \times \\
&\times \left(1 - e^{-t\left(\min\{\alpha_\nu, \alpha_\mu\} - \left(\alpha_\nu L_{\Psi_{\sigma_\nu}, U^\xi} + \alpha_\mu L_{\Phi_{\sigma_\mu}, U^\rho}\right)\right)}\right).
\end{aligned}
$$

$\square$

## C  Auxiliary results for the Best Response flow

### C.1  Single-agent optimization

We first recall a necessary condition for the optimality of local minimizers of (1).

**Proposition C.1** (Necessary condition for optimality [18, Proposition 2.5]). *Let Assumptions 2.1, 2.2 hold. If $\nu_\sigma^* \in \mathcal{P}_1(\mathbb{R}^d)$ is a minimizer of $F^\sigma$, then $\nu_\sigma^*$ is equivalent to $\lambda$, and for $\lambda$-a.a. $x \in \mathbb{R}^d$,*

$$\nu_\sigma^*(\mathrm{d}x) = \frac{e^{-\frac{1}{\sigma}\frac{\delta F}{\delta \nu}(\nu_\sigma^*,x) - U^\xi(x)}}{\int_{\mathbb{R}^d} e^{-\frac{1}{\sigma}\frac{\delta F}{\delta \nu}(\nu_\sigma^*,x') - U^\xi(x')}\mathrm{d}x'}\mathrm{d}x.$$

Next we give the proof of the existence and uniqueness of the flow (4). While the proof is skipped in [8, Proposition 7], we present it by following a classical Picard iteration technique.

**Proposition C.2** (Existence and uniqueness of the flow). *Let Assumptions 2.1, 2.2 hold and let $\nu_0 \in \mathcal{P}_1(\mathbb{R}^d)$. Then there exists a unique solution $(\nu_t)_{t\geq0} \in C\left([0,\infty];\mathcal{P}_1(\mathbb{R}^d)\right)$ to (4) and the solution depends continuously on the initial condition. Moreover, if $\nu_0 \in \mathcal{P}_1^\lambda(\mathbb{R}^d)$, then $(\nu_t)_{t\geq0}$ admits the density $(\nu_t(\cdot))_{t\geq0} \subset \mathcal{P}_1^\lambda(\mathbb{R}^d)$ such that, for every $x \in \mathbb{R}^d$, it holds that $t \mapsto \nu_t \in C^1\left([0,\infty),\mathcal{P}_1^\lambda(\mathbb{R}^d)\right)$, and*

$$\mathrm{d}\nu_t(x) = \alpha\left(\Psi_\sigma[\nu_t](x) - \nu_t(x)\right)\mathrm{d}t, \quad t > 0, \tag{48}$$

*for some initial condition $\nu_0(x) \in \mathcal{P}_1^\lambda(\mathbb{R}^d)$.*

*Proof of Theorem C.2. Step 1: Existence of flow on $[0,T]$.* We will define a Picard iteration scheme as follows. Fix $T > 0$ and for each $n \geq 1$, fix $\nu_0^{(n)} = \nu_0^{(0)} = \nu_0 \in \mathcal{P}_1(\mathbb{R}^d)$. Then define the flow $(\nu_t^{(n)})_{t\in[0,T]}$ by

$$\nu_t^{(n)} := e^{-\alpha t}\nu_0 + \int_0^t \alpha e^{-\alpha(t-s)}\Psi_\sigma\left[\nu_s^{(n-1)}\right]\mathrm{d}s, \quad t \in [0,T]. \tag{49}$$

For fixed $T > 0$, we consider the sequence of flows $\left((\nu_t^{(n)})_{t\in[0,T]}\right)_{n=0}^\infty$ in $\left(\mathcal{P}_1(\mathbb{R}^d)^{[0,T]}, \mathcal{W}_1^{[0,T]}\right)$, where, for any $(\nu_t)_{t\in[0,T]} \in \mathcal{P}_1(\mathbb{R}^d)^{[0,T]}$, the distance $\mathcal{W}_1^{[0,T]}$ is defined by

$$\mathcal{W}_1^{[0,T]}\left((\nu_t)_{t\in[0,T]}, (\nu_t')_{t\in[0,T]}\right) := \int_0^T \mathcal{W}_1(\nu_t, \nu_t')\mathrm{d}t.$$

Since $\left(\mathcal{P}_1(\mathbb{R}^d), \mathcal{W}_1\right)$ is a complete metric space, we can apply the argument from [41, Lemma A.5] with $p = 1$ to conclude that $\left(\mathcal{P}_1(\mathbb{R}^d)^{[0,T]}, \int_0^T \mathcal{W}_1(\nu_t, \nu_t')\mathrm{d}t\right)$ is a complete metric space.

We consider the Picard iteration mapping $\varphi\left((\nu_t^{(n-1)})_{t\in[0,T]}\right) := (\nu_t^{(n)})_{t\in[0,T]}$ defined via (49) and show that $\varphi$ admits a unique fixed point $(\nu_t)_{t\in[0,T]}$ in the complete space $\left(\mathcal{P}_1(\mathbb{R}^d)^{[0,T]}, \mathcal{W}_1^{[0,T]}\right)$. Then this fixed point is the solution to (4).

**Lemma C.3.** *The mapping $\varphi\left((\nu_t^{(n-1)})_{t\in[0,T]}\right) := (\nu_t^{(n)})_{t\in[0,T]}$ defined via (49) admits a unique fixed point in $\left(\mathcal{P}_1(\mathbb{R}^d)^{[0,T]}, \mathcal{W}_1^{[0,T]}\right)$.*

*Proof of Lemma C.3. Step 1: The sequence of flows $\left((\nu_t^{(n)})_{t\in[0,T]}\right)_{n=0}^\infty$ is a Cauchy sequence in $\left(\mathcal{P}_1(\mathbb{R}^d)^{[0,T]}, \mathcal{W}_1^{[0,T]}\right)$.*

Let $\psi \in \mathrm{Lip}_1(\mathbb{R}^d)$. Then we have that

$$
\int_{\mathbb{R}^d} \psi(x) \left( \nu_t^{(n)} - \nu_t^{(n-1)} \right) (\mathrm{d}x)
$$

$$
= \int_{\mathbb{R}^d} \alpha\psi(x) \left( \int_0^t e^{-\alpha(t-s)} \Psi_\sigma \left[ \nu_s^{(n-1)} \right] \mathrm{d}s - \int_0^t e^{-\alpha(t-s)} \Psi_\sigma \left[ \nu_s^{(n-2)} \right] \mathrm{d}s \right) (\mathrm{d}x)
$$

$$
= \int_{\mathbb{R}^d} \alpha\psi(x) \left( \int_0^t e^{-\alpha(t-s)} \left( \Psi_\sigma \left[ \nu_s^{(n-1)} \right] - \Psi_\sigma \left[ \nu_s^{(n-2)} \right] \right) \mathrm{d}s \right) (\mathrm{d}x)
$$

$$
= \int_0^t \alpha e^{-\alpha(t-s)} \int_{\mathbb{R}^d} \psi(x) \left( \Psi_\sigma \left[ \nu_s^{(n-1)} \right] - \Psi_\sigma \left[ \nu_s^{(n-2)} \right] \right) (\mathrm{d}x) \mathrm{d}s \tag{50}
$$

$$
\leq \int_0^t \alpha e^{-\alpha(t-s)} \mathcal{W}_1 \left( \Psi_\sigma \left[ \nu_s^{(n-1)} \right], \Psi_\sigma \left[ \nu_s^{(n-2)} \right] \right) \mathrm{d}s
$$

$$
\leq \alpha L_{\Psi_\sigma, U^\xi} \int_0^t \mathcal{W}_1 \left( \nu_s^{(n-1)}, \nu_s^{(n-2)} \right) \mathrm{d}s,
$$

where in the third equality we used Fubini's theorem, in the first inequality we used the definition of $\mathcal{W}_1$, and in the last inequality we used Theorem 2.3 and the fact that $e^{-\alpha(t-s)} \leq 1$ for $s \in [0, t]$.

Taking supremum over $\psi$ in (50) gives

$$
\mathcal{W}_1 \left( \nu_t^{(n)}, \nu_t^{(n-1)} \right) \leq \alpha L_{\Psi_\sigma, U^\xi} \int_0^t \mathcal{W}_1 \left( \nu_s^{(n-1)}, \nu_s^{(n-2)} \right) \mathrm{d}s
$$

$$
\leq \left( \alpha L_{\Psi_\sigma, U^\xi} \right)^{n-1} \int_0^t \int_0^{t_1} \cdots \int_0^{t_{n-2}} \mathcal{W}_1 \left( \nu_{t_{n-1}}^{(1)}, \nu_{t_{n-1}}^{(0)} \right) \mathrm{d}t_{n-1} \ldots \mathrm{d}t_2 \mathrm{d}t_1
$$

$$
\leq \left( \alpha L_{\Psi_\sigma, U^\xi} \right)^{n-1} \frac{t^{n-2}}{(n-2)!} \int_0^t \mathcal{W}_1 \left( \nu_{t_{n-1}}^{(1)}, \nu_{t_{n-1}}^{(0)} \right) \mathrm{d}t_{n-1},
$$

where in the third inequality we used the bound $\int_0^{t_{n-2}} \mathrm{d}t_{n-1} \leq \int_0^t \mathrm{d}t_{n-1}$. Hence, we obtain

$$
\int_0^T \mathcal{W}_1 \left( \nu_t^{(n)}, \nu_t^{(n-1)} \right) \mathrm{d}t \leq \left( \alpha L_{\Psi_\sigma, U^\xi} \right)^{n-1} \frac{T^{n-1}}{(n-1)!} \int_0^T \mathcal{W}_1 \left( \nu_{t_{n-1}}^{(1)}, \nu_{t_{n-1}}^{(0)} \right) \mathrm{d}t_{n-1}.
$$

Using the definition of $\mathcal{W}_1^{[0,T]}$, the last inequality becomes

$$
\mathcal{W}_1^{[0,T]} \left( (\nu_t^{(n)})_{t \in [0,T]}, (\nu_t^{(n-1)})_{t \in [0,T]} \right)
$$

$$
\leq \left( \alpha L_{\Psi_\sigma, U^\xi} \right)^{n-1} \frac{T^{n-1}}{(n-1)!} \mathcal{W}_1^{[0,T]} \left( (\nu_t^{(1)})_{t \in [0,T]}, (\nu_t^{(0)})_{t \in [0,T]} \right).
$$

By choosing $n$ sufficiently large, we conclude that $\left( (\nu_t^{(n)})_{t \in [0,T]} \right)_{n=0}^\infty$ is a Cauchy sequence. By completeness of $\left( \mathcal{P}_1(\mathbb{R}^d)^{[0,T]}, \mathcal{W}_1^{[0,T]} \right)$, the sequence admits a limit point $(\nu_t)_{t \in [0,T]} \in \left( \mathcal{P}_1(\mathbb{R}^d)^{[0,T]}, \mathcal{W}_1^{[0,T]} \right)$.

*Step 2: The limit point $(\nu_t)_{t \in [0,T]}$ is a fixed point of $\varphi$.* From Step 1, we obtain that for Lebesgue-almost all $t \in [0, T]$ we have

$$
\mathcal{W}_1(\nu_t^{(n)}, \nu_t) \to 0, \quad \text{as } n \to \infty.
$$

Therefore, by Theorem 2.3, for Lebesgue-almost all $t \in [0, T]$, we have that

$$
\mathcal{W}_1 \left( \Psi_\sigma \left[ \nu_t^{(n)} \right], \Psi_\sigma \left[ \nu_t \right] \right) \to 0, \quad \text{as } n \to \infty.
$$

Hence, letting $n \to \infty$ in (49) and using the dominated convergence theorem (which is possible since $\Psi_\sigma$ is uniformly bounded in $n$ due to Assumption 2.1), we conclude that $(\nu_t)_{t \in [0,T]}$ is a fixed point of $\varphi$.

*Step 3: The fixed point $(\nu_t)_{t \in [0,T]}$ of $\varphi$ is unique.* Suppose, for the contrary, that $\varphi$ admits two fixed points $(\nu_t)_{t \in [0,T]}$ and $(\bar{\nu}_t)_{t \in [0,T]}$ such that $\nu_0 = \bar{\nu}_0$. Then repeating the same calculations from (50), we arrive at

$$\mathcal{W}_1(\nu_t, \bar{\nu}_t) \leq \alpha L_{\Psi_\sigma, U^\xi} \int_0^t \mathcal{W}_1(\nu_s, \bar{\nu}_s) \mathrm{d}s.$$

For each $t \in [0,T]$, denote $f(t) := \int_0^t \mathcal{W}_1(\nu_s, \bar{\nu}_s) \mathrm{d}s$. Observe that $f \geq 0$ and $f(0) = 0$. Then, by Gronwall's lemma, we obtain

$$0 \leq f(t) \leq e^{t\alpha L_{\Psi_\sigma, U^\xi}} f(0) = 0,$$

and hence

$$\mathcal{W}_1(\nu_t, \bar{\nu}_t) = 0,$$

for Lebesgue-almost all $t \in [0,T]$, which implies

$$\nu_t = \bar{\nu}_t,$$

for Lebesgue-almost all $t \in [0,T]$. Therefore, the fixed point $(\nu_t)_{t \in [0,T]}$ of $\varphi$ must be unique.

From *Steps 1, 2 and 3*, we obtain the existence and uniqueness of a flow $(\nu_t)_{t \in [0,T]}$ satisfying (4) for any $T > 0$. □

Having proved Lemma C.3, we return to the proof of Theorem C.2.

*Step 2: Existence of the flow on $[0, \infty)$.*

From Lemma C.3, for any $T > 0$, there exists unique flow $(\nu_t)_{t \in [0,T]}$ satisfying (4). It remains to prove that the existence of this flow could be extended to $[0, \infty)$. Let $(\nu_t)_{t \in [0,T]}, (\nu'_t)_{t \in [0,T]} \in \mathcal{P}_1(\mathbb{R}^d)$. Then, using the calculations from Lemma C.3, we have that

$$\mathcal{W}_1(\nu_t, \bar{\nu}_t) \leq \alpha L_{\Psi_\sigma, U^\xi} \int_0^t \mathcal{W}_1(\nu_s, \bar{\nu}_s) \mathrm{d}s,$$

which shows that $(\nu_t)_{t \in [0,T]}$ does not blow up in any finite time, and therefore we can extend $(\nu_t)_{t \in [0,T]}$ globally to $(\nu_t)_{t \in [0,\infty)}$. By definition, $\Psi_\sigma[\nu]$ admits a density of the form

$$\Psi_\sigma[\nu_t](x) = \frac{\exp\left(-\frac{1}{\sigma} \frac{\delta F}{\delta \nu}(\nu_t, x) - U^\xi(x)\right)}{\int_{\mathbb{R}^d} \exp\left(-\frac{1}{\sigma} \frac{\delta F}{\delta \nu}(\nu_t, x') - U^\xi(x')\right) \mathrm{d}x'}.$$

For any fixed $x \in \mathbb{R}^d$, the flat derivative $t \mapsto \frac{\delta F}{\delta \nu}(\nu_t, x)$ is continuous on $[0, \infty)$ due to the fact that $\nu_t \in C\left([0, \infty), \mathcal{P}_1(\mathbb{R}^d)\right)$ and $\nu \mapsto \frac{\delta F}{\delta \nu}(\nu, x)$ is continuous. Moreover, the flat derivative $\frac{\delta F}{\delta \nu}(\nu_t, x)$ is bounded for every $x \in \mathbb{R}^d$ and all $t \geq 0$ due to Assumption 2.1. Therefore, both terms $Z_\sigma(\nu_t)$ and $\exp\left(-\frac{1}{\sigma} \frac{\delta F}{\delta \nu}(\nu_t, x) - U^\xi(x)\right)$ are continuous in $t$ and bounded for every $x \in \mathbb{R}^d$ and every $t \geq 0$. Hence, we have that $\Psi_\sigma[\nu_t](x)$ is continuous in $t$ and bounded for every $x \in \mathbb{R}^d$. Since by assumption $\nu_0 \in \mathcal{P}_1(\mathbb{R}^d)$ admits a density $\nu_0(x)$, we define the density of $\nu_t$ by

$$\nu_t(x) := e^{-\alpha t} \nu_0(x) + \int_0^t \alpha e^{-\alpha(t-s)} \Psi_\sigma[\nu_s](x) \mathrm{d}s.$$

By definition $[0, \infty) \ni t \mapsto \nu_t(x)$ is continuous for every $x \in \mathbb{R}^d$. Since $s \mapsto e^{-\alpha(t-s)} \Psi_\sigma[\nu_s](x)$ is continuous and bounded in $s$ for every $t \geq 0$, it follows that $t \mapsto \nu_t \in C^1\left([0, \infty), \mathcal{P}_1^\lambda(\mathbb{R}^d)\right)$ and $\nu_t(x)$ satisfies (48). □

## C.2  Min-max problems

We recall a necessary condition for the optimality of MNEs of (6).

**Proposition C.4** (Necessary condition for optimality; [11, Theorem 3.1])**.** *Let Assumptions 3.1, 3.2 hold and let $\sigma_\nu, \sigma_\mu > 0$. If the pair $\left(\nu^*_{\sigma_\nu}, \mu^*_{\sigma_\mu}\right) \in \mathcal{P}_1(\mathbb{R}^d) \times \mathcal{P}_1(\mathbb{R}^d)$ is an MNE, then $\nu^*_{\sigma_\nu}, \mu^*_{\sigma_\mu}$ are equivalent to $\lambda$, and for $\lambda$-a.a. $(x, y) \in \mathbb{R}^d \times \mathbb{R}^d$,*

$$\nu^*_{\sigma_\nu}(\mathrm{d}x) = \frac{\exp\left(-\frac{1}{\sigma_\nu} \frac{\delta F}{\delta \nu}(\nu^*_{\sigma_\nu}, \mu^*_{\sigma_\mu}, x) - U^\xi(x)\right)}{\int_{\mathbb{R}^d} \exp\left(-\frac{1}{\sigma_\nu} \frac{\delta F}{\delta \nu}(\nu^*_{\sigma_\nu}, \mu^*_{\sigma_\mu}, x') - U^\xi(x')\right) \mathrm{d}x'} \mathrm{d}x,$$

$$\mu_{\sigma_\mu}^*(\mathrm{d}y) = \frac{\exp\left(\frac{1}{\sigma_\mu}\frac{\delta F}{\delta \mu}(\nu_{\sigma_\nu}^*, \mu_{\sigma_\mu}^*, y) - U^\rho(y)\right)}{\int_{\mathbb{R}^d} \exp\left(\frac{1}{\sigma_\mu}\frac{\delta F}{\delta \mu}(\nu_{\sigma_\nu}^*, \mu_{\sigma_\mu}^*, y') - U^\rho(y')\right)\mathrm{d}y'}\mathrm{d}y.$$

Now we give the proof of the existence and uniqueness of the flow (8), which is an extension of the proof of Proposition C.2. Note that the proof closely follows the argument of [23, Theorem 1], with the key difference being that we use the Wasserstein distance in place of the Total Variation distance. We include the proof here for completeness.

**Proposition C.5** (Existence and uniqueness of the flow). *Let Assumptions 3.1, 3.2 hold and let $(\nu_0, \mu_0) \in \mathcal{P}_1(\mathbb{R}^d) \times \mathcal{P}_1(\mathbb{R}^d)$. Then there exists a unique solution $(\nu_t, \mu_t)_{t\geq 0} \in C\left([0,\infty]; \mathcal{P}_1(\mathbb{R}^d) \times \mathcal{P}_1(\mathbb{R}^d)\right)$ to (8) and the solution depends continuously on the initial condition. Moreover, if $(\nu_0, \mu_0) \in \mathcal{P}_1^\lambda(\mathbb{R}^d) \times \mathcal{P}_1^\lambda(\mathbb{R}^d)$, then $(\nu_t, \mu_t)_{t\geq 0}$ admits the density $(\nu_t(\cdot), \mu_t(\cdot))_{t\geq 0} \subset \mathcal{P}_1^\lambda(\mathbb{R}^d) \times \mathcal{P}_1^\lambda(\mathbb{R}^d)$ such that, for every $x, y \in \mathbb{R}^d$, it holds that $t \mapsto \nu_t \in C^1\left([0,\infty), \mathcal{P}_1^\lambda(\mathbb{R}^d)\right), t \mapsto \mu_t \in C^1\left([0,\infty), \mathcal{P}_1^\lambda(\mathbb{R}^d)\right)$ and*

$$\begin{cases} \mathrm{d}\nu_t(x) = \alpha_\nu\left(\Psi_{\sigma_\nu}[\nu_t, \mu_t](x) - \nu_t(x)\right)\mathrm{d}t, \\ \mathrm{d}\mu_t(y) = \alpha_\mu\left(\Phi_{\sigma_\mu}[\nu_t, \mu_t](y) - \mu_t(y)\right)\mathrm{d}t, \quad t > 0, \end{cases} \tag{51}$$

*for some initial condition $(\nu_0(x), \mu_0(y)) \in \mathcal{P}_1^\lambda(\mathbb{R}^d) \times \mathcal{P}_1^\lambda(\mathbb{R}^d)$.*

*Proof of Theorem C.5. Step 1: Existence of flow on $[0, T]$.* We will define a Picard iteration scheme as follows. Fix $T > 0$ and for each $n \geq 1$, fix $\nu_0^{(n)} = \nu_0^{(0)} = \nu_0 \in \mathcal{P}_1(\mathbb{R}^d)$ and

$$\mu_0^{(n)} = \mu_0^{(0)} = \mu_0 \in \mathcal{P}_1(\mathbb{R}^d).$$

Then define the flow $(\nu_t^{(n)}, \mu_t^{(n)})_{t\in[0,T]}$ by

$$\nu_t^{(n)} := e^{-\alpha_\nu t}\nu_0 + \int_0^t \alpha_\nu e^{-\alpha_\nu(t-s)}\Psi_{\sigma_\nu}\left[\nu_s^{(n-1)}, \mu_s^{(n-1)}\right]\mathrm{d}s, \quad t \in [0, T], \tag{52}$$

$$\mu_t^{(n)} := e^{-\alpha_\mu t}\mu_0 + \int_0^t \alpha_\mu e^{-\alpha_\mu(t-s)}\Phi_{\sigma_\mu}\left[\nu_s^{(n-1)}, \mu_s^{(n-1)}\right]\mathrm{d}s, \quad t \in [0, T]. \tag{53}$$

For fixed $T > 0$, we consider the sequence of flows $\left((\nu_t^{(n)}, \mu_t^{(n)})_{t\in[0,T]}\right)_{n=0}^\infty$ in $\left(\mathcal{P}_1(\mathbb{R}^d)^{[0,T]} \times \mathcal{P}_1(\mathbb{R}^d)^{[0,T]}, \mathcal{W}_1^{[0,T]}\right)$, where, for any $(\nu_t, \mu_t)_{t\in[0,T]} \in \mathcal{P}_1(\mathbb{R}^d)^{[0,T]} \times \mathcal{P}_1(\mathbb{R}^d)^{[0,T]}$, the distance $\mathcal{W}_1^{[0,T]}$ is defined by

$$\mathcal{W}_1^{[0,T]}\left((\nu_t, \mu_t)_{t\in[0,T]}, (\nu_t', \mu_t')_{t\in[0,T]}\right) := \int_0^T \mathcal{W}_1(\nu_t, \nu_t')\mathrm{d}t + \int_0^T \mathcal{W}_1(\mu_t, \mu_t')\mathrm{d}t.$$

Since $\left(\mathcal{P}_1(\mathbb{R}^d), \mathcal{W}_1\right)$ is a complete metric space, we can apply the argument from [41, Lemma A.5] with $p = 1$ to conclude that $\left(\mathcal{P}_1(\mathbb{R}^d)^{[0,T]}, \int_0^T \mathcal{W}_1(\nu_t, \nu_t')\mathrm{d}t\right)$ and $\left(\mathcal{P}_1(\mathbb{R}^d)^{[0,T]}, \int_0^T \mathcal{W}_1(\mu_t, \mu_t')\mathrm{d}t\right)$ are a complete metric spaces, and hence so is $\left(\mathcal{P}_1(\mathbb{R}^d)^{[0,T]} \times \mathcal{P}_1(\mathbb{R}^d)^{[0,T]}, \mathcal{W}_1^{[0,T]}\right)$.

We consider the Picard iteration mapping $\varphi\left((\nu_t^{(n-1)}, \mu_t^{(n-1)})_{t\in[0,T]}\right) := (\nu_t^{(n)}, \mu_t^{(n)})_{t\in[0,T]}$ defined via (52) and (53) and show that $\varphi$ admits a unique fixed point $(\nu_t, \mu_t)_{t\in[0,T]}$ in the complete space $\left(\mathcal{P}_1(\mathbb{R}^d)^{[0,T]} \times \mathcal{P}_1(\mathbb{R}^d)^{[0,T]}, \mathcal{W}_1^{[0,T]}\right)$. Then this fixed point is the solution to (8).

**Lemma C.6.** *The mapping $\varphi\left((\nu_t^{(n-1)}, \mu_t^{(n-1)})_{t\in[0,T]}\right) := (\nu_t^{(n)}, \mu_t^{(n)})_{t\in[0,T]}$ defined via (52) and (53) admits a unique fixed point in $\left(\mathcal{P}_1(\mathbb{R}^d)^{[0,T]} \times \mathcal{P}_1(\mathbb{R}^d)^{[0,T]}, \mathcal{W}_1^{[0,T]}\right)$.*

*Proof of Lemma C.6.* **Step 1:** *The sequence of flows* $\left( (\nu_t^{(n)}, \mu_t^{(n)})_{t \in [0,T]} \right)_{n=0}^{\infty}$ *is a Cauchy sequence in* $\left( \mathcal{P}_1(\mathbb{R}^d)^{[0,T]} \times \mathcal{P}_1(\mathbb{R}^d)^{[0,T]}, \mathcal{W}_1^{[0,T]} \right).$

Let $\psi \in \mathrm{Lip}_1(\mathbb{R}^d)$. Then we have that

$$
\int_{\mathbb{R}^d} \psi(x) \left( \nu_t^{(n)} - \nu_t^{(n-1)} \right)(\mathrm{d}x)
$$

$$
= \int_{\mathbb{R}^d} \psi(x) \left( \int_0^t e^{-\alpha_\nu(t-s)} \Psi_{\sigma_\nu} \left[ \nu_s^{(n-1)}, \mu_s^{(n-1)} \right] \mathrm{d}s \right.
$$

$$
\left. - \int_0^t e^{-\alpha_\nu(t-s)} \Psi_{\sigma_\nu} \left[ \nu_s^{(n-2)}, \mu_s^{(n-2)} \right] \mathrm{d}s \right)(\mathrm{d}x)
$$

$$
= \int_{\mathbb{R}^d} \psi(x) \left( \int_0^t e^{-\alpha_\nu(t-s)} \left( \Psi_{\sigma_\nu} \left[ \nu_s^{(n-1)}, \mu_s^{(n-1)} \right] - \Psi_{\sigma_\nu} \left[ \nu_s^{(n-2)}, \mu_s^{(n-2)} \right] \right) \mathrm{d}s \right)(\mathrm{d}x) \qquad (54)
$$

$$
= \int_0^t e^{-\alpha_\nu(t-s)} \int_{\mathbb{R}^d} \psi(x) \left( \Psi_{\sigma_\nu} \left[ \nu_s^{(n-1)}, \mu_s^{(n-1)} \right] - \Psi_{\sigma_\nu} \left[ \nu_s^{(n-2)}, \mu_s^{(n-2)} \right] \right)(\mathrm{d}x)\mathrm{d}s
$$

$$
\leq \int_0^t e^{-\alpha_\nu(t-s)} \mathcal{W}_1 \left( \Psi_{\sigma_\nu} \left[ \nu_s^{(n-1)}, \mu_s^{(n-1)} \right], \Psi_{\sigma_\nu} \left[ \nu_s^{(n-2)}, \nu_s^{(n-2)} \right] \right) \mathrm{d}s
$$

$$
\leq L_{\Psi_{\sigma_\nu}, U^\xi} \int_0^t \left( \mathcal{W}_1 \left( \nu_s^{(n-1)}, \nu_s^{(n-2)} \right) + \mathcal{W}_1 \left( \mu_s^{(n-1)}, \mu_s^{(n-2)} \right) \right) \mathrm{d}s,
$$

where in the third equality we used Fubini's theorem, in the first inequality we used the definition of $\mathcal{W}_1$, and in the last inequality we used (39) and the fact that $e^{-\alpha(t-s)} \leq 1$ for $s \in [0, t]$.

A similar argument using (40) yields

$$
\int_{\mathbb{R}^d} \psi(x) \left( \mu_t^{(n)} - \mu_t^{(n-1)} \right)(\mathrm{d}x)
$$

$$
\leq L_{\Phi_{\sigma_\mu}, U^\rho} \int_0^t \left( \mathcal{W}_1 \left( \nu_s^{(n-1)}, \nu_s^{(n-2)} \right) + \mathcal{W}_1 \left( \mu_s^{(n-1)}, \mu_s^{(n-2)} \right) \right) \mathrm{d}s. \qquad (55)
$$

Taking supremum over $\psi$ in (54) and (55) and adding both together gives

$$
\mathcal{W}_1 \left( \nu_t^{(n)}, \nu_t^{(n-1)} \right) + \mathcal{W}_1 \left( \mu_t^{(n)}, \mu_t^{(n-1)} \right)
$$

$$
\leq \left( L_{\Psi_{\sigma_\nu}, U^\xi} + L_{\Phi_{\sigma_\mu}, U^\rho} \right) \int_0^t \left( \mathcal{W}_1 \left( \nu_s^{(n-1)}, \nu_s^{(n-2)} \right) + \mathcal{W}_1 \left( \mu_s^{(n-1)}, \mu_s^{(n-2)} \right) \right) \mathrm{d}s
$$

$$
\leq \left( L_{\Psi_{\sigma_\nu}, U^\xi} + L_{\Phi_{\sigma_\mu}, U^\rho} \right)^{n-1} \times
$$

$$
\times \int_0^t \int_0^{t_1} \cdots \int_0^{t_{n-2}} \left( \mathcal{W}_1 \left( \nu_{t_{n-1}}^{(1)}, \nu_{t_{n-1}}^{(0)} \right) + \mathcal{W}_1 \left( \mu_{t_{n-1}}^{(1)}, \mu_{t_{n-1}}^{(0)} \right) \right) \mathrm{d}t_{n-1} \ldots \mathrm{d}t_2 \mathrm{d}t_1
$$

$$
\leq \left( L_{\Psi_{\sigma_\nu}, U^\xi} + L_{\Phi_{\sigma_\mu}, U^\rho} \right)^{n-1} \frac{t^{n-2}}{(n-2)!} \int_0^t \left( \mathcal{W}_1 \left( \nu_{t_{n-1}}^{(1)}, \nu_{t_{n-1}}^{(0)} \right) + \mathcal{W}_1 \left( \mu_{t_{n-1}}^{(1)}, \mu_{t_{n-1}}^{(0)} \right) \right) \mathrm{d}t_{n-1},
$$

where in the third inequality we used the bound $\int_0^{t_{n-2}} \mathrm{d}t_{n-1} \leq \int_0^t \mathrm{d}t_{n-1}$. Hence, we obtain

$$
\int_0^T \left( \mathcal{W}_1 \left( \nu_t^{(n)}, \nu_t^{(n-1)} \right) + \mathcal{W}_1 \left( \mu_t^{(n)}, \mu_t^{(n-1)} \right) \right) \mathrm{d}t
$$

$$
\leq \left( L_{\Psi_{\sigma_\nu}, U^\xi} + L_{\Phi_{\sigma_\mu}, U^\rho} \right)^{n-1} \frac{T^{n-1}}{(n-1)!} \int_0^T \left( \mathcal{W}_1 \left( \nu_{t_{n-1}}^{(1)}, \nu_{t_{n-1}}^{(0)} \right) + \mathcal{W}_1 \left( \mu_{t_{n-1}}^{(1)}, \mu_{t_{n-1}}^{(0)} \right) \right) \mathrm{d}t_{n-1}.
$$

Using the definition of $\mathcal{W}_1^{[0,T]}$, the last inequality becomes

$$\mathcal{W}_1^{[0,T]}\left((\nu_t^{(n)},\mu_t^{(n)})_{t\in[0,T]},(\nu_t^{(n-1)},\mu_t^{(n-1)})_{t\in[0,T]}\right)$$

$$\leq \left(L_{\Psi_{\sigma_\nu},U^\xi}+L_{\Phi_{\sigma_\mu},U^\rho}\right)^{n-1}\frac{T^{n-1}}{(n-1)!}\mathcal{W}_1^{[0,T]}\left((\nu_t^{(1)},\mu_t^{(1)})_{t\in[0,T]},(\nu_t^{(0)},\mu_t^{(0)})_{t\in[0,T]}\right).$$

By choosing $n$ sufficiently large, we conclude that $\left((\nu_t^{(n)},\mu_t^{(n)})_{t\in[0,T]}\right)_{n=0}^\infty$ is a Cauchy sequence. By completeness of $\left(\mathcal{P}_1(\mathbb{R}^d)^{[0,T]}\times\mathcal{P}_1(\mathbb{R}^d)^{[0,T]},\mathcal{W}_1^{[0,T]}\right)$, the sequence admits a limit point $(\nu_t,\mu_t)_{t\in[0,T]}\in\left(\mathcal{P}_1(\mathbb{R}^d)^{[0,T]}\times\mathcal{P}_1(\mathbb{R}^d)^{[0,T]},\mathcal{W}_1^{[0,T]}\right)$.

*Step 2: The limit point $(\nu_t,\mu_t)_{t\in[0,T]}$ is a fixed point of $\varphi$.* From Step 1, we obtain that for Lebesgue-almost all $t\in[0,T]$ we have

$$\mathcal{W}_1(\nu_t^{(n)},\nu_t)\to 0,\quad \mathcal{W}_1(\mu_t^{(n)},\mu_t)\to 0,\ \text{ as } n\to\infty.$$

Therefore, by (39) and (40), for Lebesgue-almost all $t\in[0,T]$, we have that

$$\mathcal{W}_1\left(\Psi_{\sigma_\nu}\left[\nu_t^{(n)},\mu_t^{(n)}\right],\Psi_{\sigma_\nu}\left[\nu_t,\mu_t\right]\right)\to 0,\ \mathcal{W}_1\left(\Phi_{\sigma_\mu}\left[\nu_t^{(n)},\mu_t^{(n)}\right],\Phi_{\sigma_\mu}\left[\nu_t,\mu_t\right]\right)\to 0,\ \text{ as } n\to\infty.$$

Hence, letting $n\to\infty$ in (52) and in (53) and using the dominated convergence theorem (which is possible since $\Psi_{\sigma_\nu},\Phi_{\sigma_\mu}$ are uniformly bounded in $n$ due to Assumption 3.1), we conclude that $(\nu_t,\mu_t)_{t\in[0,T]}$ is a fixed point of $\varphi$.

*Step 3: The fixed point $(\nu_t,\mu_t)_{t\in[0,T]}$ of $\varphi$ is unique.* Suppose, for the contrary, that $\varphi$ admits two fixed points $(\nu_t,\mu_t)_{t\in[0,T]}$ and $(\bar\nu_t,\bar\mu_t)_{t\in[0,T]}$ such that $\nu_0=\bar\nu_0$ and $\mu_0=\bar\mu_0$. Then repeating the same calculations from (54) and (55), we arrive at

$$\mathcal{W}_1(\nu_t,\bar\nu_t)+\mathcal{W}_1(\mu_t,\bar\mu_t)\leq\left(L_{\Psi_{\sigma_\nu},U^\xi}+L_{\Phi_{\sigma_\mu},U^\rho}\right)\int_0^t\left(\mathcal{W}_1(\nu_s,\bar\nu_s)+\mathcal{W}_1(\mu_s,\bar\mu_s)\right)\mathrm{d}s.$$

For each $t\in[0,T]$, denote $f(t):=\int_0^t\left(\mathcal{W}_1(\nu_s,\bar\nu_s)+\mathcal{W}_1(\mu_s,\bar\mu_s)\right)\mathrm{d}s$. Observe that $f\geq 0$ and $f(0)=0$. Then, by Gronwall's lemma, we obtain

$$0\leq f(t)\leq e^{\left(L_{\Psi_{\sigma_\nu},U^\xi}+L_{\Phi_{\sigma_\mu},U^\rho}\right)t}f(0)=0,$$

and hence

$$\mathcal{W}_1(\nu_t,\bar\nu_t)+\mathcal{W}_1(\mu_t,\bar\mu_t)=0,$$

for Lebesgue-almost all $t\in[0,T]$, which implies

$$\nu_t=\bar\nu_t,\quad \mu_t=\bar\mu_t,$$

for Lebesgue-almost all $t\in[0,T]$. Therefore, the fixed point $(\nu_t,\mu_t)_{t\in[0,T]}$ of $\varphi$ must be unique.

From *Steps 1, 2 and 3*, we obtain the existence and uniqueness of a flow $(\nu_t,\mu_t)_{t\in[0,T]}$ satisfying (8) for any $T>0$. $\qquad\square$

Having proved Lemma C.6, we return to the proof of Theorem C.5.

*Step 2: Existence of the flow on $[0,\infty)$.*

From Lemma C.6, for any $T>0$, there exists unique flow $(\nu_t,\mu_t)_{t\in[0,T]}$ satisfying (8). It remains to prove that the existence of this flow could be extended to $[0,\infty)$. Let $(\nu_t,\mu_t)_{t\in[0,T]},(\nu_t',\mu_t')_{t\in[0,T]}\in\mathcal{P}_1(\mathbb{R}^d)$. Then, using the calculations from Lemma C.6, we have that

$$\mathcal{W}_1(\nu_t,\bar\nu_t)+\mathcal{W}_1(\mu_t,\bar\mu_t)\leq\left(L_{\Psi_{\sigma_\nu},U^\xi}+L_{\Phi_{\sigma_\mu},U^\rho}\right)\int_0^t\left(\mathcal{W}_1(\nu_s,\bar\nu_s)+\mathcal{W}_1(\mu_s,\bar\mu_s)\right)\mathrm{d}s,$$

which shows that $(\nu_t,\mu_t)_{t\in[0,T]}$ does not blow up in any finite time, and therefore we can extend $(\nu_t,\mu_t)_{t\in[0,T]}$ globally to $(\nu_t,\mu_t)_{t\in[0,\infty)}$. By definition, $\Psi_{\sigma_\nu}[\nu,\mu]$ admits a density of the form

$$\Psi_{\sigma_\nu}[\nu_t,\mu_t](x)=\frac{\exp\left(-\frac{1}{\sigma_\nu}\frac{\delta F}{\delta\nu}(\nu_t,\mu_t,x)-U^\xi(x)\right)}{\int_{\mathbb{R}^d}\exp\left(-\frac{1}{\sigma_\nu}\frac{\delta F}{\delta\nu}(\nu_t,\mu_t,x')-U^\xi(x')\right)\mathrm{d}x'}.$$

For any fixed $x \in \mathbb{R}^d$, the flat derivative $t \mapsto \frac{\delta F}{\delta \nu}(\nu_t, \mu_t, x)$ is continuous on $[0, \infty)$ due to the fact that $\nu_t, \mu_t \in C\left([0, \infty), \mathcal{P}_1(\mathbb{R}^d)\right)$ and $(\nu, \mu) \mapsto \frac{\delta F}{\delta \nu}(\nu, \mu, x)$ is continuous. Moreover, the flat derivative $\frac{\delta F}{\delta \nu}(\nu_t, \mu_t, x)$ is bounded for every $x \in \mathbb{R}^d$ and all $t \geq 0$ due to Assumption 3.1. Therefore, both terms $Z_{\sigma_\nu}(\nu_t, \mu_t)$ and $\exp\left(-\frac{1}{\sigma_\nu} \frac{\delta F}{\delta \nu}(\nu_t, \mu_t, x) - U^\xi(x)\right)$ are continuous in $t$ and bounded for every $x \in \mathbb{R}^d$ and every $t \geq 0$. Hence, we have that $\Psi_{\sigma_\nu}[\nu_t, \mu_t](x)$ is continuous in $t$ and bounded for every $x \in \mathbb{R}^d$. Since by assumption $\nu_0 \in \mathcal{P}_1(\mathbb{R}^d)$ admits a density $\nu_0(x)$, we define the density of $\nu_t$ by

$$\nu_t(x) := e^{-\alpha_\nu t}\nu_0(x) + \int_0^t \alpha_\nu e^{-\alpha_\nu(t-s)} \Psi_{\sigma_\nu}[\nu_s, \mu_s](x)\mathrm{d}s.$$

By definition $[0, \infty) \ni t \mapsto \nu_t(x)$ is continuous for every $x \in \mathbb{R}^d$. Since $s \mapsto e^{-\alpha_\nu(t-s)}\Psi_{\sigma_\nu}[\nu_s, \mu_s](x)$ is continuous and bounded in $s$ for every $t \geq 0$, it follows that $t \mapsto \nu_t \in C^1\left([0, \infty), \mathcal{P}_1^\lambda(\mathbb{R}^d)\right)$ and $\nu_t(x)$ satisfies (51) The same argument gives that $\Phi_{\sigma_\mu}[\nu_t, \mu_t](y)$ is continuous in $t$ and bounded for every $y \in \mathbb{R}^d$, consequently that $\mu_t$ admits a density $\mu_t(y)$ that satisfies (51) and $t \mapsto \mu_t \in C^1\left([0, \infty), \mathcal{P}_1^\lambda(\mathbb{R}^d)\right)$. $\qquad\square$

# D  Notation for MDPs

In this section, we present standard notation for MDPs, following the conventions used in [24]. Let $(E, d)$ denote a Polish space, i.e., a complete separable metric space. We endow $E$ with its Borel $\sigma$-alebra $\mathcal{B}(E)$. Let $B_b(E)$ denote the space of bounded measurable functions $f : E \to \mathbb{R}$ endowed with the supremum norm $|f|_{B_b(E)} = \sup_{x \in E} |f(x)|$. Let $\mathcal{M}(E)$ denote the Banach space of finite signed measures $m$ on $E$ endowed with the total variation norm $|m|_{\mathcal{M}(E)} = |m|(E)$, where $|m|$ is the total-variation measure. We denote by $\mathcal{M}_+(E) \subset \mathcal{M}(E)$ the subset of finite positive measures. For a measure $m \in \mathcal{M}_+(E)$ and measurable function $f : E \to \mathbb{R}$, let

$$\operatorname*{ess\,sup}_{x \in E} f := \inf\{c \in \mathbb{R} : m\{x \in E : f(x) > c\} = 0\}.$$

For given Polish spaces $(E_1, d_1)$ and $(E_2, d_2)$, denote by $b\mathcal{K}(E_1|E_2)$ the Banach space of bounded signed kernels $k : E_2 \to \mathcal{M}(E_1)$ endowed with the norm $|k|_{b\mathcal{K}(E_1|E_2)} = \sup_{x \in E_2} |k(x)|_{\mathcal{M}(E_1)}$; that is, $k(U|\cdot) : E_2 \to \mathbb{R}$ is measurable for all $U \in \mathcal{M}(E_1)$ and $k(\cdot|x) \in \mathcal{M}(E_1)$ for all $x \in E_2$. For a fixed reference measure $\eta \in \mathcal{M}_+(E_1)$, we denote by $b\mathcal{K}_\eta(E_1|E_2)$ the space of bounded kernels that are absolutely continuous with respect to $\eta$. Every kernel $k \in b\mathcal{K}(E_1|E_2)$ induces a bounded linear operator $T_k \in \mathcal{L}(\mathcal{M}(E_2), \mathcal{M}(E_1))$ defined by

$$T_k\eta(\mathrm{d}y) = \eta k(\mathrm{d}y) = \int_{E_2} \eta(\mathrm{d}x)k(\mathrm{d}y|x).$$

Moreover, we have

$$|k|_{b\mathcal{K}(E_1|E_2)} = \sup_{x \in E_2} \sup_{\substack{h \in B_b(E_1) \\ |h|_{B_b(E_1)} \leq 1}} \int_{E_1} h(y)k(\mathrm{d}y|x) = |T_k|_{\mathcal{L}(\mathcal{M}(E_2), \mathcal{M}(E_1))}, \tag{56}$$

where the latter is the operator norm. Thus, $b\mathcal{K}(E|E)$ is a Banach algebra with the product defined via composition of the corresponding linear operators. In particular, for given $k \in b\mathcal{K}(E|E)$,

$$T_k^n\mu(\mathrm{d}y) = \mu k^n(\mathrm{d}y) = \int_{E^n} \mu(\mathrm{d}x_0)k(\mathrm{d}x_1|x_0)\cdots k(\mathrm{d}x_{n-1}|x_{n-2})k(\mathrm{d}y|x_{n-1}). \tag{57}$$

Let $\mathcal{P}(E_1|E_2)$ denote the convex subspace of $b\mathcal{K}(E_1|E_2)$ such that $P(\cdot|x) \in \mathcal{P}(E_1)$ for all $x \in E_2$.

We denote by $\left((S \times A)^{\mathbb{N}}, \mathcal{F}\right)$ a sample space, where the elements of $(S \times A)^{\mathbb{N}}$ are state-action pairs $(s_n, a_n)_{n=0}^\infty$ with $(s_n, a_n) \in S \times A$, for each $n \in \mathbb{N}$, and $\mathcal{F}$ is the associated $\sigma$-algebra. By [3, Proposition 7.28], for a given initial distribution $\gamma \in \mathcal{P}(S)$ and policy $\pi \in \mathcal{P}(A|S)$, there exists a unique product probability measure $\mathbb{P}_\gamma^\pi$ on $\left((S \times A)^{\mathbb{N}}, \mathcal{F}\right)$ such that for every $n \in \mathbb{N}$, we have

1. $\mathbb{P}_\gamma^\pi(s_0 \in \mathcal{S}) = \gamma(\mathcal{S})$,
2. $\mathbb{P}^\pi(a_n \in \mathcal{A}|(s_0, a_0, \ldots, s_n)) = \pi(a_n|s_n)$,

3. $\mathbb{P}^\pi_\gamma(s_{n+1} \in \mathcal{S} | (s_0, a_0, \ldots, s_n, a_n)) = P(\mathcal{S}|s_n, a_n)$,

for all $\mathcal{S} \in \mathcal{B}(S)$ and $\mathcal{A} \in \mathcal{B}(A)$. Thus, $\{s_n\}_{n \geq 0}$ is a Markov chain with transition kernel $P_\pi \in \mathcal{P}(S|S)$ defined by

$$P_\pi(\mathrm{d}s'|s) := \int_A P(\mathrm{d}s'|s, a')\pi(\mathrm{d}a'|s). \tag{58}$$

The expectation corresponding to $\mathbb{P}^\pi_\gamma$ is denoted by $\mathbb{E}^\pi_\gamma$. For given $s \in S$, we denote $\mathbb{E}^\pi_s := \mathbb{E}^\pi_{\delta_s}$, where $\delta_s \in \mathcal{P}(S)$ denotes the Dirac measure at $s \in S$.

# E  Auxiliary results for MDPs

In this section, we present two sets of auxiliary results, each corresponding to Section 2 and Section 3, respectively.

## E.1  Single-agent optimization

First, we prove that the value function $V_\tau$ in the MDP setting of Section 2.2 satisfies Assumption 2.1. Our proof closely follows the argument in [24, Theorem 2.4], but we include it here for completeness.

Before we give the proof, we need to derive an alternative expression of $V_\tau$ to (2.2). Again, for full details on notation, we refer to Appendix D. By the Bellman principle (see, e.g., [20, Lemma B.2]), for all $\pi \in \mathcal{P}^\eta(A|S)$ and $s \in S$, it follows that

$$V_\tau^\pi(s) = \int_A \left( Q_\tau^\pi(s, a) + \tau \log \frac{\mathrm{d}\pi}{\mathrm{d}\eta}(a|s) \right) \pi(\mathrm{d}a|s). \tag{59}$$

For a given policy $\pi \in \mathcal{P}^\eta(A|S)$, the occupancy kernel $d^\pi \in \mathcal{P}(S|S)$ is defined by

$$d^\pi(\mathrm{d}s'|s) = (1 - \delta) \sum_{n=0}^\infty \delta^n P_\pi^n(\mathrm{d}s'|s), \tag{60}$$

where $P_\pi^0(\mathrm{d}s'|s) := \delta_s(\mathrm{d}s')$, for the Dirac measure $\delta_s$ at $s \in S$, $P_\pi^n$ is a product of kernels in the sense of (57), and the convergence of the series is understood in $b\mathcal{K}(S|S)$. For a given initial distribution $\gamma \in \mathcal{P}(S)$, the state occupancy measure $d_\gamma^\pi \in \mathcal{P}(S)$ is defined by

$$d_\gamma^\pi(\mathrm{d}s') := \int_S d^\pi(\mathrm{d}s'|s)\gamma(\mathrm{d}s).$$

Using (12) and (58) in (59) gives for all $s \in S$ that

$$V_\tau^\pi(s) = \int_A \left( c(s, a) + \tau \log \frac{\mathrm{d}\pi}{\mathrm{d}\eta}(a|s) \right) \pi(\mathrm{d}a|s) + \delta \int_S V_\tau^\pi(s') P_\pi(\mathrm{d}s'|s).$$

Applying this identity recursively and using (60) yields for all $s \in S$ that

$$V_\tau^\pi(s) = \frac{1}{1 - \delta} \int_S \int_A \left[ c(s', a) + \tau \log \frac{\mathrm{d}\pi}{\mathrm{d}\eta}(a|s') \right] \pi(\mathrm{d}a|s') d^\pi(\mathrm{d}s'|s).$$

Finally, integrating over $\gamma \in \mathcal{P}(S)$ leads to

$$V_\tau^\pi(\gamma) = \frac{1}{1 - \delta} \int_S \int_A \left[ c(s', a) + \tau \log \frac{\mathrm{d}\pi}{\mathrm{d}\eta}(a|s') \right] \pi(\mathrm{d}a|s') d_\gamma^\pi(\mathrm{d}s'). \tag{61}$$

Now, we introduce the following necessary notation for the proof. For $k \in \{0, 1\}$, let $\mathfrak{C}^k$ be the set of jointly measurable functions $h : \mathbb{R}^d \times S \times A \to \mathbb{R}$ such that for Lebesgue-almost all $\theta \in \mathbb{R}^d$, all $s \in S$, and $\eta$-almost all $a \in A$, $h$ is $k$-times differentiable in $\theta$, and satisfies

$$|h|_{\mathfrak{C}^k} := \max_{k \in \{0,1\}} \operatorname*{ess\,sup}_{a \in A} \sup_{s \in S} \operatorname*{ess\,sup}_{\theta \in \mathbb{R}^d} |\nabla_\theta^k h(\theta, s, a)| < \infty,$$

where the essential supremum over $A$ is defined relative to the reference measure $\eta$ and the essential supremum over $\mathbb{R}^d$ is defined relative to the Lebesgue measure.

For each $\nu \in \mathcal{P}_1(\mathbb{R}^d)$ and $\gamma \in \mathcal{P}(S)$ we define the maps

$$\mathcal{P}_1(\mathbb{R}^d) \ni \nu \mapsto \Pi(\nu)(\mathrm{d}a|s) := \pi_\nu(\mathrm{d}a|s) \in \mathcal{P}_\eta(A|S),$$

and

$$\mathcal{P}_1(\mathbb{R}^d) \ni \nu \mapsto \hat{V}_\tau(\nu)(\gamma) := V_\tau^{\pi_\nu}(\gamma) \in \mathbb{R},$$

where $\pi_\nu(\mathrm{d}a|s)$ and $\bar{V}_\tau(\nu)(\gamma)$ are given by (13) and (61), respectively.

The proof will depend on the following key lemmas, which we state without proof, as they are available in [24]. The first is Lemma [24, Lemma 2.2].

**Lemma E.1** (Flat derivative of $\Pi$)**.** *If $f \in \mathfrak{C}^0$, then the function $\Pi : \mathcal{P}_1(\mathbb{R}^d) \to \mathcal{P}_\eta(A|S)$ has a flat derivative $\frac{\delta\Pi}{\delta\nu} : \mathcal{P}_1(\mathbb{R}^d) \times \mathbb{R}^d \to b\mathcal{K}_\eta(A|S)$ given by*

$$\frac{\delta\Pi}{\delta\nu}(\nu, \theta)(\mathrm{d}a|s) = \left( f(\theta, s, a) - \int_A f(\theta, s, a')\Pi(\nu)(\mathrm{d}a'|s) \right) \Pi(\nu)(\mathrm{d}a|s). \tag{62}$$

The second is [24, Lemma 2.3], which can be viewed as a policy gradient theorem.

**Lemma E.2** (Flat derivative of $\hat{V}_\tau$)**.** *If $f \in \mathfrak{C}^1$, then the function $\hat{V}_\tau : \mathcal{P}_1(\mathbb{R}^d) \to \mathbb{R}$ has a flat derivative $\frac{\delta\hat{V}_\tau}{\delta\nu} : \mathcal{P}_1(\mathbb{R}^d) \times \mathbb{R}^d \to \mathbb{R}$ given by*

$$\frac{\delta\hat{V}_\tau}{\delta\nu}(\nu, \theta)(\gamma) = \frac{1}{1-\delta} \int_S \int_A \left( Q_\tau^{\Pi(\nu)}(s, a) + \tau \log \frac{\mathrm{d}\Pi(\nu)}{\mathrm{d}\eta}(a|s) \right) \frac{\delta\Pi}{\delta\nu}(\nu, \theta)(\mathrm{d}a|s) d_\gamma^{\Pi(\nu)}(\mathrm{d}s). \tag{63}$$

We are ready to prove that $\hat{V}_\tau$ satisfies Assumption 2.1.

**Proposition E.3.** *Let $S$ and $A$ be Polish spaces, $c \in B_b(S \times A)$, $\eta \in \mathcal{M}_+(A)$, and $\tau > 0$. Then for all $\gamma \in \mathcal{P}(S)$ the function $F(\cdot) := \hat{V}_\tau(\cdot)(\gamma)$ satisfies Assumption 2.1 with*

$$C_F := \frac{2}{(1-\delta)^2} \left( |c|_{B_b(S \times A)} + \tau \left( 2|f|_{\mathfrak{C}^0} + |\log \eta(A)| \right) \right) |f|_{\mathfrak{C}^0},$$

$$L_F := |f|_{\mathfrak{C}^1} \left( \frac{1}{(1-\delta)^2} \left( |c|_{B_b(S \times A)} + \tau \left( 2|f|_{\mathfrak{C}^0} + |\log \eta(A)| \right) \right) \max\left\{ 2, \frac{5}{1-\delta}|f|_{\mathfrak{C}^0} \right\} + 4\tau|f|_{\mathfrak{C}^0} \right).$$

*Proof.* For given $(s, a) \in S \times A$ and $\nu \in \mathcal{P}_1(\mathbb{R}^d)$, let

$$\bar{Q}_\tau^{\Pi(\nu)}(s, a) := \frac{1}{1-\delta} \left( Q_\tau^{\Pi(\nu)}(s, a) + \tau \log \frac{\mathrm{d}\Pi(\nu)}{\mathrm{d}\eta}(a|s) \right).$$

Since $c \in B_b(S \times A)$, $f \in \mathfrak{C}^0$ and $\eta \in \mathcal{M}_+(A)$, it follows by [24, Lemma A.4] that

$$\left| \bar{Q}_\tau^{\Pi(\nu)}(s, a) \right| \leq \frac{1}{(1-\delta)^2} \left( |c|_{B_b(S \times A)} + \tau \left( 2|f|_{\mathfrak{C}^0} + |\log \eta(A)| \right) \right), \tag{64}$$

for all $\nu \in \mathcal{P}_1(\mathbb{R}^d)$, $s \in S$ and $a \in A$ $\eta$−a.e. Therefore, by Lemma E.2, we obtain

$$\left| \frac{\delta\hat{V}_\tau}{\delta\nu}(\nu, \theta)(\gamma) \right|$$

$$\leq \int_S \int_A \left| \bar{Q}_\tau^{\Pi(\nu)}(s, a) \right| \left| f(\theta, s, a) - \int_A f(\theta, s, a')\Pi(\nu)(\mathrm{d}a'|s) \right| \Pi(\nu)(\mathrm{d}a|s) d_\gamma^{\Pi(\nu)}(\mathrm{d}s)$$

$$\leq \frac{2}{(1-\delta)^2} \left( |c|_{B_b(S \times A)} + \tau \left( 2|f|_{\mathfrak{C}^0} + |\log \eta(A)| \right) \right) |f|_{\mathfrak{C}^0} =: C_F,$$

for all $\gamma \in \mathcal{P}(S)$ and $(\nu, \theta) \in \mathcal{P}_1(\mathbb{R}^d) \times \mathbb{R}^d$, which proves that $\frac{\delta\hat{V}_\tau}{\delta\nu}$ is uniformly bounded.

Next we show that, given $\gamma \in \mathcal{P}(S)$, $(\nu, \theta) \mapsto \frac{\delta \hat{V}_\tau}{\delta \nu}(\nu, \theta)(\gamma)$ is Lipschitz. For any $(\nu', \theta'), (\nu, \theta) \in \mathcal{P}_1(\mathbb{R}^d) \times \mathbb{R}^d$, we have

$$
\left| \frac{\delta \hat{V}_\tau}{\delta \nu}(\nu', \theta')(\gamma) - \frac{\delta \hat{V}_\tau}{\delta \nu}(\nu, \theta)(\gamma) \right|
$$
$$
\leq \left| \frac{\delta \hat{V}_\tau}{\delta \nu}(\nu', \theta')(\gamma) - \frac{\delta \hat{V}_\tau}{\delta \nu}(\nu', \theta)(\gamma) \right| + \left| \frac{\delta \hat{V}_\tau}{\delta \nu}(\nu', \theta)(\gamma) - \frac{\delta \hat{V}_\tau}{\delta \nu}(\nu, \theta)(\gamma) \right|.
\tag{65}
$$

Note that since $f \in \mathfrak{C}^1$, we have

$$
|f(\theta', s, a) - f(\theta, s, a)| \leq |f|_{\mathfrak{C}^1} |\theta' - \theta|,
$$

for all $\theta \in \mathbb{R}^d$, $s \in S$ and $a \in A$ $\eta$−a.e. Then, for the first term in (65), we obtain

$$
\left| \frac{\delta \hat{V}_\tau}{\delta \nu}(\nu', \theta')(\gamma) - \frac{\delta \hat{V}_\tau}{\delta \nu}(\nu', \theta)(\gamma) \right|
$$
$$
= \left| \int_S \int_A \bar{Q}_\tau^{\Pi(\nu')}(s, a) \left( \frac{\delta \Pi}{\delta \nu}(\nu', \theta')(\mathrm{d}a|s) - \frac{\delta \Pi}{\delta \nu}(\nu', \theta)(\mathrm{d}a|s) \right) d_\gamma^{\Pi(\nu')}(\mathrm{d}s) \right|
$$
$$
= \left| \int_S \int_A \bar{Q}_\tau^{\Pi(\nu')}(s, a) \bigg( f(\theta', s, a) - f(\theta, s, a) \right.
$$
$$
\left. - \int_A (f(\theta', s, a') - f(\theta, s, a')) \, \Pi(\nu')(\mathrm{d}a'|s) \bigg) \Pi(\nu')(\mathrm{d}a|s) d_\gamma^{\Pi(\nu')}(\mathrm{d}s) \right|
\tag{66}
$$
$$
\leq \int_S \int_A |\bar{Q}_\tau^{\Pi(\nu')}(s, a)| \bigg( |f(\theta', s, a) - f(\theta, s, a)|
$$
$$
+ \int_A |f(\theta', s, a') - f(\theta, s, a')| \, \Pi(\nu')(\mathrm{d}a'|s) \bigg) \Pi(\nu')(\mathrm{d}a|s) d_\gamma^{\Pi(\nu')}(\mathrm{d}s)
$$
$$
\leq \frac{2|f|_{\mathfrak{C}^1}}{(1-\delta)^2} \left( |c|_{B_b(S \times A)} + \tau \left( 2|f|_{\mathfrak{C}^0} + |\log \eta(A)| \right) \right) |\theta' - \theta|.
$$

for all $\nu' \in \mathcal{P}_1(\mathbb{R}^d)$ and all $\theta', \theta \in \mathbb{R}^d$.

For the second term in (65), for $\theta \in \mathbb{R}^d$ and $\nu, \nu' \in \mathcal{P}_1(\mathbb{R}^d)$, we have

$$
\frac{\delta \hat{V}_\tau}{\delta \nu}(\nu', \theta)(\gamma) - \frac{\delta \hat{V}_\tau}{\delta \nu}(\nu, \theta)(\gamma)
$$
$$
= \underbrace{\int_S \int_A \bar{Q}_\tau^{\Pi(\nu')}(s, a) \frac{\delta \Pi}{\delta \nu}(\nu', \theta)(\mathrm{d}a|s) [d_\gamma^{\Pi(\nu')} - d_\gamma^{\Pi(\nu)}](\mathrm{d}s)}_{:= I_1}
$$
$$
+ \underbrace{\int_S \int_A \left[ \bar{Q}_\tau^{\Pi(\nu')}(s, a) - \bar{Q}_\tau^{\Pi(\nu)}(s, a) \right] \frac{\delta \Pi}{\delta \nu}(\nu', \theta)(\mathrm{d}a|s) d_\gamma^{\Pi(\nu)}(\mathrm{d}s)}_{:= I_2}
$$
$$
+ \int_S \int_A \bar{Q}_\tau^{\Pi(\nu)}(s, a) \left[ \frac{\delta \Pi}{\delta \nu}(\nu', \theta)(\mathrm{d}a|s) - \frac{\delta \Pi}{\delta \nu}(\nu, \theta)(\mathrm{d}a|s) \right] d_\gamma^{\Pi(\nu)}(\mathrm{d}s)
$$
$$
= I_1 + I_2 + \underbrace{\int_S \int_A \bar{Q}_\tau^{\Pi(\nu)}(s, a) \int_A f(\theta, s, a') [\Pi(\nu) - \Pi(\nu')](\mathrm{d}a'|s) \Pi(\nu)(\mathrm{d}a|s) d_\gamma^{\Pi(\nu)}(\mathrm{d}s)}_{:= I_3}
$$
$$
+ \underbrace{\int_S \int_A \bar{Q}_\tau^{\Pi(\nu)}(s, a) \left( f(\theta, s, a) - \int_A f(\theta, s, a') \Pi(\nu')(\mathrm{d}a'|s) \right) [\Pi(\nu') - \Pi(\nu)](\mathrm{d}a|s) d_\gamma^{\Pi(\nu)}(\mathrm{d}s)}_{:= I_4}.
$$

By [24, Corollary A.5], we have for all $\nu', \nu \in \mathcal{P}_1(\mathbb{R}^d)$ that

$$|d^{\Pi(\nu')} - d^{\Pi(\nu)}|_{b\mathcal{K}(S|S)} \leq 2|f|_{\mathfrak{C}^1} \frac{\delta}{1-\delta} \mathcal{W}_1(\nu', \nu).$$

Hence, using (64), we obtain

$$|I_1| \leq \frac{4\delta}{(1-\delta)^3} \left( |c|_{B_b(S \times A)} + \tau \left( 2|f|_{\mathfrak{C}^0} + |\log \eta(A)| \right) \right) |f|_{\mathfrak{C}^0} |f|_{\mathfrak{C}^1} \mathcal{W}_1(\nu', \nu). \qquad (67)$$

By the proof of [24, Corollary A.5], we have for all $\nu', \nu \in \mathcal{P}_1(\mathbb{R}^d)$ that

$$|\Pi(\nu') - \Pi(\nu)|_{b\mathcal{K}(A|S)} \leq 2|f|_{\mathfrak{C}^1} \mathcal{W}_1(\nu', \nu). \qquad (68)$$

Hence, by the definition of the $b\mathcal{K}(A|S)-$norm (see (56)), for any measurable $h : \mathbb{R}^d \times S \times A \to \mathbb{R}$, we have

$$\int_A h(\theta, s, a)(\Pi(\nu') - \Pi(\nu))(\mathrm{d}a|s) \leq 2|f|_{\mathfrak{C}^1} |h|_{B_b(A)} \mathcal{W}_1(\nu', \nu).$$

Thus, using (68) and (64), we deduce

$$|I_3 + I_4| \leq \frac{4}{(1-\delta)^2} \left( |c|_{B_b(S \times A)} + \tau \left( 2|f|_{\mathfrak{C}^0} + |\log \eta(A)| \right) \right) |f|_{\mathfrak{C}^0} |f|_{\mathfrak{C}^1} \mathcal{W}_1(\nu', \nu). \qquad (69)$$

To estimate $I_2$, applying Definition F.1 we observe that for all $(s, a) \in S \times A$, we have

$$\log \frac{\mathrm{d}\Pi(\nu')}{\mathrm{d}\eta}(a|s) - \log \frac{\mathrm{d}\Pi(\nu)}{\mathrm{d}\eta}(a|s)$$
$$= \int_0^1 \int_{\mathbb{R}^d} \left( f(\theta, s, a) - \int_A f(\theta, s, a') \Pi(\nu + \varepsilon(\nu' - \nu))(\mathrm{d}a'|s) \right) (\nu' - \nu)(\mathrm{d}\theta)\mathrm{d}\varepsilon$$
$$\leq 2|f|_{\mathfrak{C}^1} \mathcal{W}_1(\nu', \nu).$$

Now, taking $\gamma = \delta_s$ in $\hat{V}_\tau(\nu)(\gamma) = \int_S \hat{V}_\tau(\nu)(s)\gamma(\mathrm{d}s)$ gives

$$\hat{V}_\tau(\nu)(s) = \hat{V}_\tau(\nu)(\delta_s),$$

for all $s \in S$, and therefore

$$Q_\tau^{\Pi(\nu')}(s, a) - Q_\tau^{\Pi(\nu)}(s, a)$$
$$= \delta \int_S (\hat{V}_\tau(\nu')(s') - \hat{V}_\tau(\nu)(s'))P(\mathrm{d}s'|s, a)$$
$$= \delta \int_S \int_0^1 \int_{\mathbb{R}^d} \frac{\delta \hat{V}_\tau}{\delta \nu}(\nu + \varepsilon(\nu' - \nu), \theta)(\delta_{s'})(\nu' - \nu)(\mathrm{d}\theta)\mathrm{d}\varepsilon P(\mathrm{d}s'|s, a)$$
$$\leq \delta \mathcal{W}_1(\nu', \nu) \frac{2}{(1-\delta)^2} \left( |c|_{B_b(S \times A)} + \tau \left( 2|f|_{\mathfrak{C}^0} + |\log \eta(A)| \right) \right) |f|_{\mathfrak{C}^1} \int_S P(\mathrm{d}s'|s, a)$$
$$= \frac{2\delta}{(1-\delta)^2} \left( |c|_{B_b(S \times A)} + \tau \left( 2|f|_{\mathfrak{C}^0} + |\log \eta(A)| \right) \right) |f|_{\mathfrak{C}^1} \mathcal{W}_1(\nu', \nu),$$

where the first inequality follows from the fact that the map

$$\theta \mapsto \frac{\delta \hat{V}_\tau}{\delta \nu}(\nu + \varepsilon(\nu' - \nu), \theta)(\delta_{s'})$$

is Lipschitz due to (66) and the definition of $\mathcal{W}_1$ (see (9)), and last equality follows from the fact that $P \in \mathcal{P}(S|S \times A)$. Hence, we obtain

$$|I_2| \leq 4 \left( \frac{\delta}{(1-\delta)^2} \left( |c|_{B_b(S \times A)} + \tau \left( 2|f|_{\mathfrak{C}^0} + |\log \eta(A)| \right) \right) + \tau \right) |f|_{\mathfrak{C}^0} |f|_{\mathfrak{C}^1} \mathcal{W}_1(\nu', \nu). \qquad (70)$$

Putting together (67), (69) and (70) yields that

$$\left| \frac{\delta \hat{V}_\tau}{\delta \nu}(\nu', \theta)(\gamma) - \frac{\delta \hat{V}_\tau}{\delta \nu}(\nu, \theta)(\gamma) \right|$$

$$\leq \left[ \frac{4}{(1-\delta)^2} \left( \frac{\delta}{1-\delta} + 1 + \delta \right) \times \right.$$

$$\times \left. \left( |c|_{B_b(S \times A)} + \tau \left( 2|f|_{\mathfrak{C}^0} + |\log \eta(A)| \right) \right) + 4\tau \right] |f|_{\mathfrak{C}^0} |f|_{\mathfrak{C}^1} \mathcal{W}_1(\nu', \nu) \tag{71}$$

$$= \left[ \frac{4}{(1-\delta)^3} \left( -\left( \frac{1}{2} - \delta \right)^2 + \frac{5}{4} \right) \times \right.$$

$$\times \left. \left( |c|_{B_b(S \times A)} + \tau \left( 2|f|_{\mathfrak{C}^0} + |\log \eta(A)| \right) \right) + 4\tau \right] |f|_{\mathfrak{C}^0} |f|_{\mathfrak{C}^1} \mathcal{W}_1(\nu', \nu)$$

$$\leq \left( \frac{5}{(1-\delta)^3} \left( |c|_{B_b(S \times A)} + \tau \left( 2|f|_{\mathfrak{C}^0} + |\log \eta(A)| \right) \right) + 4\tau \right) |f|_{\mathfrak{C}^0} |f|_{\mathfrak{C}^1} \mathcal{W}_1(\nu', \nu).$$

for all $\nu', \nu \in \mathcal{P}_1(\mathbb{R}^d)$ and $\theta \in \mathbb{R}^d$. Combining (66) and (71) with (65) gives

$$\left| \frac{\delta \hat{V}_\tau}{\delta \nu}(\nu', \theta')(\gamma) - \frac{\delta \hat{V}_\tau}{\delta \nu}(\nu, \theta)(\gamma) \right|$$

$$\leq \frac{2|f|_{\mathfrak{C}^1}}{(1-\delta)^2} \left( |c|_{B_b(S \times A)} + \tau \left( 2|f|_{\mathfrak{C}^0} + |\log \eta(A)| \right) \right) |\theta' - \theta|$$

$$+ \left( \frac{5}{(1-\delta)^3} \left( |c|_{B_b(S \times A)} + \tau \left( 2|f|_{\mathfrak{C}^0} + |\log \eta(A)| \right) \right) + 4\tau \right) |f|_{\mathfrak{C}^0} |f|_{\mathfrak{C}^1} \mathcal{W}_1(\nu', \nu)$$

$$\leq |f|_{\mathfrak{C}^1} \max \left\{ \frac{2}{(1-\delta)^2} \left( |c|_{B_b(S \times A)} + \tau \left( 2|f|_{\mathfrak{C}^0} + |\log \eta(A)| \right) \right), \right.$$

$$\left. \left( \frac{5}{(1-\delta)^3} \left( |c|_{B_b(S \times A)} + \tau \left( 2|f|_{\mathfrak{C}^0} + |\log \eta(A)| \right) \right) + 4\tau \right) |f|_{\mathfrak{C}^0} \right\} \left( |\theta' - \theta| + \mathcal{W}_1(\nu', \nu) \right)$$

$$\leq |f|_{\mathfrak{C}^1} \left( \max \left\{ \frac{2}{(1-\delta)^2} \left( |c|_{B_b(S \times A)} + \tau \left( 2|f|_{\mathfrak{C}^0} + |\log \eta(A)| \right) \right), \right. \right.$$

$$\left. \frac{5}{(1-\delta)^3} \left( |c|_{B_b(S \times A)} + \tau \left( 2|f|_{\mathfrak{C}^0} + |\log \eta(A)| \right) \right) |f|_{\mathfrak{C}^0} \right\} + 4\tau |f|_{\mathfrak{C}^0} \right) \left( |\theta' - \theta| + \mathcal{W}_1(\nu', \nu) \right)$$

$$= |f|_{\mathfrak{C}^1} \left( \frac{1}{(1-\delta)^2} \left( |c|_{B_b(S \times A)} + \tau \left( 2|f|_{\mathfrak{C}^0} + |\log \eta(A)| \right) \right) \max \left\{ 2, \frac{5}{1-\delta} |f|_{\mathfrak{C}^0} \right\} + 4\tau |f|_{\mathfrak{C}^0} \right) \times$$

$$\times \left( |\theta' - \theta| + \mathcal{W}_1(\nu', \nu) \right),$$

for all $\nu', \nu \in \mathcal{P}_1(\mathbb{R}^d)$ and $\theta', \theta \in \mathbb{R}^d$, where the last inequality follows from the standard estimate $\max\{a, c + d\} \leq d + \max\{a, c\}$. Hence, we set

$$L_F := |f|_{\mathfrak{C}^1} \left( \frac{1}{(1-\delta)^2} \left( |c|_{B_b(S \times A)} + \tau \left( 2|f|_{\mathfrak{C}^0} + |\log \eta(A)| \right) \right) \max \left\{ 2, \frac{5}{1-\delta} |f|_{\mathfrak{C}^0} \right\} + 4\tau |f|_{\mathfrak{C}^0} \right).$$

$\square$

**Remark E.4.** *If $A$ is a finite action space, a natural choice for $\eta$ is the uniform distribution over $A$. In contrast, when $A$ is continuous, $\eta$ can be taken as a Gaussian or any other probability measure on $A$. Assuming $\eta \in \mathcal{P}(A)$, it follows that $|\log \eta(A)| = 0$. Note that the constants $C_F$ and $L_F$ in Proposition E.3 scale linearly with $\tau$. Therefore, as indicated by condition (10), increasing $\tau$*

*necessitates a corresponding increase in $\sigma$. This is intuitive, since the mapping $\nu \mapsto \mathrm{KL}(\pi_\nu | \eta)$ is non-convex. As a result, choosing a large $\tau$ in (61) amplifies the non-convexity of $\hat{V}_\tau(\nu)(\gamma)$, requiring a larger $\sigma$ to mitigate it.*

---

**Algorithm 1:** Langevin-based Best Response for bandits

---

**Input:** cost function $c$, activation function $f$, potential $U^\xi$, reference measure $\eta$ chosen according to Remark E.4, regularization parameter $\sigma$ as prescribed by (10), where $C_F, L_F$ are explicitly computed in Proposition E.3, $\tau$, initial distribution of parameters $\nu_0$, outer and inner step sizes $h_{\text{out}}$ and $h_{\text{in}}$, horizons $T$ and $K$, learning rate $\alpha > 0$, and number of parameters $N$.

1 Generate i.i.d. $(\theta_0^i)_{i=1}^N \sim \nu_0$ and set $(\theta_{0,0}^i)_{i=1}^N := (\theta_0^i)_{i=1}^N$;

2 **for** $t = 0,..., T-1$ **do**

3      **for** $k = 0, \ldots, K-1$ **do**

4          **for** $i = 1, 2, \ldots, N$ **do**

5              $\theta_{t,k+1}^i = \theta_{t,k}^i - h_{\text{in}}\big(\nabla_\theta \frac{\delta V_\tau^{\pi_\nu}}{\delta \nu}(\nu_{t,k}^N, \theta_{t,k}^i) + \sigma \nabla_\theta U^\xi(\theta_{t,k}^i)\big) + \sqrt{2\sigma h_{\text{in}}}\,\mathcal{N}_{t,k}^i$;

6          **end**

7      **end**

8      $\nu_{t+1}^N = (1 - \alpha h_{\text{out}})\nu_{t+1}^N + \alpha h_{\text{out}}\Psi_\sigma[\nu_t^N]$;

9 **end**

**Output:** $\nu_T^N = \frac{1}{N}\sum_{i=1}^N \delta_{\theta_T^i}$.

---

## E.2 Two-player zero-sum Markov game

We demonstrate how our results on min-max games can be applied to policy optimization in a two-player Markov game, thus extending the examples proposed in [21, Section 5] from convex-concave to non-convex-non-concave value function with softmax parametrized policies.

The setup is largely analogous to the MDP setting in Section 2.2. The goal of both players is to learn a policy that constitutes a Nash equilibrium of the game. For given strategies $\pi, \zeta : S \to \mathcal{P}^\eta(A)$, the $(\tau_1, \tau_2)$-entropy regularized total expected cumulative cost is given by

$$V_{\tau_1,\tau_2}^{\pi,\zeta}(s) := \mathbb{E}_s^{\pi,\zeta}\left[\sum_{n=0}^\infty \delta^n\left(c(s_n, a_n, b_n) + \tau_1 \log\frac{\mathrm{d}\pi}{\mathrm{d}\eta}(a_n|s_n) - \tau_2 \log\frac{\mathrm{d}\zeta}{\mathrm{d}\eta}(b_n|s_n)\right)\right].$$

For a fixed initial distribution $\gamma \in \mathcal{P}(S)$, the players' objective is then to solve

$$\min_{\pi \in \mathcal{P}^\eta(A|S)} \max_{\zeta \in \mathcal{P}^\eta(A|S)} V_{\tau_1,\tau_2}^{\pi,\zeta}(\gamma), \text{ with } V_{\tau_1,\tau_2}^{\pi,\zeta}(\gamma) := \int_S V_{\tau_1,\tau_2}^{\pi,\zeta}(s)\gamma(\mathrm{d}s). \tag{72}$$

Assume that there exists a unique MNE $(\pi_{\tau_1}^*, \zeta_{\tau_2}^*)$ of the game (72) independent of $\gamma$, i.e., there exists $(\pi_{\tau_1}^*, \zeta_{\tau_2}^*)$ such that

$$\min_{\pi \in \mathcal{P}^\eta(A|S)} \max_{\zeta \in \mathcal{P}^\eta(A|S)} V_{\tau_1,\tau_2}^{\pi,\zeta}(\gamma) = \max_{\zeta \in \mathcal{P}^\eta(A|S)} \min_{\pi \in \mathcal{P}^\eta(A|S)} V_{\tau_1,\tau_2}^{\pi,\zeta}(\gamma) = V_{\tau_1,\tau_2}^{\pi_{\tau_1}^*,\zeta_{\tau_2}^*}(\gamma),$$

and it is given by

$$\pi_{\tau_1}^*(\mathrm{d}a|s) \propto \exp\left(-\frac{1}{\tau_1}\int_A Q_{\tau_1,\tau_2}^{\pi_{\tau_1}^*,\zeta_{\tau_2}^*}(s,a)\zeta_{\tau_2}^*(\mathrm{d}b|s)\right)\eta(\mathrm{d}a),$$

$$\zeta_{\tau_2}^*(\mathrm{d}b|s) \propto \exp\left(\frac{1}{\tau_2}\int_A Q_{\tau_1,\tau_2}^{\pi_{\tau_1}^*,\zeta_{\tau_2}^*}(s,a)\pi_{\tau_1}^*(\mathrm{d}a|s)\right)\eta(\mathrm{d}b).$$

For example, such MNE exists for Markov games with finite state and action spaces [36].

To avoid requiring the players to search for the MNE $(\pi_{\tau_1}^*(\cdot|s), \zeta_{\tau_2}^*(\cdot|s))$ at each state $s \in S$ across the full space of probability measures, we assume that the strategies $\pi_\nu$ and $\zeta_\mu$ are of the softmax form

$$\pi_\nu(\mathrm{d}a|s) \propto \exp\left(f_\nu(s,a)\right)\eta(\mathrm{d}a)$$

and

$$\zeta_\mu(\mathrm{d}b|s) \propto \exp\left(g_\mu(s,b)\right)\eta(\mathrm{d}b),$$

where $f_\nu$ and $g_\mu$ are mean-field neural networks with activation functions $f, g : \mathbb{R}^d \times S \times A \to \mathbb{R}$, cf. equation (13). Thus, the goal of the players is now to find a global MNE of the game

$$\min_{\nu \in \mathcal{P}_1(\mathbb{R}^d)} \max_{\mu \in \mathcal{P}_1(\mathbb{R}^d)} V_{\tau_1,\tau_2}^{\pi_\nu,\zeta_\mu}(\gamma). \tag{73}$$

Owing to the presence of the normalization constant in (13), the value function $(\nu, \mu) \mapsto V_{\tau_1,\tau_2}^{\pi_\nu,\zeta_\mu}(\gamma)$ is non-convex-non-concave. Consequently, convergence of gradient flows to a global MNE cannot generally be expected without additional entropic regularization with respect to $\nu$ and $\mu$ (cf. (6)) as supported by our theoretical findings. In contrast, [21] considered the setting with non-parametric policies in (72), solving the problem by directly optimizing over the policies. This approach requires computing the MNE at each individual state, which can be computationally infeasible for large state spaces. By instead parametrizing the policies using the softmax form in (13) and optimizing over these parameters as in (73), the players are able to learn the MNE of (72) across all states simultaneously. Proposition E.5 states that $(\nu, \mu) \mapsto V_{\tau_1,\tau_2}^{\pi_\nu,\zeta_\mu}$ satisfies Assumption 3.1, and hence all our main results (Theorem B.2, Corollary B.3, Theorem B.4) and Theorem 3.3 apply to min-max problems (6) corresponding to energy functions $F^{\sigma_\nu,\sigma_\mu}(\nu, \mu) := V_{\tau_1,\tau_2}^{\pi_\nu,\zeta_\mu}(\gamma) + \sigma_\nu \, \mathrm{KL}(\nu|\xi) - \sigma_\mu \, \mathrm{KL}(\mu|\rho)$ for any reference measures $\xi, \rho$ satisfying Assumption 3.2.

As in the previous subsection, we state that the value function $V_{\tau_1,\tau_2}$ in the Markov games setting satisfies Assumption 3.1.

**Proposition E.5.** *Let $S$ and $A$ be Polish spaces, $c \in B_b(S \times A \times A)$, $\eta \in \mathcal{M}_+(A)$, and $\tau_1, \tau_2 > 0$. Then for all $\gamma \in \mathcal{P}(S)$ the function $F(\nu, \mu) := V_{\tau_1,\tau_2}^{\pi_\nu,\zeta_\mu}(\gamma)$ satisfies Assumption 3.1 with constants $C_F, L_F > 0$ depending on $\delta, \tau_1, |c|_{B_b(S \times A \times A)}, |f|_{\mathfrak{C}^0}, |f|_{\mathfrak{C}^1}, |\log \eta(A)|$, as in Proposition E.3, and constants $\bar{C}_F, \bar{L}_F > 0$ depending in an analogous way on $\delta, \tau_2, |c|_{B_b(S \times A \times A)}, |g|_{\mathfrak{C}^0}, |g|_{\mathfrak{C}^1}, |\log \eta(A)|$.*

*Proof.* The proof follows the same argument as the proof of Proposition E.3. ∎

# F  Additional notation and definitions

Following [6, Definition 5.43], we recall the notion of differentiability on the space of probability measure that we utilize throughout the paper.

**Definition F.1** (Flat differentiability on $\mathcal{P}_1(\mathbb{R}^d)$). *We say a function $F : \mathcal{P}_1(\mathbb{R}^d) \to \mathbb{R}$ is in $\mathcal{C}^1$, if there exists a continuous function $\frac{\delta F}{\delta \mu} : \mathcal{P}_1(\mathbb{R}^d) \times \mathbb{R}^d \to \mathbb{R}$, with respect to the product topology on $\mathcal{P}_1(\mathbb{R}^d) \times \mathbb{R}^d$, called the flat derivative of $F$, for which there exists $\kappa > 0$ such that for all $(\mu, x) \in \mathcal{P}_1(\mathbb{R}^d) \times \mathbb{R}^d, \left|\frac{\delta F}{\delta \mu}(\mu, x)\right| \leq \kappa (1 + |x|)$, and for all $\mu' \in \mathcal{P}_1(\mathbb{R}^d)$,*

$$\lim_{\varepsilon \searrow 0} \frac{F(\mu^\varepsilon) - F(\mu)}{\varepsilon} = \int_{\mathbb{R}^d} \frac{\delta F}{\delta \mu}(\mu, x)(\mu' - \mu)(\mathrm{d}x), \quad \text{with } \mu^\varepsilon = \mu + \varepsilon(\mu' - \mu), \tag{74}$$

*and $\int_{\mathbb{R}^d} \frac{\delta F}{\delta \mu}(\mu, x)\mu(\mathrm{d}x) = 0$.*

**Remark F.2.** *One can show that if $F : \mathcal{P}_1(\mathbb{R}^d) \to \mathbb{R}^d$ admits a flat derivative $\frac{\delta F}{\delta \mu}$, then for all $\mu, \mu' \in \mathcal{P}_1(\mathbb{R}^d)$, the function $[0,1] \ni \varepsilon \mapsto F(\mu^\varepsilon)$ is continuous on $[0,1]$ and differentiable on $(0,1)$ with derivative $\frac{\mathrm{d}}{\mathrm{d}\varepsilon}F(\mu^\varepsilon) = \int_{\mathbb{R}^d} \frac{\delta F}{\delta \mu}(\mu^\varepsilon, x)(\mu' - \mu)(\mathrm{d}x)$ (see [19, Theorem 2.3]). Hence, by the fundamental theorem of calculus, $F(\mu') - F(\mu) = \int_0^1 \int_{\mathbb{R}^d} \frac{\delta F}{\delta \mu}(\mu^\varepsilon, x)(\mu' - \mu)(\mathrm{d}x)\mathrm{d}\varepsilon$, provided that $\varepsilon \mapsto \int \frac{\delta F}{\delta \mu}(\mu^\varepsilon, x)(\mu' - \mu)(\mathrm{d}x)$ is integrable.*

