# OpenReview forum: "Non-convex entropic mean-field optimization via Best Response flow"
_NeurIPS.cc/2025/Conference — NeurIPS 2025 poster_

### Official Review · Reviewer_YdjA · 2025-06-29

**Clarity:** 2
**Significance:** 3
**Originality:** 3
**Rating:** 4
**Confidence:** 3

**Summary:**

The paper considers the setting of minimizing non-convex functionals over the space of probability measures, under the presence of entropy regularization. The author utilizes best response flow and shows the global convergence in this case, under the condition that the regularization strength is chosen so that the best response operator becomes a contraction. The paper also extends the results to a two-player setting for solving entropy-regularized nonconvex nonconcave minimax problems. Applications include policy optimization in RL with softmax parameterized policy.

**Questions:**

I have the following questions and suggestions, centered around the weakness mentioned above. I will be delighted to increase my recommendation for this paper if the questions can be addressed. Some of them might be due to my misunderstanding of the work; in this case, clarifications are certainly welcome.

1. What's the benefit of considering a softmax parameterized policy, which likely causes the non-convexity issue presented in the RL example in the paper? Does using a softmax policy result in stronger or faster convergence, or what are its benefits? Adding a paragraph discussing this after introducing softmax policy would improve paper readability. A clear comparison in terms of the theoretical results that can be achieved between the softmax policy or regular policy would definitely make the motivation of the work more straightforward.

2. The convergence results seem only to be established in high regularization regime, which seems to be a less practical scenario. Can you elaborate on how restrictive eq (12) is, and what's the technical difficulty for extending the convergence result to any sigma > 0?

3. The paper is an extension of results in [1] and [2]. What are the challenges faced in the extension, and what novel theoretical tools/techniques are used that lead to the improved results in this paper? A clear highlight of this in the paper can help the reader better understand the paper's significance and contribution.

4. I am aware of another thread of work [3,4,5,6], which also studies policy optimization in RL using mean-field NN techniques. These works include both single-agent and two-player minimax scenarios, and they haven't been discussed in the paper. Could you briefly elaborate on how your work differs from or relates to these papers in terms of problem setting, assumptions, etc? I noticed that [3,4,5,6] mostly achieves a sublinear rate O(1/T), which is different from the exponential convergence in this paper, and I do wonder what leads to this. I believe that including discussions of these relevant papers will better help the broader community understand the position and importance of this work in the literature.

**Reference**:

[1] Chen, Fan, Zhenjie Ren, and Songbo Wang. "Entropic fictitious play for mean field optimization problem." Journal of Machine Learning Research 24.211 (2023): 1-36.

[2] Lascu, Razvan-Andrei, Mateusz B. Majka, and Łukasz Szpruch. "Entropic mean-field min–max problems via best response flow." Applied Mathematics & Optimization 91.2 (2025): 48.

[3] Zhang, Yufeng, et al. "Can Temporal-Diﬀerence and Q-Learning Learn Representation? A Mean-Field Theory." Advances in Neural Information Processing Systems 33 (2020): 19680-19692.

[4] Zhang, Yufeng, et al. "Wasserstein flow meets replicator dynamics: A mean-field analysis of representation learning in actor-critic." Advances in Neural Information Processing Systems 34 (2021): 15993-16006.

[5] Zhu, Yuchen, et al. "A Mean-Field Analysis of Neural Stochastic Gradient Descent-Ascent for Functional Minimax Optimization." arXiv preprint arXiv:2404.12312 (2024).

[6] Yamamoto, Kakei, et al. "Mean field langevin actor-critic: Faster convergence and global optimality beyond lazy learning." Forty-first International Conference on Machine Learning. 2024.

**Ethical Concerns:**

["NO or VERY MINOR ethics concerns only"]

**Final Justification:**

The paper is a technically solid paper with a good theoretical contribution to the field. Therefore, I recommend borderline acceptance.

**Limitations:**

yes, limitations are addressed.

**Quality:**

3

**Strengths And Weaknesses:**

**Strength:**
1. The paper considers a seemingly challenging problem setting, which is establishing global convergence rate for functional optimization without assuming convexity/concavity.
2. The studied setting is pretty comprehensive, including both single agent and two-player scenarios, important for practical relevant applications such as policy optimization in RL.
3. The technical results are solid.

**Weakness:**
1. Paper readability could be improved. The motivation for using the theoretical tools and the setting considered is not very clear. The presentation of the results makes it hard for readers who are not extremely familiar with the literature to follow. See the **questions section** for details.

2. Clarity of paper contributions, novelty, and the position of this work in the literature. In my opinion, this paper could have a much larger audience by clearly discussing more of the relevant literature (in the intro or appendix), and elucidating the difference of results in convergence/convergence rate under different sets of assumptions. While this paper addressed a challenging, new setting, the key part in the proof that leads to this novel result is not highlighted. A better, clearer emphasis on the central contribution of this paper would make its impact more profound. See the **questions section** for details.

---

> ### Author Rebuttal · Authors · 2025-07-30
>
> We would like to thank the reviewer for appreciating the contribution of our paper, for their feedback and questions. We would like to address the weaknesses and questions formulated by the reviewer:
> ### Questions
> + The need for working with parametrized policies (in our case softmax) instead of vanilla policies comes from the application to MDPs and subsequently Markov games. In MDPs we parametrize because policies depend on the state, thus when the state changes, so does the policy. Hence one cannot optimize directly over policies because for each state one needs to solve a different optimization problem. To work around this, one assumes that the policies belong to a certain parametrization class and optimizes over the parameters, which are state-independent. Hence regardless of what state the player is at, the player can still learn the policy by optimizing the parameters. So, one needs to solve only one optimization problem over parameters instead of potentially many optimization problems (one for each state) over policies. According to Section 3 in [Aga], there are more classes of policy parametrization $(\pi_{\theta})\_{\theta \in \Theta},$ where $\theta$ are the parameters and $\Theta$ is the parameter space, as follows:
>         1. Direct parametrization: $\pi_{\theta}(a|s) = \theta(s,a)$ such that $\theta(s,a) \geq 0$ and $\sum_{a}\theta(s,a) = 1$ for all $a \in A$ and $s \in S,$
>
>         2. Softmax parametrization: $\pi_{\theta}(a|s) = \frac{e^{\theta(s,a)}}{\sum_{a' \in A}e^{\theta(s,a')}}$ for all $\theta \in \mathbb R^{|S||A|}.$
>
>        3. Log-linear and neural policies.
>
>     The exponential form of softmax policies is motivated by the formula satisfied by the optimal policy, see $\pi_{\tau}^*$ in Section 2.2, hence their popularity in the literature on MDPs, e.g., [1, 21, 24, 26, 37, [Aga]].
>
>     Considering softmax policies for parameters in the space $\mathbb R^{|S||A|}$ can be computationally challenging as the size of the sets $S$ and $A$ becomes large. To overcome this and also to cover the continuous state-action spaces, we consider softmax policies approximated by a mean-field neural network. As we explain in the introduction, this approach was also considered in other works such as  [1, 21, 24, 37].
>
>     References:
> [Aga] Agarwal et al. On the Theory of Policy Gradient Methods: Optimality, Approximation, and Distribution Shift
>
> + We believe that without adding any regularization, the problem of solving the non-convex infinite dimensional optimization problem (1) becomes intractable.
>     The challenge is, on one hand, in establishing convergence of an appropriately chosen gradient flow, but also in determining whether the  measure that the flow converges to is actually the global minimizer of the problem under consideration (note that without any regularization, the problem may have multiple local and global minimizers). Hence the crucial question is to establish the minimal threshold for $\sigma$ that allows one to find a gradient flow converging to the global minimizer of (1), and determine the specific mathematical property that guarantees this convergence. The main novelty in our paper is the observation that one can use the mathematical tools developed in [7] to establish that for a certain range of $\sigma$ the Best Response flow from [7] becomes a $W_1$-contraction and then conclude how this property implies the existence of a unique minimizer to (1), as well as the exponential convergence of the flow to that minimizer. We note that the $W_1$-contraction property is fundamentally different than strong convexity (it would seem like a more natural idea to solve the problem by increasing $\sigma$ to turn it into a strongly convex problem, but it is not what we choose to do). Indeed, The lower bound on $\sigma$ in Theorem 2.3 does not directly imply that the functional $F^{\sigma}$ in problem (1) is strongly convex. In fact, it is not possible to straightforwardly show that there exists a lower bound on $\sigma$ such that $F^{\sigma}$ becomes strongly convex in problem (1). The limitation is illustrated by the following computation. Note that
> \begin{align*}
>     F^{\sigma}(\nu')-F^{\sigma}(\nu) - \int \frac{\delta F^{\sigma}}{\delta \nu}(\nu,x)(\nu'-\nu)(dx) =  F(\nu')-F(\nu) - \int \frac{\delta F}{\delta \nu}(\nu,x)(\nu'-\nu)(dx)+\sigma\operatorname{KL}(\nu'|\xi) - \sigma\operatorname{KL}(\nu|\xi) - \sigma\int \frac{\delta \operatorname{KL}(\cdot|\xi)}{\delta \nu}(\nu,x)(\nu'-\nu)(dx).
> \end{align*}
> Remark B.2 and Assumption 2.1 give
> \begin{align*}
>     \left|F(\nu')-F(\nu) - \int \frac{\delta F}{\delta \nu}(\nu,x)(\nu'-\nu)(dx)\right|&= \left|\int_0^1 \int \left(\frac{\delta F}{\delta \nu}(\nu+\varepsilon(\nu'-\nu),x) - \frac{\delta F}{\delta \nu}(\nu,x)\right)(\nu'-\nu)(dx)d\varepsilon\right|
>     \leq L_F\int_0^1 \int \mathcal{W}_1(\nu+\varepsilon(\nu'-\nu),\nu)|\nu'-\nu|(dx)d\varepsilon
>     = \frac{L_F}{2}\operatorname{TV}(\nu',\nu)\int_0^1 \mathcal{W}_1(\nu+\varepsilon(\nu'-\nu),\nu)d\varepsilon
>     \leq \frac{L_F}{2} \operatorname{TV}(\nu',\nu) \mathcal{W}_1(\nu',\nu)\int_0^1\varepsilon d\varepsilon
>     = \frac{L_F}{4}\operatorname{TV}(\nu',\nu)\mathcal{W}_1(\nu',\nu),
> \end{align*}
> where the second inequality uses the convexity of the Wasserstein distance.
>
> On the other hand, since $\frac{\delta \operatorname{KL}(\cdot|\xi)}{\delta \nu}(\nu,x) = \log \frac{d \nu}{d \xi}(x) -\operatorname{KL(\nu|\xi)},$ we have
> \begin{equation*}
>     \operatorname{KL}(\nu'|\xi) - \operatorname{KL}(\nu|\xi) - \int \frac{\delta \operatorname{KL}(\cdot|\xi)}{\delta \nu}(\nu,x)(\nu'-\nu)(dx) = \operatorname{KL}(\nu'|\nu) \quad (Eqn 1).
> \end{equation*}
> Moreover, by Pinker's inequality $\operatorname{TV}^2(\nu',\nu)\leq \frac{1}{2}\operatorname{KL}(\nu'|\nu),$ so we obtain
> \begin{equation*}
>     F^{\sigma}(\nu')-F^{\sigma}(\nu) - \int \frac{\delta F^{\sigma}}{\delta \nu}(\nu,x)(\nu'-\nu)(dx) \geq -\frac{L_F}{4}\operatorname{TV}(\nu',\nu)\mathcal{W}_1(\nu',\nu) + 2\sigma\operatorname{TV}^2(\nu',\nu).
> \end{equation*}
> This shows that while the KL divergence has natural strong convexity in terms of $\operatorname{TV}^2$, this does not align with the $\mathcal{W}_1$-based Lipschitz continuity of $F$. Therefore, it does not seem possible to directly deduce strong convexity of $F^{\sigma}$ in the $\operatorname{TV}^2$ sense simply by choosing $\sigma$ large enough relative to $L_F$.
>
> This is one of the motivations for proving convergence via contraction of the Best Response operator rather than by attempting to enforce strong convexity of the objective function.
>
> As a further remark, it would be possible to modify Assumption 2.1 to assume Lipschitz continuity with respect to TV rather than $W_1$, then we would obtain:
> $$F^{\sigma}(\nu')-F^{\sigma}(\nu) - \int \frac{\delta F^{\sigma}}{\delta \nu}(\nu,x)(\nu'-\nu)(dx) \geq -\frac{L_F}{4}\operatorname{TV}^2(\nu',\nu) + 2\sigma\operatorname{TV}^2(\nu',\nu),$$ so choosing $\sigma \geq \frac{1}{8}L_F$ would ensure strong convexity of $F^{\sigma}$ relative to $\operatorname{TV}^2.$ Even though existence and uniqueness of a minimizer could now be established, it is not obvious how to prove that $\operatorname{TV}^2(\nu_t, \nu_\sigma^*) \to 0$ for the Best Response flow $(\nu_t)_{t \geq 0}$. Moreover, this would change the setting of the problem, which would no longer be comparable to [7,21], which work in the Wasserstein space.
>
> Finally, we emphasize that this argument eliminates any dependence on the choice of the reference measure $\xi$ (cf. equation (Eqn 1)) and hence any lower bound on $\sigma$ obtained this way, which would guarantee strong convexity of $F^{\sigma}$ with respect to $\operatorname{TV}^2$, would be independent of the choice of $\xi$. On the other hand, one of the important conclusions from our paper is that by following the proof via $W_1$-contractivity of the Best Response operator, we can identify the connection between the choice of $\sigma$ and the reference measure $\xi$ expressed in terms of inequality (12), that guarantees convergence of the BR flow to the global minimizer. This allows one to choose a smaller value of $\sigma$ if the measure $\xi$ is chosen appropriately.
>
> + We will add additional explanations to the revised version of the paper, and in particular we will re-write Subsection 1.2 (Our contributions) to highlight the complexity of the problem (see also our response to the previous point).
>
> + We thank the reviewer for pointing out the references. We will add them to the revised version of the paper, along with a short comment about their relation to our paper. Note that [3,4,5,6] consider the mean-field approach to a class of actor-critic methods in reinforcement learning (RL), which is a different setting to the one studied in our paper (and hence the results are not directly comparable). One can interpret our results as addressing the actor component, specifically using a policy gradient method, namely, the Best Response (BR) flow, to learn an optimal policy that optimizes the value function $V$. Accordingly, our convergence analysis pertains to the actor's learning dynamics.
>     As a direction for future work, we agree it would be interesting to extend our framework to incorporate the critic component as well. In a typical actor-critic setup, the critic estimates the action-value function $Q$ and value function $V$ using methods such as temporal-difference learning or Q-learning, by minimizing the mean-squared Bellman error, while the actor updates the policy based on the critic's feedback.
>     Finally, we note that the min-max game studied in our paper is a two-player Markov game and is not directly related to the min-max formulations commonly encountered in actor-critic learning. To our knowledge, there is no existing actor-critic work where the actor learns the policy via the Best Response flow. This presents a wide array of open problems to be considered in future work.

---

> > ### Comment · Reviewer_YdjA · 2025-08-02
> >
> > Thanks for the detailed response! My concerns are mostly well-addressed. I will raise my recommendation of this work, given that the promised revision is included in the updated manuscript, including additional explanations of the challenging part of the studied problem, the novelty of the proof techniques, and the discussion of the missing references [3,4,5,6]. Good luck!

---

> > > ### Author Response · Authors · 2025-08-03
> > > **Official Comment by Authors**
> > >
> > > We thank the reviewer for considering our rebuttal and raising their score. We will include the discussed changes in the camera-ready version if the paper is accepted.

---

### Official Review · Reviewer_p8Yk · 2025-07-02

**Clarity:** 2
**Significance:** 2
**Originality:** 2
**Rating:** 4
**Confidence:** 3

**Summary:**

This paper studies entropy-regularized optimization over probability measures, extending convergence guarantees of the Best Response (BR) flow from convex to certain non-convex settings. The authors consider non-convex functionals regularized by KL divergence and provide sufficient conditions under which the BR operator becomes a contraction in the Wasserstein-1 distance. This ensures global convergence to a unique minimizer or mixed Nash equilibrium in both single-agent and two-player min-max problems, under appropriate regularization. Applications to MDP are also discussed.

**Questions:**

Please see above.

**Ethical Concerns:**

["NO or VERY MINOR ethics concerns only"]

**Final Justification:**

I appreciate the authors for the response, which addresses some of my concerns. I will raise the score.

**Limitations:**

Please see above.

**Quality:**

2

**Strengths And Weaknesses:**

## Strengths
* Theoretical Advancement: Extends recent results on BR flow to non-convex (and non-concave) problems.

* Rigorous Analysis: The mathematical development is thorough, with formal proofs of contractivity, uniqueness, and exponential convergence rates in Wasserstein-1 distance.

## Weakness
* Lack of Empirical Evaluation: The paper is purely theoretical, and no experiments such as that in [18] are provided to illustrate the convergence behavior of the best-response flow in practice.

* Choice and Estimation of $\sigma$: The convergence results rely on $\sigma$ being “sufficiently large,” but the paper does not provide explicit guidance on how to estimate or choose an appropriate value of $\sigma$ in practice. This limits the direct applicability of the theory to algorithm design.

* Convergence in Different Metrics: The convergence guarantees are stated in terms of the Wasserstein-1 ($W_1$) distance. It would strengthen the contribution to discuss whether similar results hold under other metrics such as $W_2$ or KL divergence, which are more commonly used in measure-based optimization.

* Limited Discussion of Technical Contributions: The theoretical guarantees crucially rely on the regularization parameter $\sigma$ being sufficiently large. Please correct me if I am wrong, I guess large enough $\sigma$ helps eliminate most of difficulties in proof. This raises the question of how much of the difficulty is resolved by the regularization itself. A more detailed explanation of the core technical contributions would better highlight the paper’s originality.

 * Exposition and Readability: While the results are theoretically interesting, definitions, assumptions, and several technical arguments are densely written. Improving the organization would make the work more accessible to a broader audience.

---

> ### Author Rebuttal · Authors · 2025-07-30
>
> We would like to thank the reviewer for appreciating the contribution of our paper, for their feedback and questions. We would like to address the weaknesses and questions formulated by the reviewer:
>    + Since the first version of this paper was submitted in May, we have managed to implement the Best Response flow for an example of the bandit problem described on page 2 for which we explicitly compute the constants $C_F$ and $L_F$ in Proposition A.1. We would be happy to add a discussion about this implementation to the paper, using the extra page available in the camera-ready version. For more details on the algorithm that we used, see the Questions section of reviewer tmuc.
>
> + Inequality (12) in Theorem 2.3, while looking cumbersome at first glance, is explicit in terms of the model parameters and hence can be used in practice to find the required value of $\sigma$ for specific models. In the context of the bandit problem, we verify our assumptions in Proposition A.1 and we find explicit formulas for the constants appearing in formula (12). This can be then used in the algorithm described above to choose the right value of $\sigma$ in the numerical experiments (note that once the reference measure $\xi$ is chosen, finding its first moment is easy and hence the whole problem behind evaluating the bound in formula (12) is in finding the values of the constants $C_F$ and $L_F$). We will add the details in the camera-ready version of the paper. Two additional remarks about this:
> 1. In the current version of Proposition A.1 the constants relevant for formula (12) are not written explicitly but they can be easily recovered from the proof. In the revised version we will showcase the formulas for these constants in the statement of the theorem
>
>  2. Since the original submission in May, we have managed to extend the proof of Proposition A.1 to more general MDP models and we will include this extended result in the revised version of the supplementary material. This allows us to find the explicit bound on $\sigma$ for a wider class of models.
>
>
> + We have not been able to obtain similar results in $W_2$, but it is possible to obtain convergence in $\operatorname{KL}$-divergence under some additional regularity assumptions.
>     1. For $W_2$, the technical reason why our proof does not work is that Proposition 7.10 from [Villani], which is crucial for our proof, states that for $p \geq 1,$ we have
>     \begin{equation*}
>         W_p^p(f\mu,g\mu) \leq C_{p}\||\cdot|^p(f-g)\|\_{L\_{\mu}^1(\mathbb R^d)}.
>     \end{equation*}
>     For $p=1$ we can use this result to obtain contraction for the Best Response operator, however for $p>1$ there is a mismatch between the powers on both sides (for our proof to work, we would need the norm on the right hand side to be raised to the power $p$). We have not managed to find a workaround for this issue. However, it is not surprising that for non-convex optimization, proving a $W_1$ contraction is easier than proving a $W_2$ contraction. In general, a $W_2$ contraction implies the corresponding result in $W_1$, but not vice versa, and in the optimal transport theory there are known examples where $W_1$-contraction holds, but $W_2$-contraction does not (or $W_2$-contraction requires stronger assumptions). For instance, in the context of Langevin SDEs corresponding to the Wasserstein gradient flows, their contractivity is analysed in non-convex settings only in $W_1$ in [EGZ], and for related results in $W_2$ additional assumptions are needed [Mon]. We believe that analysing the convergence of the Best Response flow in $W_2$ is a challenging problem that will require additional future work.
>     2. For $\operatorname{KL}$ divergence, we can prove that $\operatorname{KL}(\nu_t|\Psi[\nu_t]) \to 0$ as $t \to \infty$ exponentially fast when $\sigma$ is sufficiently large, if we additionally assume that $F$ has a bounded second order flat derivative. Indeed, we can calculate
>     \begin{equation*}
>         \partial_t \operatorname{KL}(\nu_t|\Psi[\nu_t]) = -\alpha(\operatorname{KL}(\nu_t|\Psi[\nu_t]) + \operatorname{KL}(\Psi[\nu_t]|\nu_t)) - \frac{\alpha}{\sigma}\int \int \frac{\delta^2 F}{\delta \nu^2}(\nu_t,x,x')(\Psi[\nu_t] - \nu_t)(dx')(\Psi[\nu_t] - \nu_t)(dx).
>     \end{equation*}
>     Then we can upper bound
>     \begin{equation*}
>         \int \int \frac{\delta^2 F}{\delta \nu^2}(\nu_t,x,x')(\Psi[\nu_t] - \nu_t)(dx')(\Psi[\nu_t] - \nu_t)(dx) \leq L_F\operatorname{TV}^2(\nu_t, \Psi[\nu_t])
>     \end{equation*}
> Next, by Pinsker's inequality, $\operatorname{TV}^2(\nu_t, \Psi[\nu_t]) \leq \frac{1}{2}\operatorname{KL}(\nu_t|\Psi[\nu_t]),$ we get
>     \begin{equation*}
>         \partial_t \operatorname{KL}(\nu_t|\Psi[\nu_t]) \leq -\alpha(1-\frac{L_F}{2\sigma})\operatorname{KL}(\nu_t|\Psi[\nu_t]).
>     \end{equation*}
>     If $\sigma > \frac{L_F}{2},$ then we obtain
>     \begin{equation*}
>         \operatorname{KL}(\nu_t|\Psi[\nu_t]) \leq e^{-\alpha t(1-\frac{L_F}{2\sigma})}\operatorname{KL}(\nu_0|\Psi[\nu_0]).
>     \end{equation*}
>     We would like to stress, however, that this convergence result requires us first to prove the $W_1$-contraction from Theorem 2.3, which guarantees that the fixed point problem $\nu=\Psi[\nu]$ has a unique solution that is actually the global minimizer of (1), then to add one more assumption about the regularity of $F$ and an additional lower bound on $\sigma$ (note that our inequality (12) from Theorem 2.3 does not imply that $\sigma > \frac{L_F}{2}$).
>
>
>    References:
> Villani, Topics in Optimal Transportation;
>
> EGZ:  Andreas Eberle, Arnaud Guillin, and Raphael Zimmer. Quantitative Harris-type theorems for diffusions and McKean-Vlasov processes. Trans. Amer. Math. Soc., 371(10):7135–7173, 2019;
>
> Mon: Pierre Monmarche. Wasserstein contraction and Poincare inequalities for elliptic diffusions at high temperature. Annales de l’Institut Henri Lebesgue 6 (2023) 941-973
>
> + We believe that without adding any regularization, the problem of solving the non-convex infinite dimensional optimization problem (1) becomes intractable.
>     The challenge is, on one hand, in establishing convergence of an appropriately chosen gradient flow, but also in determining whether the  measure that the flow converges to is actually the global minimizer of the problem under consideration (note that without any regularization, the problem may have multiple local and global minimizers). Hence the crucial question is to establish the minimal threshold for $\sigma$ that allows one to find a gradient flow converging to the global minimizer of (1), and determine the specific mathematical property that guarantees this convergence. The main novelty in our paper is the observation that one can use the mathematical tools developed in [7] to establish that for a certain range of $\sigma$ the Best Response flow from [7] becomes a $W_1$-contraction and then conclude how this property implies the existence of a unique minimizer to (1), as well as the exponential convergence of the flow to that minimizer. We note that the $W_1$-contraction property is fundamentally different than strong convexity (it would seem like a more natural idea to solve the problem by increasing $\sigma$ to turn it into a strongly convex problem, but it is not what we choose to do). Indeed, The lower bound on $\sigma$ in Theorem 2.3 does not directly imply that the functional $F^{\sigma}$ in problem (1) is strongly convex. The limitation is illustrated by the computation in Weaknesses section of reviewer tmuc.
>
> + We agree that certain parts of the exposition need to be reformulated and clarified. We are going to use the feedback from all the referees to improve the presentation of our results and to make them more accessible.

---

> > ### Comment · Reviewer_p8Yk · 2025-08-02
> >
> > I appreciate the authors for the response, which addresses some of my concerns. I will raise the score.

---

> > > ### Author Response · Authors · 2025-08-03
> > > **Official Comment by Authors**
> > >
> > > We thank the reviewer for considering our rebuttal and raising their score. We will include the discussed changes in the camera-ready version if the paper is accepted.

---

### Official Review · Reviewer_tmuc · 2025-07-03

**Clarity:** 3
**Significance:** 3
**Originality:** 3
**Rating:** 4
**Confidence:** 3

**Summary:**

This paper investigates the problem of minimizing non-convex functionals over probability measure spaces. Given a non-convex functional, the authors demonstrate that by selecting an appropriate regularization term, the best-response operator can be rendered a contraction mapping with respect to the $L^1$-Wasserstein distance. They prove that the fixed point of this contraction mapping is the unique global minimizer of the optimization problem. Additionally, the study extends to address the corresponding issues in entropy-regularized non-convex-non-concave minimax problems.

**Questions:**

Does the author believe that the softmax-form non-convex problem considered in this paper can be solved with computational guarantees, i.e., using discrete-time, discrete-space Langevin algorithms?

**Ethical Concerns:**

["NO or VERY MINOR ethics concerns only"]

**Final Justification:**

The authors' responses in their rebuttal have adequately addressed my concerns, leading me to decide to maintain my original score.

**Limitations:**

See weakness

**Paper Formatting Concerns:**

There are no formatting issues.

**Quality:**

3

**Strengths And Weaknesses:**

Strengths:
（1）Based on weaker assumptions, this paper proves the convergence of the best-response flow for non-convex probability density function optimization problems.
（2）Building on conclusions from non-convex optimization, the paper demonstrates the best-response flow in modified L₁-Wasserstein metrics for non-convex-non-concave minimax problems, rendering the content more robust.

Weaknesses:
（1）According to the statement of Theorem 2.3, $\sigma$ appears to be of constant order. For Problem (1), is a large $\sigma$ acceptable? If $\sigma$ reaches the magnitude required by Theorem 2.3, does it transform Problem (1) into a strongly convex problem?

---

> ### Author Rebuttal · Authors · 2025-07-30
>
> We would like to thank the reviewer for appreciating the contribution of our paper, for their feedback and questions. We would like to address the weaknesses and questions formulated by the reviewer:
> ### Weaknesses
> The lower bound on $\sigma$ in Theorem 2.3 does not directly imply that the functional $F^{\sigma}$ in problem (1) is strongly convex. In fact, it is not possible to straightforwardly show that there exists a lower bound on $\sigma$ such that $F^{\sigma}$ becomes strongly convex in problem (1). The limitation is illustrated by the following computation. Note that
> \begin{align*}
>     F^{\sigma}(\nu')-F^{\sigma}(\nu) - \int \frac{\delta F^{\sigma}}{\delta \nu}(\nu,x)(\nu'-\nu)(dx) =  F(\nu')-F(\nu) - \int \frac{\delta F}{\delta \nu}(\nu,x)(\nu'-\nu)(dx)+\sigma\operatorname{KL}(\nu'|\xi) - \sigma\operatorname{KL}(\nu|\xi) - \sigma\int \frac{\delta \operatorname{KL}(\cdot|\xi)}{\delta \nu}(\nu,x)(\nu'-\nu)(dx).
> \end{align*}
> Remark B.2 and Assumption 2.1 give
> \begin{align*}
>     \left|F(\nu')-F(\nu) - \int \frac{\delta F}{\delta \nu}(\nu,x)(\nu'-\nu)(dx)\right|= \left|\int_0^1 \int \left(\frac{\delta F}{\delta \nu}(\nu+\varepsilon(\nu'-\nu),x) - \frac{\delta F}{\delta \nu}(\nu,x)\right)(\nu'-\nu)(dx)d\varepsilon\right|\\
>     \leq L_F\int_0^1 \int \mathcal{W}_1(\nu+\varepsilon(\nu'-\nu),\nu)|\nu'-\nu|(dx)d\varepsilon\\
>     = \frac{L_F}{2}\operatorname{TV}(\nu',\nu)\int_0^1 \mathcal{W}_1(\nu+\varepsilon(\nu'-\nu),\nu)d\varepsilon\\
>     \leq \frac{L_F}{2} \operatorname{TV}(\nu',\nu) \mathcal{W}_1(\nu',\nu)\int_0^1\varepsilon d\varepsilon\\
>     = \frac{L_F}{4}\operatorname{TV}(\nu',\nu)\mathcal{W}_1(\nu',\nu),
> \end{align*}
> where the second inequality uses the convexity of the Wasserstein distance.
>
> On the other hand, since $\frac{\delta \operatorname{KL}(\cdot|\xi)}{\delta \nu}(\nu,x) = \log \frac{d \nu}{d \xi}(x) -\operatorname{KL(\nu|\xi)},$ we have
>    $$ \operatorname{KL}(\nu'|\xi) - \operatorname{KL}(\nu|\xi) - \int \frac{\delta \operatorname{KL}(\cdot|\xi)}{\delta \nu}(\nu,x)(\nu'-\nu)(dx) = \operatorname{KL}(\nu'|\nu) \quad (Eqn. 1).$$
> Moreover, by Pinsker's inequality $\operatorname{TV}^2(\nu',\nu)\leq \frac{1}{2}\operatorname{KL}(\nu'|\nu),$ so we obtain
> \begin{equation*}
>     F^{\sigma}(\nu')-F^{\sigma}(\nu) - \int \frac{\delta F^{\sigma}}{\delta \nu}(\nu,x)(\nu'-\nu)(dx) \geq -\frac{L_F}{4}\operatorname{TV}(\nu',\nu)\mathcal{W}_1(\nu',\nu) + 2\sigma\operatorname{TV}^2(\nu',\nu).
> \end{equation*}
> This shows that while the KL divergence has natural strong convexity in terms of $\operatorname{TV}^2$, this does not align with the $\mathcal{W}_1$-based Lipschitz continuity of $F$. Therefore, it does not seem possible to directly deduce strong convexity of $F^{\sigma}$ in the $\operatorname{TV}^2$ sense simply by choosing $\sigma$ large enough relative to $L_F$.
>
> This is one of the motivations for proving convergence via contraction of the Best Response operator rather than by attempting to enforce strong convexity of the objective function.
>
> As a further remark, it would be possible to modify Assumption 2.1 to assume Lipschitz continuity with respect to TV rather than $W_1$, then we would obtain:
> \begin{equation*}
> F^{\sigma}(\nu')-F^{\sigma}(\nu) - \int \frac{\delta F^{\sigma}}{\delta \nu}(\nu,x)(\nu'-\nu)(dx) \geq -\frac{L_F}{4}\operatorname{TV}^2(\nu',\nu) + 2\sigma\operatorname{TV}^2(\nu',\nu),
> \end{equation*}
> so choosing $\sigma \geq \frac{1}{8}L_F$ would ensure strong convexity of $F^{\sigma}$ relative to $\operatorname{TV}^2.$ Even though existence and uniqueness of a minimizer could now be established, it is not obvious how to prove that $\operatorname{TV}^2(\nu_t, \nu_\sigma^*) \to 0$ for the Best Response flow $(\nu_t)_{t \geq 0}$. Moreover, this would change the setting of the problem, which would no longer be comparable to [7,21], which work in the Wasserstein space.
>
> Finally, we emphasize that this argument eliminates any dependence on the choice of the reference measure $\xi$ (cf. equation (Eqn. 1)) and hence any lower bound on $\sigma$ obtained this way, which would guarantee strong convexity of $F^{\sigma}$ with respect to $\operatorname{TV}^2$, would be independent of the choice of $\xi$. On the other hand, one of the important conclusions from our paper is that by following the proof via $\mathcal{W}_1$-contractivity of the Best Response operator, we can identify the connection between the choice of $\sigma$ and the reference measure $\xi$ expressed in terms of inequality (12), that guarantees convergence of the BR flow to the global minimizer. This allows one to choose a smaller value of $\sigma$ if the measure $\xi$ is chosen appropriately.
>
> ### Questions
> The problem considered in this paper can indeed be solved computationally via a Langevin-based algorithm. Crucially, we can explicitly compute the constants $C_F$ and $L_F$ for the bandit problem by following the proof of Proposition A.1. Then we can construct an algorithm which updates the parameters $\nu$ of the mean-field neural network using a Langevin particle algorithm involving two nested loops.
>
> We take as inputs the cost function $c,$ the regularization parameter $\tau,$ the activation function $f$ of the network, the reference measure $\eta$ (which for simplicity we could consider to be a Gaussian over $A$ so that $\eta(A) =1$ and $|\log \eta (A)| = 0$), the parameter $\sigma$ as prescribed by (12), where $C_F, L_F$ are explicitly computed in Proposition A.1, the potential $U^{\xi},$ step sizes $h_1,$ $h_2,$ and the learning rate $\alpha.$
>
> We approximate $\nu$ by its empirical measure $\nu^N = \frac{1}{N}\sum_{i=1}^N \delta_{X^i},$ where $(X^i)_{i=1}^N$ represent the parameters of the neural network appearing in the soft-max parametrization of the policy.
>
> The algorithm works as follows. We initialize the parameters of the network by sampling iid $(X_0^i)\_{i=1}^{N}$ from $\nu_0.$ Then since the Best Response operator $\Psi_{\sigma}$ is a probability measure with exponential form, we can compute it by running Langevin dynamics.
>
> Recall that $$F^{\sigma}(\nu) = V_\tau^{\pi_\nu} + \sigma\operatorname{KL}(\nu|e^{-U^{\xi}}).$$ Set $(X_{0,0}^i)\_{i=1}^{N} = (X_0^i)\_{i=1}^{N}.$ Then for $K$ steps we do
> \begin{equation}
>     X\_{t,k+1}^i = X\_{t,k}^i -h_1\left(\nabla \frac{\delta {V_\tau}^{\pi\_{\nu}}}{\delta \nu}(\nu,\cdot) + \sigma \nabla U^{\xi}\right) +\sqrt{2h_1\sigma}\mathcal{N}\_{t,k}^i, \quad i = 1,...,N,
> \end{equation}
> where $\mathcal{N}\_{t,k}^i$ are iid Gaussians. Note that $\frac{\delta V_\tau^{\pi_\nu}}{\delta \nu}(\nu,\cdot)$ is explicitly computed in Proposition A.1. Then
> \begin{equation*}
>     \Psi_{\sigma}[\nu^N_t] = \frac{1}{N}\sum_{i=1}^N \delta_ {X_{t,K}^i}
> \end{equation*}
> The Best Response flow (4) can be computed in an outer loop for $T$ steps using an explicit Euler scheme
> \begin{equation*}
>     \nu_{t+1}^N = (1-\alpha h_2)\nu_{t}^N + \alpha h_2\Psi_{\sigma}[\nu^N_t].
> \end{equation*}
> Note that since $\nu_{t+1}^N$ is a convex combination of probability measures, we are guaranteed that $\nu_{t+1}^N$ stays a probability measure.
>
> The output of the algorithm is then the final distribution of the parameters of the network $\nu_{T}^N = \frac{1}{N}\sum_{i=1}^N \delta_{X_T^i}.$
>
> For $T$ large enough, we expect due to our convergence results that $\nu_T^N \to \nu_{\sigma}^{\ast},$ and consequently due to the softmax form of the policy $\pi_\nu,$ the learned policy $\pi_{\nu_{\sigma}^{\ast}}$ is approximately equal to the optimal policy $\pi_\tau^{\ast}.$

---

> > ### Comment · Reviewer_tmuc · 2025-08-02
> >
> > I appreciate the authors' comprehensive response to my previous comments.  The revisions appear to have adequately addressed the concerns I initially raised.  To further strengthen the manuscript and preempt potential reader inquiries, however, I would recommend incorporating a more detailed discussion of $\sigma$ in the main text.  I elect to retain my original assessment score.

---

> > > ### Author Response · Authors · 2025-08-03
> > > **Official Comment by Authors**
> > >
> > > We thank the reviewer for considering our rebuttal. We will include the discussed changes in the camera-ready version if the paper is accepted.

---

### Decision · Program_Chairs · 2025-09-17

**Decision:**

Accept (poster)

**Comment:**

This paper studies minimizing non-convex functionals over probability measure spaces. It extends convergence guarantees of the Best Response (BR) flow from convex to non-convex functions with certain assumptions. The authors proved global convergence of BR flow to a unique minimizer in single-agent problems or mixed Nash equilibrium in min-max problems, under appropriate regularization. It is a theory heavy paper without much empirical validations. But overall reviewers are content with the theoretical contributions extending recent results on BR flow to non-convex (and non-concave) problems, and consider the mathematical development is thorough and solid. Clarify of the writings could be improved. Reviewer YdjA asked for more clarity around novelty and positioning of the work in the literature. The authors were able to provided updated draft including these discussions. The reviewer then in turn raised the recommendation.